EMBO
Molecular Medicine

# CDK12/CDK13 inhibition disrupts transcriptional elongation and replication fork progression in glioblastoma

Silje Lier[1,2,15], Sara B Markusson [1,2,3,15], Anja Kocijancic [1], Martine Narum[2], Solveig O Lund[1,3], Bianka Böllering [4], Anuja Lipsa[5], Mirra L C Søegaard [3,6], Idun D Rein[7], Petra Santha[1,3], Preeti Jain[1,3], Anna Lång[1,3], Emma Lång[1,3], Niklas Meyer[1,2], Aparajita Dutta[8], Santosh Anand [9], Sugith B Badugu[10], Gaute J Nesse[1,3], Rune J Forstrøm[1,3], Arne Klungland [1,3], Ashish Anand[11], Steven M Pollard [12], Stig O Bøe[1,3], Johanne E Rinholm[1,2], Katrin B M Frauenknecht [4,13], Anna Golebiewska [5], Simone P Niclou[5,14], Kumar Somyajit [10], Mads Lerdrup [3,6]✉ & Deo P Pandey [1,3]✉

## Abstract

Glioblastomas are the most prevalent and aggressive malignant brain tumors, characterized by hypertranscription and dependence on neurodevelopmental transcription factors. The transcriptional cycle is regulated by phosphorylation of the C-terminal domain (CTD) of RNA polymerase II (RNAPII) by transcriptional cyclin-dependent kinases (tCDKs), including CDK7, CDK9, CDK12, and CDK13. Here we find that glioblastoma stem cells (GSCs) are selectively sensitive to CDK12/CDK13 inhibition, whereas CDK7 and CDK9 inhibition cause non-specific cytotoxicity. This selective targeting halts GSC and organoid proliferation, curtails GSC invasion and suppresses tumor growth in a xenograft mouse model. In GSCs, CDK12/CDK13 inhibition leads to a rapid and genome-wide loss of serine-2 phosphorylation (pSer2) of the RNAPII CTD, abolishing transcriptional elongation and a transcriptional program sustained by key neurodevelopmental transcription factors. CDK12/CDK13 inhibition unexpectedly arrests DNA replication and replication fork progression in a manner distinct from the effect of inhibiting other tCDKs. This dramatic arrest precedes DNA damage response activation and cell cycle arrest, directly linking RNAPII elongation to replication fork dynamics and revealing a previously unrecognized dependence of DNA replication on CDK12/CDK13-RNAPII regulation.

**Keywords** Transcriptional Addiction; Transcriptional Cycle; Transcriptional Cyclin-Dependent Kinases (tCDKs); DNA Replication; Glioblastoma

Subject Categories Cancer; Neuroscience

## Introduction

IDH-wildtype glioblastoma is the most prevalent and aggressive tumor of the central nervous system and has a median overall survival of 13.9 months and a 5-year survival rate of 5.3% (Brennan et al, 2013; Verhaak et al, 2010). The standard of care for glioblastoma patients is surgical resection followed by radiotherapy and chemotherapy with temozolomide (TMZ) (Hegi et al, 2005), which increases median overall survival by 1.7 months compared to radiotherapy alone (Perry et al, 2017). It is therefore critical to identify new specific vulnerabilities of glioblastomas that can be targeted pharmacologically.

Aberrant transcriptional programs are key to the development and maintenance of cancer cells, which become addicted to a repurposed and specific transcriptional machinery, and they are often hypersensitive to targeting of these addictions (Bradner et al, 2017). Glioblastomas are characterized by hypertranscription, which is driven by RNAPII dysregulation resulting from the altered occupancy of RNAPII at enhancers and promoters, leading to excessive gene expression (Henikoff et al, 2023; Henikoff et al, 2025). Furthermore, glioblastoma propagation and resistance to existing therapies are driven by stem cell-like states, which depend on neurodevelopmental transcription factors (TF) to maintain a specific transcriptional program and sustain

[1]Department of Microbiology, Rikshospitalet, Oslo University Hospital, Oslo, Norway. [2]Institute of Basic Medical Sciences and Institute of Clinical Medicine, University of Oslo, Oslo, Norway. [3]CRESCO, Centre for Embryology and Healthy Development, University of Oslo, Oslo, Norway. [4]Institute of Neuropathology, University Medical Center, Johannes Gutenberg University Mainz, Mainz, Germany. [5]NORLUX Neuro-Oncology Laboratory, Department of Cancer Research, Luxembourg Institute of Health, Strassen, Luxembourg. [6]Center for Chromosome Stability, Department of Cellular and Molecular Medicine, University of Copenhagen, Copenhagen, Denmark. [7]Department of Core Facilities, Institute for Cancer Research, Oslo University Hospital, Oslo, Norway. [8]National Institute of technology, Silchar, India. [9]Division of Biological Sciences, University of Missouri, Columbia 65211 MO, USA. [10]Department of Biochemistry and Molecular Biology, University of Southern Denmark, Odense, Denmark. [11]Department of Computer Science and Engineering, Indian Institute of Technology, Guwahati, Assam, India. [12]Institute for Regeneration and Repair & Cancer Research UK Scotland Centre, Edinburgh, United Kingdom. [13]National Center of Pathology, Laboratoire national de santé, Dudelange, Luxembourg. [14]Department of Life Sciences and Medicine, University of Luxembourg, Belval, Luxembourg. [15]These authors contributed equally: Silje Lier, Sara B Markusson. ✉E-mail: mlerdrup@sund.ku.dk; deopan@ous-hf.no

proliferation (Mack et al, 2019; Suva et al, 2014). Glioblastoma hypertranscription, driven by dysregulated transcription elongation (Miller et al, 2017; Qiu et al, 2022), enhancer activity, and neurodevelopmental transcription factor activation, sustains tumor proliferation and survival, highlighting the transcriptional machinery as a promising therapeutic target.

RNAPII-dependent transcription is required for the transcriptional programs that sustain specific cell lineages and identities, including that of glioblastomas. The transcription cycle of RNAPII is regulated by tCDKs, including CDK7, CDK9, CDK12, and CDK13, that phosphorylate the RNAPII CTD and facilitate key steps of transcription initiation and elongation (Chou et al, 2020). CDK7 is involved in regulating transcription initiation by phosphorylating serine-5 (pSer5) of the RNAPII CTD (Christensen et al, 2014). CDK9, CDK12, and CDK13 phosphorylate serine-2 (pSer2) of RNAPII CTD, regulating transcription elongation (Olson et al, 2018; Zhang et al, 2016). Targeting cancer hypertranscription by inhibiting tCDKs offers an attractive therapeutic approach, and several highly specific inhibitors were recently reported (Chou et al, 2020; Vervoort et al, 2022), including the CDK9-inhibitor NVP-2, the allosteric CDK7-inhibitor THZ1, and the allosteric CDK12/CDK13-inhibitor THZ531. These inhibitors target a cysteine residue outside the kinase domain, thereby resulting in much higher specificity (Kwiatkowski et al, 2014; Zhang et al, 2016). THZ1-mediated CDK7 inhibition leads to loss of RNAPII phosphorylation mainly at Ser5 and has anti-cancer properties in adult and pediatric glioma cells (Greenall et al, 2017; Meng et al, 2018; Nagaraja et al, 2017). THZ531 treatment reduces RNAPII phosphorylation mainly at Ser2, and can reduce neuroblastoma, osteosarcoma, and Ewing sarcoma proliferation (Bayles et al, 2019; Krajewska et al, 2019). Furthermore, the specific CDK12/CDK13 inhibitor SR-4835 reduces the proliferation of triple-negative breast cancer cells (Quereda et al, 2019).

Hypertranscription and transcriptional addiction in glioblastoma create a vulnerability that can be exploited by targeted inhibition of tCDKs, which potentially disrupts oncogenic transcriptional programs and impairs tumor survival, thereby offering a promising therapeutic strategy. Here, we explore whether tCDK inhibitors can inhibit glioblastoma cell proliferation using a panel of glioblastoma patient-derived stem-like cells (GSCs). We demonstrate that glioblastoma cell proliferation is specifically and strongly reduced due to immediate loss of RNAPII phosphorylation, transcriptional shutdown, and disruption of a glioblastoma-specific transcriptional program using THZ531 and SR-4835 that inhibit CDK12/CDK13 and reduce RNAPII pSer2 (Quereda et al, 2019; Zhang et al, 2016). Finally, we demonstrate a new function of CDK12/CDK13 in cell cycle and replication fork progression that is distinct compared to CDK9, revealing a new link between replication and RNAPII elongation. Altogether, our results present a promising therapeutic strategy for glioblastoma by simultaneously halting DNA replication in GSCs and suppressing their hyperactive transcriptional program.

# Results

## Inhibition of CDK12/CDK13 arrests glioblastoma cell proliferation

Provided the transcriptional addiction of glioblastomas and the limited prior testing of tCDK inhibitors for their ability to suppress glioblastoma, we examined the dose-response effect of small-molecule inhibitors against CDK7, CDK9, and CDK12/CDK13 on the viability of GSCs and non-GBM cancer cells (Figs. 1A and EV1A–C). We found that compared to non-GBM cancer cells, GSCs were exceptionally vulnerable to the two small-molecule CDK12/CDK13 inhibitors THZ531 and SR-4835 (Figs. 1A,B and EV1C,D). In contrast, CDK7 and CDK9 inhibitors, THZ1 and NVP-2, unselectively reduced the viability of both GSC and non-GBM cancer cells (Fig. EV1A,B). Importantly, all GSCs tested were sensitive to THZ531 and SR-4835 treatment, and their $IC_{50}$ values, ranging from 20 to 200 nM were substantially lower than those of cells from other cancer types (Figs. 1B and EV1C). GSCs are grown in serum-free media, while control non-GBM cancer cells are cultured in serum-containing media. To determine whether the presence or absence of serum affects the response to CDK12/CDK13 inhibition, we tested both conditions and found that the response was independent of serum in the culture media (Fig. EV1E). In agreement with a recent study reporting that THZ531 inhibited proliferation of liver cancer cells (Wang et al, 2020), we also found human hepatoma HepG2 cells to be sensitive to THZ531 (Fig. 1A). Moreover, 100 nM and 500 nM THZ531 led to a strong reduction in proliferation and colony formation for the GSCs, but not for Hela cells, which exhibit similarly rapid proliferation with high levels of endogenous DNA damage (Figs. 1C,D and EV1D). Next, we examined whether the THZ531-mediated effect on proliferation is caused by the induction of apoptosis or the activation of the DNA damage response (DDR). To this end, we performed immunoblotting to detect apoptosis or DDR activation using the well-known markers cleavage of PARP and γH2AX, respectively. We found that treatment with THZ531 for 24 hours (h) or longer induced apoptosis and DNA damage in the GSCs G7 and G144, whereas 6 h treatment had minimal effects. In contrast, HeLa cells were unaffected under the same conditions (Fig. 1E). Additionally, induction of apoptosis in GSCs after 24 h of THZ531 treatment was confirmed using a TUNEL assay (Fig. 1F).

To further validate the results obtained using small-molecule inhibitors, we investigated the effect of genetic ablation of CDK12/CDK13 on the proliferation of GSCs using a CRISPR/Cas9-based competition assay. Positive control sgRNA against MCM2, RPS19, and CDK9 inhibited glioblastoma cells, whereas a negative control had no effect. Importantly, each of the three independent sgRNAs targeting CDK12 or CDK13, revealed that genetic ablation of these targets significantly inhibited the glioma cell proliferation (Fig. 1G). In summary, these experiments demonstrate that GSCs are highly sensitive to the CDK12/CDK13 inhibitors THZ531 and SR-4835.

## CDK12 and CDK13 are expressed in human glioblastoma patients

To explore the clinical relevance of CDK12 and CDK13 in glioblastoma, we analyzed the relationship between the expression levels of various tCDKs and the survival outcomes of glioblastoma patients. For this, we utilized GlioVis (Bowman et al, 2017) to analyze and visualize gene expression of large-scale datasets from TCGA (The Cancer Genome Atlas) to glioblastoma and other brain tumors. Our analyses revealed that glioblastoma patients with higher CDK13 expression have significantly poorer overall survival (HR = 0.66; $p = 0.0058$). In contrast, no significant correlation was observed for other tCDKs (Fig. EV2A), suggesting that inhibiting

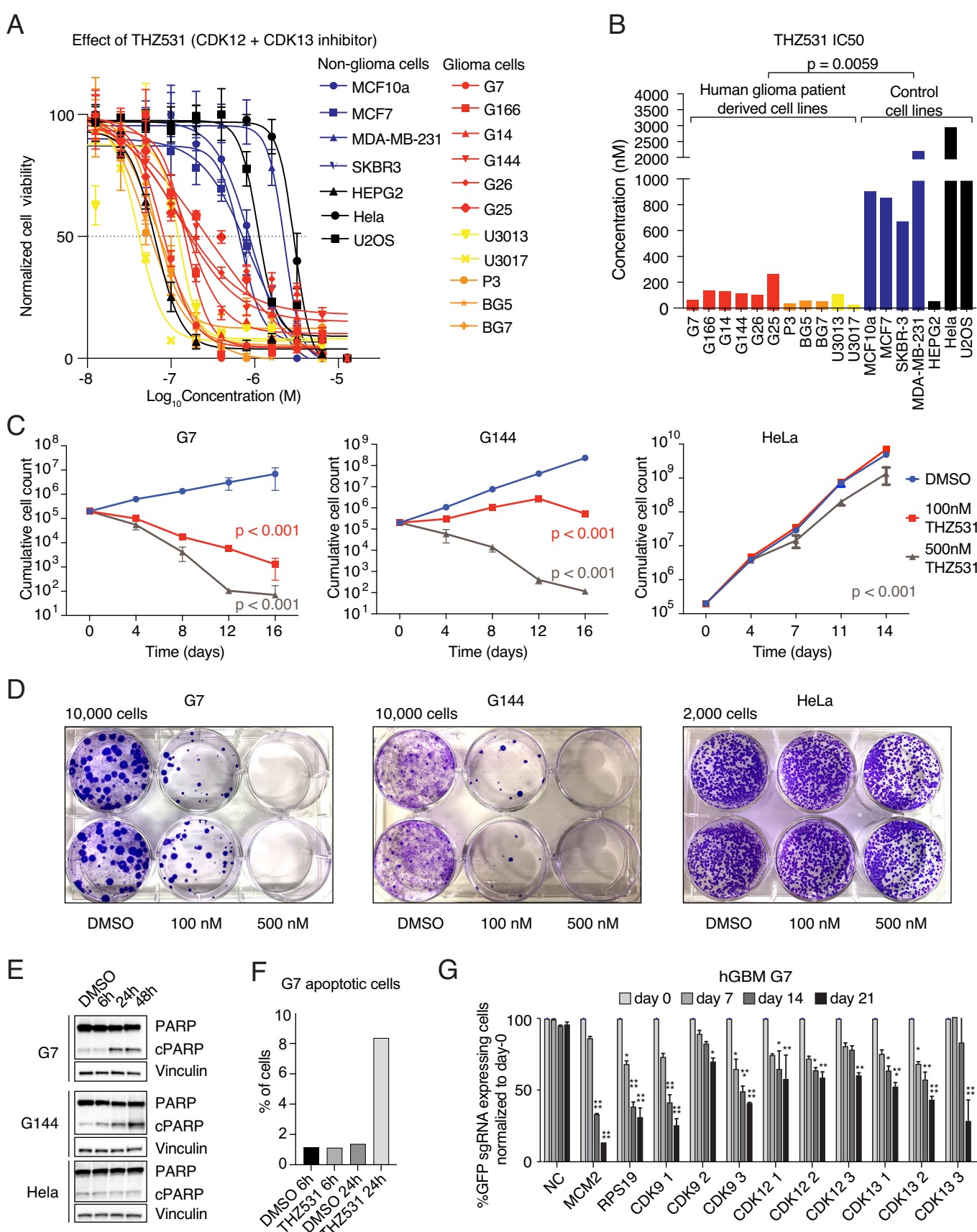

Figure 1. Inhibition of CDK12 and CDK13 specifically affects the proliferation of glioblastoma cells.

(A) Dose-response curves from MTT assays for eleven high-grade glioblastoma and seven non-glioblastoma cell lines treated with THZ531. Data represent mean ± SD of three replicates. (B) Bar graph showing IC50 values from the MTT assays in (A). Statistical analyses confirm that the IC50 values for GSCs are significantly different from those of non-GSC cells. Data were analyzed by the Mann–Whitney $t$-test. (C) In vitro cell proliferation assay of the GSCs G7 and G144, and HeLa cells treated as indicated. Data represent mean ± SD of three replicates. Data were analyzed by two-way ANOVA followed by Tukey's multiple comparisons test. Exact $p$ values are provided in Appendix Table S4. (D) Clonogenic survival assay of G7, G144, and HeLa cells treated as indicated. (E) Immuno-blot analyses from G7, G144, and HeLa cells that were treated with DMSO, 500 nM THZ531 for 6, 24, and 48 h to assess total and cleaved PARP as a marker of apoptosis, with Vinculin used as a loading control. (F) Flow cytometry quantification of the TUNEL assay showing the percentage of apoptotic G7 cells treated with 500 nM THZ531 for 6 and 24 h. (G) Competition assays of sgRNAs targeting CDK9, CDK12 and CDK13, as well as positive controls (essential genes, MCM2 and RPS19). Non-targeting sgRNA (NC) was used as a negative control. Data were analyzed by two-way ANOVA followed by Tukey's multiple comparisons test. *$p < 0.05$, **$p < 0.01$, ***$p < 0.001$, ****$p < 0.0001$, significant differences as compared to the day 0. Exact $p$ values are provided in Appendix Table S4. Source data are available online for this figure.

CDK13 may potentially affect glioblastoma. We next evaluated CDK12 and CDK13 expression in glioblastoma patients using immunohistochemistry (IHC), using our in-house developed specific antibodies, as multiple commercial antibodies were tested but found unsuitable for IHC assays. Additionally, we utilized Ki-67 as a marker for cell proliferation. Five glioblastoma tissue samples, two CNS tissue samples without glioblastoma and one peripheral tissue with fasciitis were subjected to CDK12, CDK13 and Ki-67 immunohistochemistry (Appendix Table S1). In fasciitis (C12), we observed CDK12 and CDK13 expression mainly in the nuclei of inflammatory cells. In glioblastoma tissue (C17, C19, and C21), we found heterogeneous nuclear expression of CDK12 and CDK13, ranging from low to high levels (Fig. 2A,B). Glioblastoma tumor tissues stained positively for Ki-67, whereas control CNS tissue did not show Ki-67 expression (Fig. EV2B). Omission of the primary antibody did not lead to any staining of tissue structures (Fig. EV2C). Following analyses of Ki-67 and CDK12 expression in the three glioblastoma cases, we found that CDK12 levels correlated with proliferative activity, showing a moderate but significant association (Fig. EV2D). Furthermore, we examined the mRNA expression of tCDKs in GSCs and non-GBM cancer cells used in our study and found all of these to be expressed (Fig. EV2E). Collectively, we find that higher CDK13 expression is associated with poorer survival outcomes in glioblastoma patients and both CDK12 and CDK13 are well expressed with nuclear localization in glioblastoma.

## Inhibition of CDK12/CDK13 compromises the survival of glioblastoma organoids

We next assessed the effect of CDK12/CDK13 inhibitors on glioblastoma patient-derived organoids. We used short-term cultures of mechanically dissociated tumor tissue to generate primary organoids, followed by intracranial implantation as orthotopic xenografts (Golebiewska et al, 2020; Oudin et al, 2021). To standardize treatment readouts, we used a methodology in which organoids are reconstituted solely from glioblastoma cells isolated from patient-derived xenografts after removing micro-environmental components, yielding compact, heterogeneous, and size-uniform organoids suitable for reproducible drug testing (Ermini et al, 2025; Klein et al, 2020; Salvato et al, 2025). Four glioblastoma organoid models were selected, exhibiting a typical range of key glioblastoma genetic alterations, including deletion of CDKN2A/B, amplification of CDK4/6 and EGFR, mutations of TP53, PTEN, PIK3CA, and EGFR (Fig. EV2F). To benchmark the effect of the CDK12/CDK13 inhibitors THZ531 and SR-4835, we

included two inhibitors of clinical significance for glioblastoma treatment: Lomustine and Abemaciclib. Glioblastoma organoids were subjected to these four inhibitors for 72 h. Organoid growth and viability was evaluated via image-based assessment of their size and structure as well as ATP levels proportional to the numbers of viable, metabolically active cells. Organoids showed notable sensitivity to THZ531, SR-4835, and abemaciclib. All three compounds strongly affected the morphology of the organoids and led to a dose-dependent decrease in viability at 72 h post-treatment (Fig. 2C). Organoids were most sensitive to SR-4835, while the effect of THZ531 was comparable to Abemaciclib (Fig. 2D,E). All organoids were relatively resistant to Lomustine, which led to a well-preserved organoid structure and only a minor decrease in viable cell numbers. These findings were consistent with responses observed in GSCs (Fig. EV2G,H). Altogether, we find that CDK12 and CDK13 are expressed in glioblastoma patients and that their inhibition reduced organoid proliferation.

## Inhibition of CDK12/CDK13 suppresses GBM migration, invasion, and tumor growth in vivo

Cell motility is central for the invasive capacity of malignant glioblastoma (Cuddapah et al, 2014), and we therefore assessed the effect of THZ531 on GSC motility compared to the EGFR-inhibitor Gefitinib serving as a positive control (Lång et al, 2018). High-content live-cell microscopy revealed that THZ531 significantly reduced the movement of G7, G144, G14, and G166 GSCs to an extent that even exceeded the ability of Gefitinib (Figs. 3A and EV2I; Movies EV1 and EV2). Next, we investigated the effect of THZ531 on glioblastoma invasion using H2B-mCherry-expressing G7 spheroids, which were added to murine organotypic brain slices to assess the invasive potential through fluorescence tracking. We then assessed their invasive capacity by measuring the distance these cells migrated under control conditions or different concentrations of THZ531 and found that THZ531 treatment robustly decreased the glioblastoma migration and invasion (Fig. 3B,C). Additionally, THZ531 treatment also reduced the number of cells in the spheroids in line with the previously observed diminished GSC and organoid proliferation (Figs. 1A–D and 2C,D).

To investigate the effect of CDK12/CDK13 inhibitors on tumor growth in vivo, we decided to use the CDK12/CDK13 inhibitor SR-4835, as it previously had been used against triple-negative breast tumor xenografts in mice (Quereda et al, 2019), while THZ531 has not yet been used in vivo. We first determined that the maximum tolerated dose of SR-4835 in mice was 20 mg/kg, as higher doses

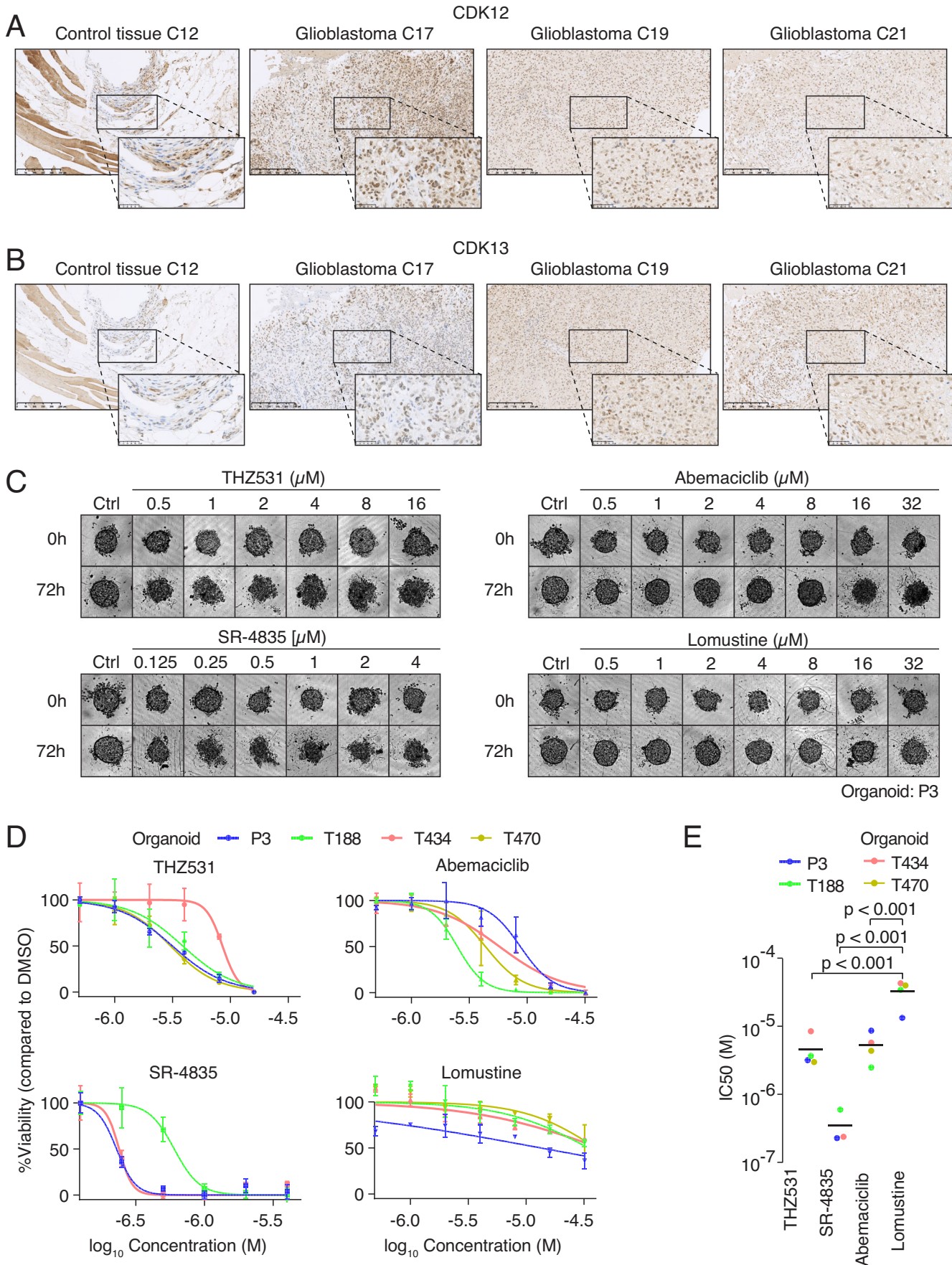

**Figure 2. CDK12 and CDK13 are expressed in glioblastoma patients and inhibition of CDK12/CDK13 compromises organoid viability.**

(A) Representative images of CDK12 immunohistochemistry in control and glioblastoma tissue. The leftmost image shows control staining, while the next three images (from left to right) depict CDK12 expression in three different glioblastoma patients (for information on patients, see Appendix Table S1). Scale bar: 50 μm. (B) Representative images of CDK13 immunohistochemistry in control and glioblastoma tissue. Same as in panel (A). (C) Representative images of organoids showing the effect of different inhibitors with indicated doses after 72 h treatment. (D) Dose-response curves of individual organoid models, displayed as % of viability normalized to untreated DMSO control ± SEM (nonlinear fit, n = 3 per treatment dose, n = 18 per DMSO control). (E) Dot-plot showing mean IC50 values upon exposure to inhibitors in four organoid models tested with one-way ANOVA, n = 4 per treatment. P values compared to Lomustine. THZ531: p = 0.00000978; SR-4835: p = 0.00000364; Abemaciclib: p = 0.0000131. (*p value <0.05, ***p value <0.01, ****p value <0.001). Source data are available online for this figure.

(30 mg/kg daily or on alternate days) were toxic. Mass spectrometry of brain tissue collected 24 h after the final dose showed concentrations below the limit of quantification in all but one sample, indicating that SR-4835 does not effectively cross the blood–brain barrier (BBB) (Appendix Table S2). Therefore, we used a mouse subcutaneous xenograft model based on U87-MG cells to test the efficacy of the CDK12/CDK13 inhibitor SR-4835 and compared it to TMZ treatment. SR-4835 and THZ531 reduced in vitro U87-MG proliferation with IC50 similar to what we observed for GSCs (Fig. 3D,E). Nine days after injection with U87-MG cells, mice were dosed with SR-4835 or TMZ for two weeks (Fig. 3F). Growth of subcutaneous tumors was strongly inhibited by 20 mg/kg SR-4835, 5 mg/kg, or 2 mg/kg TMZ (Fig. 3F). Importantly, the constant body weight of the mice indicated that all treatments were well tolerated. Finally, we tested the effect of SR-4835 on a large panel of cancer cell lines encompassing pancreatic, ovarian, uterine and prostate cancer, and found that GSCs were most sensitive to CDK12/CDK13 inhibition (Fig. 3G). Altogether, we found that inhibition of CDK12/CDK13 has a strong and specific effect on glioblastoma growth, which compares favorably with the existing treatment.

## Inhibition of CDK12/CDK13 leads to global loss of RNAPII CTD phosphorylation in GSCs

To establish the underlying molecular mechanisms, we first analyzed the phosphorylation level of key residues in the RNAPII CTD following THZ531 treatment. Using 500 nM THZ531 as in previous studies of other cell types (Dubbury et al, 2018; Fan et al, 2020; Krajewska et al, 2019; Zhang et al, 2016), we found that 6 h of treatment almost completely abolished Ser2 phosphorylation and strongly affected pThr4 in GSC G7 cells, while the total RNAPII levels remained unchanged up to 24 h (Fig. 4A). In contrast, no substantial changes were observed in the levels of phosphorylated species of RNAPII within 48 h of treatment for HeLa cells (Fig. 4A). Additionally, we employed quantitative image-based cytometry (QIBC) (Somyajit et al, 2017; Toledo et al, 2013) to confirm that 6 h of THZ531 treatment strongly reduced RNAPII pSer2 levels (Fig. 4B), with moderate increase in total RNAPII levels (Fig. EV3A).

To investigate the effect of CDK12/CDK13 inhibition on genome-wide chromatin occupancy of total RNAPII, pSer2 and pSer5 in glioma cells, we performed CUT&RUN (Meers et al, 2019). Total RNAPII occupancy transiently increased at genes with preexisting RNAPII at transcription start sites (TSSs) after 1 h of THZ531 treatment and was similar to that of control-treated cells after 6 h of treatment (Fig. 4C). After 6 h of treatment, pSer5 levels were considerably reduced, whereas chromatin occupancy of pSer2

was almost completely lost throughout the genome. Control markers H3K27ac and H3K27me3 were enriched at TSSs of actively transcribed and silent genes, respectively (Fig. EV3B,C). In summary, these data show that THZ531 treatment strongly reduced genome-wide levels of phosphorylated RNAPII species.

## THZ531 treatment disrupts nascent mRNA expression in GSCs

Serine-2 phosphorylation is associated with elongating RNAPII, which is a critical step for transcription. Utilizing QIBC of EU (5-ethynyl uridine) to evaluate global levels of nascent RNA synthesis, we found that after 6 h of THZ531 treatment, there was a marked reduction in bulk nascent RNA synthesis, which even exceeded the inhibitory effect observed with triptolide treatment (a well-known transcription inhibitor) (Fig. 4D). To identify genome-wide consequences for nascent and steady-state transcriptomes in THZ531-treated cells, we used SLAM-seq (Muhar et al, 2018). In brief, uridines of nascent transcripts were labeled using a short pulse of 4-thiouridine (4sU) and isolated RNA was then exposed to iodoacetate, which converts incorporated uridine to cytosine. T→C conversion was used to identify nascent transcripts and compare the patterns of nascent and steady-state transcription in THZ531-treated and control G7 cells. This established that THZ531 strongly suppressed nascent mRNA synthesis (Fig. 4E). Indeed, exposure to THZ531 for 6 h caused a near total loss of newly transcribed mRNAs and had a strong impact on the composition of the steady-state transcriptome, with thousands of mRNAs being significantly up- and down-regulated (Fig. 4F,G). As expected, the expression of mRNAs showed a very high level of concordance with total RNAPII occupancy as well as RNAPII phosphorylation (Fig. 4H,I). In line with the observation that THZ531 nearly completely blocked nascent transcription, the most highly expressed transcripts with the highest levels of TSS-associated RNAPII, were also most strongly downregulated by THZ531 (Fig. 4F,G). THZ531 treatment for 6 h nearly completely abolishes pSer2 levels and considerably reduces pSer5 levels, implying that the observed effects on gene expression and transcription may not be exclusively caused by pSer2 inhibition. To assess the specific contributions of Ser2 and Ser5 phosphorylation to transcription regulation and gene expression changes caused by THZ531, we analyzed how changes in pSer2 and pSer5 levels correlated with nascent mRNA levels. Our analyses revealed that the reduction in pSer2 levels was more strongly correlated with the decrease in nascent mRNA levels compared to the reduction in pSer5 levels (Fig. EV3D,E). Taken together, THZ531 treatment profoundly disrupts transcription by targeting RNAPII phosphorylation, particularly at the Ser2 residue.

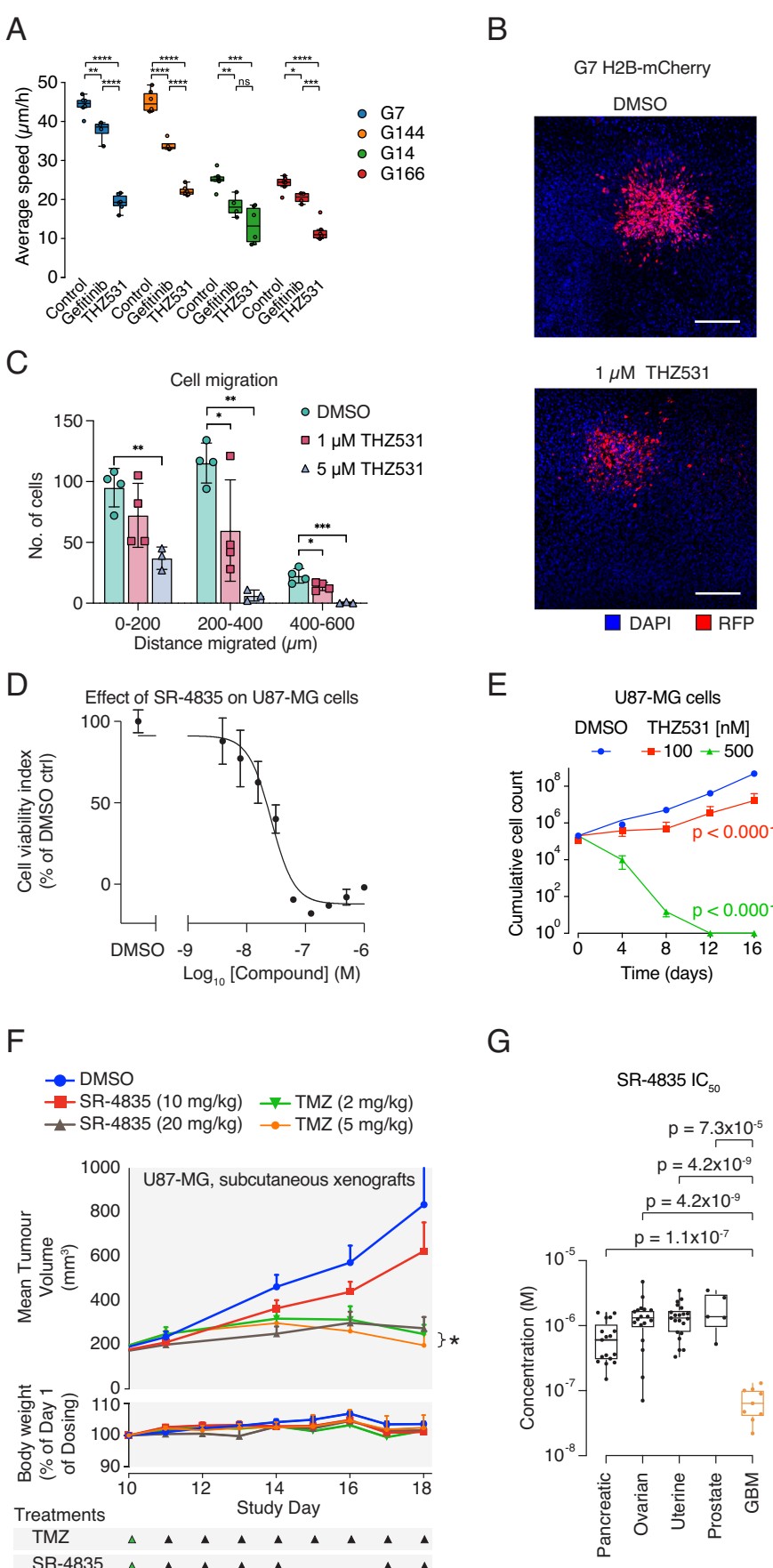

**Figure 3.   Inhibition of CDK12/CDK13 reduces glioblastoma migration and tumor burden in vivo.**

(A) Boxplot representing the average migration speed of four high-grade glioblastoma cells treated as indicated. The EGFR inhibitor Gefitinib was used as a positive control. Each box in the graph represents average migration speeds in 4–6 acquired time-lapse movies. The boxplot is representative of three independent experiments, and was plotted using the standard Seaborn boxplot and stripplot packages in Python. Lines within boxes indicate the median (Q2), bounds of boxes are the interquartile range (IQR) (25th (Q1) to 75th percentiles (Q3)), and the whiskers represent $Q1 - 1.5 \times IQR$ and $Q3 + 1.5 \times IQR$. Individual data points that fall outside the range of the whiskers are plotted as dots. Differences between groups were compared with an independent t-test, using the SciPy library. $*p < 0.05$, $**p < 0.01$, $***p < 0.001$, $****p < 0.0001$. Exact p values are provided in Appendix Table S4. (B) Representative images illustrating the invasion potential of H2B-mCherry-expressing G7 spheroids in murine organotypic brain slices under control conditions and THZ531 treatment. Scale bar: 200 μm. (C) Quantification of invasion distance under control conditions and THZ531 treatment. Data were presented as mean ± SEM. Statistical significance was calculated using one-way ANOVA with Dunnett post hoc test. $*p < 0.05$, $**p < 0.01$, significant differences as compared to the control. At 0–200 μm: DMSO vs 5 μM: $p = 0.0081$. At 200–400 μm: DMSO vs 1 μM: $p = 0.0424$; DMSO vs 5 μM: $p = 0.0017$. At 400–600 μm: DMSO vs 1 μM: $p = 0.0331$; DMSO vs 5 μM: $p = 0.0002$. $n = 4$ slices per condition, except for 5 μM drug, where $n = 3$ slices. (D) Dose-response curve for U87-MG glioma cells treated with SR-4835. Data represent mean ± SD of three replicates. (E) in vitro cell proliferation assay of U87-MG cells treated as indicated. Data represent mean ± SD of three replicates. Data were analyzed using a two-way ANOVA followed by Tukey's multiple comparisons test, significant differences as compared to the control at 16 h; 100 nM THZ531: $p = 0.00000000028$; 500 nM THZ531: $p = 0.00000000017$. (F) Mean tumor volume and relative body weight of mice over time for the indicated treatments with the dosage shown at the bottom, $+SEM$, $n = 8$ mice per group. There was a significant decrease in tumor volume of animals treated with 5 mg/kg Temozolamide versus animals treated with 20 mg/kg SR-4835: $p = 0.0418$. (G) Boxplot showing IC50 values for the dose response of SR-4835 inhibition on a panel of cancer cell lines. Bounds of boxes show the IQR, lines within boxes represent the median, and whiskers represent the minima to maxima. Pancreatic cell lines: $n = 17$; ovarian cell lines: $n = 18$; uterine cell lines: $n = 21$; prostate cell lines: $n = 5$; GBM cell lines: $n = 9$. P values are from two-sided t-tests Benjamini–Hochberg corrected for multiple testing. Source data are available online for this figure.

## THZ531 represses glioblastoma-specific transcription factors and their targets

We then investigated the impact of THZ531 treatment on distinct categories of transcripts to determine how it affects the expression of genes involved in critical cellular processes. Gene Ontology analyses revealed profound consequences of THZ531-mediated inhibition of CDK12/CDK13. In THZ531-treated GSCs (G7 and G144), the most strongly enriched functional categories among downregulated nascent and steady-state transcripts included transcription, cell cycle, and RNA metabolism (Figs. 4J and EV3F). Upregulated steady-state transcripts were enriched in biological processes, such as translation and metabolism (Fig. EV3F) in accordance with previous observations (Bayles et al, 2019; Krajewska et al, 2019). We also investigated selected transcript sets, including those encoding DDR regulators and BRCA-ness factors, which are known to depend on CDK12 for their expression (Dubbury et al, 2018; Krajewska et al, 2019). For comparison, we also examined transcripts encoding ribosomal genes, histones, cell cycle regulators and housekeeping genes, along with TFs that sustain the proliferation of glioma cells and their targets (Fig. 4K). Nascent transcripts encoding regulators of cell cycle, DDR, BRCA-ness factors or core glioma TFs and their target genes were significantly enriched among the downregulated genes (Figs. 4K,L and EV3G; Appendix Fig. S1A). In contrast, we found that nascent transcripts expressed from housekeeping genes were not significantly enriched or depleted among upregulated, down-regulated, or unchanged genes.

Transcripts encoding key neurodevelopmental TFs, including OLIG2, POU3F2 and SOX2, which are required for proliferation of glioma cells, were among the most strongly downregulated transcripts (Fig. 4L). The downregulation or ablation of these TFs has been shown to strongly suppress the proliferation of glioma cells (Suva et al, 2014). We therefore investigated the change in their expression and occupancy of total, pSer2, and pSer5 forms of RNAPII following THZ531 treatment. All three genes were highly expressed in G7 cells and were marked by the presence of super-enhancer sites containing broad H3K27ac domains, as reported earlier (Mack et al, 2019; Suva et al, 2014) (Appendix Fig. S2A).

While pSer2 was nearly abolished at these genes and pSer5 was strongly reduced, total RNAPII was also reduced following 6 h of CDK12/CDK13 inhibition. We observed a strong downregulation in the nascent transcripts for these genes, which was validated using RT-qPCR. We found that the expression of key DDR genes was strongly downregulated following THZ531 treatment, consistent with previous reports (Appendix Fig. S2B). In addition, the expression of many glioblastoma-specific neurodevelopmental TFs, including OLIG2, POU3F2 and SOX2 target genes, was profoundly downregulated (Appendix Fig. S2C), with OLIG2 target genes being most strongly affected. While the expression of many, but not all, housekeeping genes remained unchanged, cell cycle regulators showed a marked downregulation (Appendix Fig. S2D,E). In concert, these results demonstrate that THZ531-mediated inhibition of CDK12/CDK13 strongly suppresses expression of TFs (and their target genes) required for the glioblastoma transcriptional program.

Several functional categories related to the cell cycle were enriched among the most downregulated nascent transcripts in THZ531-treated G7 cells, as well as in the steady-state transcripts downregulated in G7 cells (Figs. 4J and EV3F). Subsequently, we evaluated the impact of THZ531 treatment on the transcriptome of GSCs, stratified based on their expression across cell cycle phases. Both G7 and G144 GSCs exhibited comparable responses to THZ531, and overlapping sets of differentially expressed transcripts were readily identified (Fig. EV4A). Publicly available data on cell cycle dependence of gene expression in HeLa and U2OS cells (Bostrom et al, 2017) were used to assess cell cycle dependence of transcriptional changes following THZ531 treatment in GSCs. Consistent with the impact of THZ531 on overall nascent RNA synthesis, these analyses indicated a downregulation of nascent transcripts throughout the cell cycle, with the most pronounced decrease observed in THZ531-treated G7 cells after S-phase (Fig. EV4B,C). Interestingly, prior to S-phase, the steady-state transcripts were both up- and downregulated, whereas after S-phase, the majority of the steady-state transcripts were downregulated (Fig. EV4B,C). Overall, we find that THZ531 treatment robustly suppresses nascent mRNA synthesis, significantly reducing transcription of key genes involved in cell cycle regulation, DNA

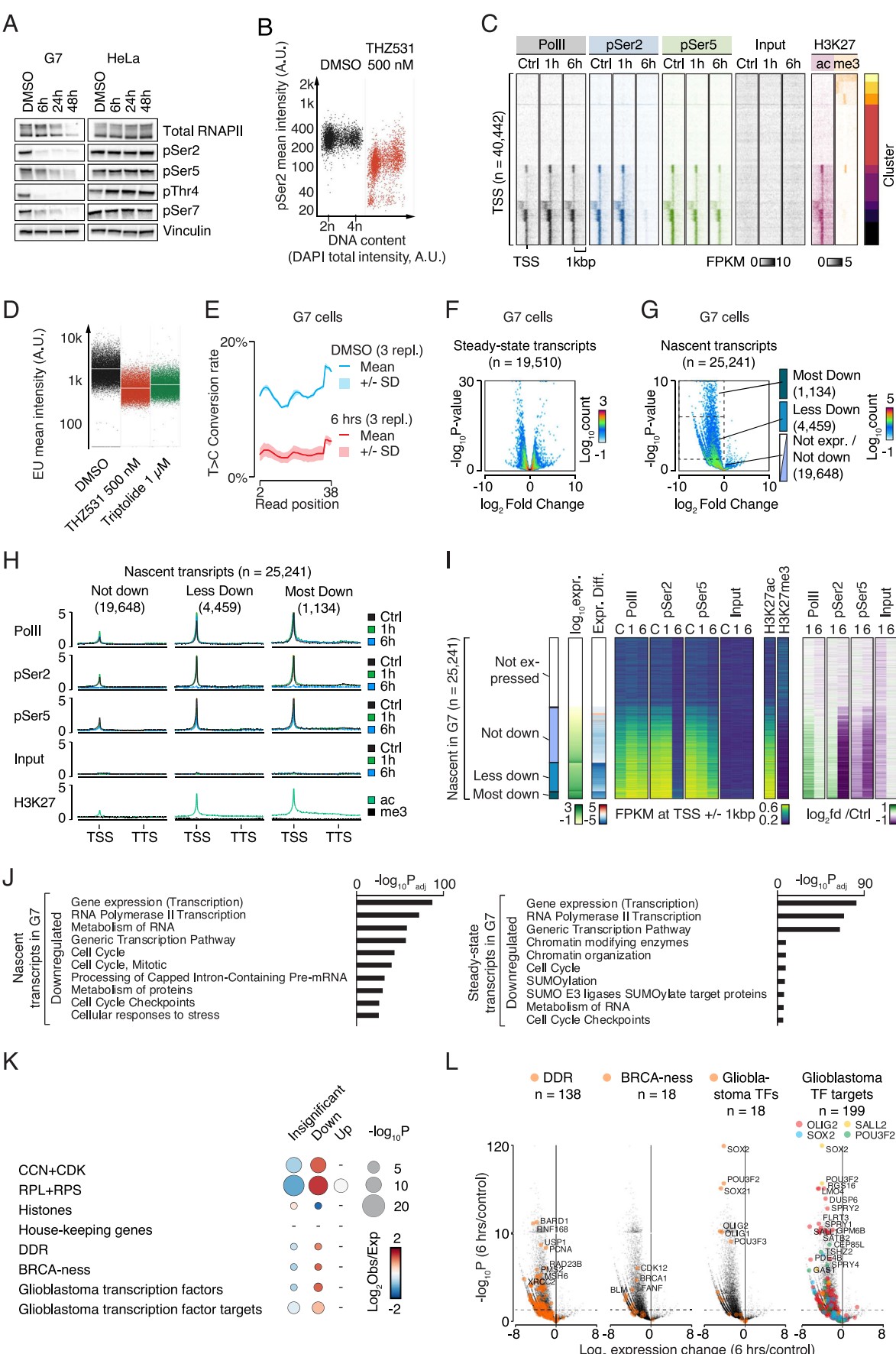

**Figure 4. CDK12/CDK13 inhibition profoundly affects RNAPII phosphorylation and transcription in GSCs.**

(A) Immuno-blot analyses from G7 and HeLa cells that were treated with vehicle, 500 nM THZ531 for 6, 24, and 48 h for the various RNAPII CTD species. The figure is representative of >3 replicates. (B) QIBC analysis of pSer2 for G7 cells treated with vehicle or 500 nM THZ531 for 6 h. (C) Heatmaps of Cut&Run signal from RNAPII, RNAPII phosphorylation states, and histone modifications at k-means clustered unique TSSs ± 1 kbp. G7 cells were treated with DMSO and THZ531 for 1 and 6 h. (D) QIBC analyses of EU incorporation in G7 cells treated with vehicle, 500 nM THZ531 or 1 µM Triptolide for 6 h. (E) Diagram showing T→C conversion in three replicates of SLAM-seq data from G7 cells treated with THZ531 or DMSO for 6 h. (F, G) Volcano plots showing the overall transcriptional differences in G7 cells treated with THZ531 for 6 h compared to DMSO control for steady-state (F) and nascent transcripts (G). X-axis shows the log2 fold differences. Y-axis shows the -log10 $p$ values derived using two-tailed Wald tests and Benjamini–Hochberg corrected for multiple testing. Dashed rectangles: populations of interest. (H) Graphs of average Cut&Run signal from RNAPII, RNAPII phosphorylation states, and histone modifications at and around TSS and TTS of genes. Levels are FPKM normalized. (I) Heatmaps illustrating transcription in relationship to Cut&Run data of RNAPII, RNAPII phosphorylation states, and histone modifications at TSSs. Groups from Fig. 4G. (J) Bar diagrams of Benjamini–Hochberg corrected $-\log10$ $p$ values from most enriched gene ontology terms in nascent or steady-state transcripts downregulated after 6 h of THZ531 treatment in G7 cells. (K) Bubble diagram of enrichment of gene groups within upregulated, downregulated or unchanged categories. $P$ values are Benjamini–Hochberg corrected. (L) Volcano plots as in Fig. 4G with the indicated gene populations colored. $P$ values were calculated with Fisher's one-tailed tests adjusted for multiple testing using g:Profiler's default g:SCS algorithm. Source data are available online for this figure.

damage response, and glioblastoma-specific neurodevelopmental transcription factors, including OLIG2, POU3F2, and SOX2. This inhibition profoundly impacts the glioblastoma transcriptional program, with cell cycle-dependent transcriptional changes and a pronounced downregulation of nascent transcripts observed after S-phase, while housekeeping gene expression remains largely unaffected.

## THZ531 treatment disrupts GSC cell cycle progression

To investigate the effect of CDK12/CDK13 inhibition on the cell cycle of glioma cells, we used the nucleoside analog of thymidine, 5-ethynyl-2′-deoxyuridine (EdU), which is incorporated into DNA during active DNA synthesis. Following 6 h THZ531 treatment, there was very little EdU incorporation in both the G7 and G144 cells, indicating a lack of DNA synthesis. This effect persisted for up to 24 h (Fig. 5A). Furthermore, we noticed a slight increase in cells in both G1- and G2-phases following the pronounced loss of cells in S-phase undergoing active DNA synthesis (Fig. 5B). This was further confirmed when cells were co-treated with THZ531 and nocodazole. While nocodazole-treated cells expectedly arrested in M phase, GSCs treated with THZ531 did not show this accumulation, suggesting that THZ531 specifically blocks cell cycle progression prior to G2/M phase (Fig. EV4D,E). Importantly, there was no change in the abundance of γH2AX after 6 h of exposure to THZ531; accumulation of γH2AX only occurred at later stages (Fig. EV4F). Moreover, apoptotic cells did not significantly increase after 6 h of THZ531 treatment but accumulated after 24 h (Fig. 1E,F). Together, this reveals that THZ531 suppresses DNA replication without causing DNA damage or apoptosis initially. Strikingly, no reduction in EdU incorporation was observed in HeLa cells after 6 h of THZ531 treatment, suggesting that the inhibition of DNA replication is specific to GSCs at the concentration used (Fig. EV4G).

To further characterize the mechanistic effect of THZ531 on DNA replication, we measured its impact on active DNA synthesis. We performed a side-by-side comparison with the potent CDK9-inhibitor NVP-2 and with hydroxyurea (HU), which depletes dNTP pools, causes replication-fork stalling, and induces replication stress. QIBC of EdU incorporation in G7 cells revealed that THZ531 treatment led to an inhibition of DNA synthesis that was nearly as potent as observed for HU, while NVP-2 treatment had minimal impact (Fig. 5C,D).

Given the distinct roles of CDK12, CDK13, and CDK9 in DNA replication, yet their shared regulation of RNAPII to control transcriptional programs, we investigated whether the reduction of cell cycle or DDR-gene expression (Fig. 4L; Appendix Fig. S2B) influences the impact of CDK12 and CDK13 on DNA synthesis. The cell cycle-specific transcriptional program is tightly regulated by RB:E2F and DREAM complexes, which control distinct transcriptional subgroups active during the G1/S and G2/M phases (Bertoli et al, 2013; Fischer et al, 2022). CDK-mediated phosphorylation of Rb (pRB) and Forkhead box M1 (pFOXM1) activates these G1/S and G2/M programs, respectively, as visualized in Fig. 5E, serving as established markers of their activation. Using pRB and pFOXM1 levels, we examined the impact of THZ531 on drivers of the cell cycle-specific transcriptional program and found that their levels stayed largely unchanged across the cell cycle, while there was no change in these markers following CDK9 inhibition or HU treatment (Fig. 5F,G). These findings indicate that while THZ531 impedes global GSC transcription, it does not specifically disrupt CDK-dependent phosphorylation events required for transcriptional activation of cell cycle programs. These observations imply a novel RNAPII-dependent mechanism in DNA replication uniquely governed by CDK12/CDK13.

## CDK12/CDK13 inhibition disrupts replication fork progression without triggering replication stress or cell cycle checkpoint activation

Finally, to understand the mechanisms underlying the immediate and robust DNA replication block after CDK12/CDK13 inhibition, we examined whether THZ531 treatment triggers replication stress by activating the DDR or cell cycle checkpoints. We compared this response to NVP-2 and HU treatments. Consistent with the lack of γH2AX induction following THZ531 treatment (Figs. 6A and EV4F), we found largely unchanged levels of FK2, which detect ubiquitinated proteins commonly associated with double-strand breaks (DSBs), and 53BP1, that is a key effector in the DNA damage response (Fig. EV5A,B). Moreover, we observed no significant changes in the levels of the G1/S marker p21 (Fig. EV5C), suggesting that CDK12/CDK13 inhibition does not activate this checkpoint pathway. We also investigated whether the dNTP pool was depleted after THZ531 treatment using expression of the major regulator of dNTP metabolism, RRM2 (ribonucleotide reductase M2) in cells and found that the RRM2 expression did

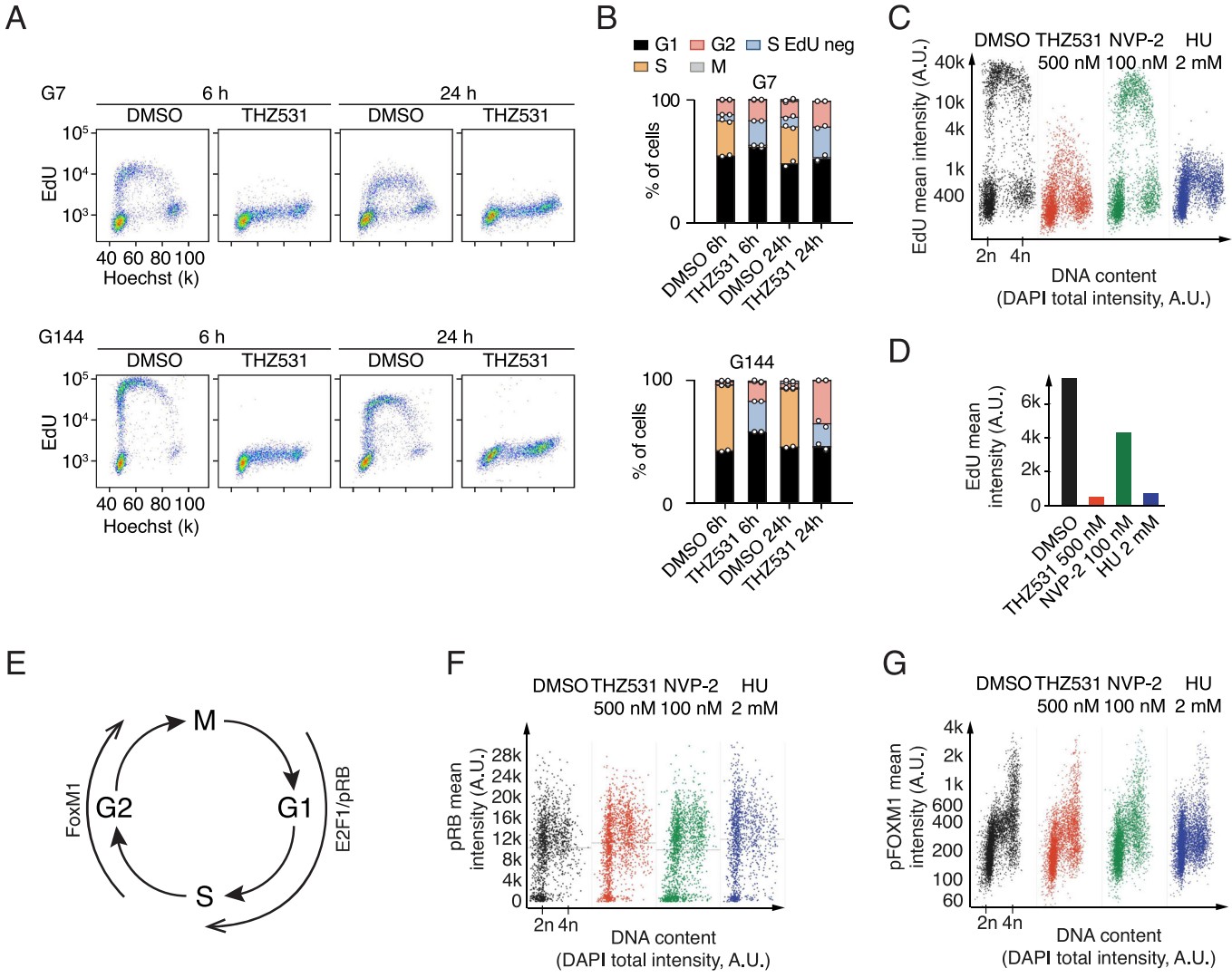

**Figure 5. CDK12/CDK13 inhibition disrupts cell cycle progression and DNA synthesis in GSCs.**

(A) Cell cycle analyses in G7 and G144 cells after 6 and 24 h treatment with DMSO or 500 nM THZ531. 10 μM EdU was added 1 h prior to harvest. Dot plots show intensity of EdU relative to DNA content (Hoechst) in interphase cells (one representative replicate shown). (B) Bar diagrams of cell cycle distributions of G7 and G144 cells treated as indicated. Data are presented as mean with a range from two repeats. (C) QIBC analyses of EdU incorporation (Y-axis) relative to DNA content (X-axis) in G7 cells under vehicle control, 500 nM THZ531 treatment for 6 h, and control treatments with 100 nM NVP-2 and 2 mM HU. (D) Quantification of EdU for treatments in (C). (E) A schematic representation depicting the key regulators of the cell cycle phase-specific transcriptional program. (F, G) QIBC analyses of pRB (F) and pFOXM1 (G) levels (Y-axis) relative to DNA content (X-axis) in G7 cells under vehicle control, 500 nM THZ531 treatment, and control treatments with 100 nM NVP-2 and 2 mM HU, respectively. Source data are available online for this figure.

not change upon CDK12/CDK13 inhibition (Fig. EV5D). Single-strand breaks (SSB) in DNA lead to phosphorylation of the single-strand binding protein RPA, which then forms foci at the site of the break, making it suitable as a marker for SSBs. After 6 h of THZ531 treatment, we did not detect any changes in phosphorylated RPA (pRPA) levels (Fig. EV5E), indicating that THZ531 did not induce SSBs at this time point. In summary, THZ531 treatment did not activate any of the three replication stress markers examined in our study, affect the dNTP pool, trigger SSBs, or activate the G1/S checkpoint.

γH2AX binds to DSBs, and it's accumulation serves as a marker for these breaks, providing an indirect measure of DNA damage. To more directly confirm that THZ531 treatment induces breaks in

DNA, including DSBs or SSBs, we employed comet assays under two different conditions: A neutral comet assay to detect DSBs, and an alkaline comet assay to detect both DSBs and SSBs. We found that THZ531 treatment did not induce DSBs or SSBs, whereas the topoisomerase inhibitor and positive control Etoposide did induce breaks in DNA as seen by the percentage of tail DNA in the alkaline comet assay (Fig. 6B,C). Similarly, we did not observe any induction of DSBs following THZ531 treatments in the neutral comet assay (Fig. EV5F,G). Finally, we performed DNA fiber assays to measure the speed and stability of DNA replication forks and evaluate the effects of CDK12/CDK13 inhibition on DNA replication relative to CDK9 inhibition. Our results showed that treatment with THZ531 significantly slowed replication fork

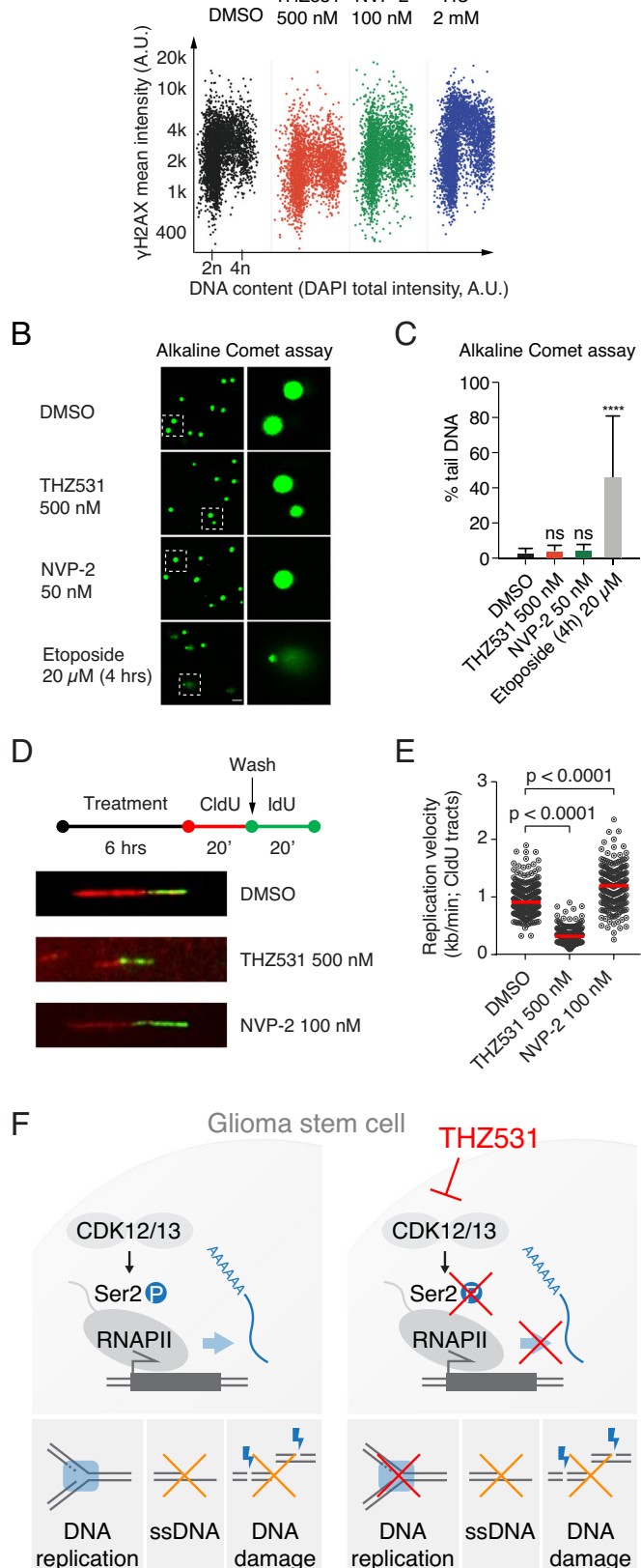

A QIBC analyses of γH2AX and EdU incorporation (Y-axis) relative to DNA content (X-axis) in G7 cells under vehicle control, 500 nM THZ531 treatment, and control treatments with 100 nM NVP-2 and 2 mM HU.

**Figure 6. CDK12/CDK13 inhibition does not trigger replication stress while disrupting replication fork progression.**

(A) QIBC analyses of γH2AX and EdU incorporation (Y-axis) relative to DNA content (X-axis) in G7 cells under vehicle control, 500 nM THZ531 treatment, and control treatments with 100 nM NVP-2 and 2 mM HU. (B) Alkaline comet assays showing tail movements depicting DNA damage in G7 cells for vehicle control, 500 nM THZ531 treatment, and control treatments with 50 nM NVP-2 and 20 μM Etoposide. Scale bar: 100 μm. (C) Quantification of the tail moments. Approximately 100 cells were measured per sample. Data were presented as mean ± SD of duplicates from three biological replicates. Statistical analysis was done using Kruskal–Wallis test with Dunn's multiple comparisons correction *$p < 0.05$, **$p < 0.01$, ***$p < 0.001$. ****$p < 0.0001$, significant differences as compared to the control: Etoposide (4 h) 20 μM: <0.000000000000001. (D) Schematic representation of the DNA fiber assays depicting treatments with vehicle, 500 nM THZ531, and 100 nM NVP-2 for 6 h, incorporating thymidine analogs CldU and IdU for 20 min. (E) Quantification of replication fork progression is shown for vehicle, 500 nM THZ531 and 100 nM NVP-2 treatments for 6 h. Replication fork speed was quantified by analyzing at least 200 DNA fibers per condition ($n = 205$). The lengths of red (CldU) and green (IdU) labeled segments were measured using Fiji ImageJ software (National Institutes of Health). Fork speed (kb/min) was calculated by multiplying the measured length in μm by a conversion factor of 2.59 kb/μm and dividing by the labeling pulse duration. Data were analyzed using a two-way ANOVA followed by Tukey's multiple comparisons test ****$p < 0.0001$, significant differences as compared to the control: THZ531 500 nM: $p < 0.000000000000001$; NVP-2 100 nM: $p < 0.000000000000001$. (F) Schematic model illustrating how CDK12/CDK13 inhibition blocks serine-2 phosphorylation of the RNAPII CTD and nascent mRNA transcription, leading to impaired glioblastoma proliferation and disrupted replication fork progression, which occurs without inducing replication stress or activating cell cycle checkpoints. Source data are available online for this figure.

progression, indicating a strong inhibitory effect on DNA replication, whereas NVP-2 treatment accelerated replication fork progression (Fig. 6D,E). In summary, our findings, as illustrated in the model in Fig. 6F, demonstrate that inhibition of CDK12/CDK13 disrupts glioblastoma cell proliferation and replication fork progression without inducing replication stress or activating cell cycle checkpoints, in addition to blocking serine-2 phosphorylation of RNAPII CTD and nascent mRNA transcription.

## Discussion

In the present study, we identify an efficient and specific way to target and disrupt the transcriptional program required for glioblastoma cell proliferation and migration. Specifically, we show how small-molecule inhibitors targeting CDK12/CDK13, which phosphorylate RNAPII, strongly perturb the transcriptional landscape in GSCs. This leads to rapid and robust downregulation of multiple glioblastoma-associated transcription factors and their targets, and subsequently a strong inhibition of GSC-proliferation, cell cycle arrest, and stalled replication fork progression, ultimately inducing cell death. The clear dependence of GSCs on CDK12 and CDK13 for proliferation was validated using CRISPR/CAS9-mediated genetic ablation of CDK12 and CDK13. We demonstrate that CDK12 and CDK13 are well-expressed in human glioblastoma tissue ranging from weak to moderate to strong expression. Moreover, the CDK12/CDK13 inhibitors affected morphology and reduced the survival of organoids, in a favorable manner

compared to CDK4/CDK6 inhibition and Lomustine, both of which are used to treat glioblastoma patients. Using GSC spheroids and murine organotypic brain slices, we demonstrated that THZ531 treatment reduced migration and invasive capacity of GSCs as well as their proliferation. While our results show that the CDK12/CDK13 inhibitor SR-4835 is well-tolerated in mice, it fails to cross the blood–brain barrier. However, the observed reduced tumor burden in a subcutaneous xenograft mouse model, encourages further tests in more clinically relevant in vivo models as well as identification of novel CDK12/CDK13 inhibitors that can pass the blood–brain barrier.

The primary activity of CDK12 and CDK13 is to phosphorylate Ser2 of RNAPII CTD (Chou et al, 2020). Exposure to THZ531 rapidly and nearly completely abolishes RNAPII Ser2 phosphorylation and strongly reduces the abundance of other RNAPII phospho-species in GSCs, with no similar effect in control cells. Furthermore, the chromatin occupancy of pSer2-RNAPII and pSer5-RNAPII is strongly reduced in THZ531-treated glioblastoma cells, while total RNAPII chromatin occupancy remains unaffected. Phosphorylation at Ser2 of the RNAPII CTD is a crucial marker of transcription elongation (Chou et al, 2020), and CDK12/CDK13 inhibition leads to a rapid loss of pSer2 levels, resulting in a near-total shutdown of nascent mRNA synthesis and large-scale changes in steady-state mRNA expression. Glioblastoma relies on aberrant activation of neurodevelopmental transcription factors (Suva et al, 2014) to drive RNAPII-dependent transcription, and THZ531 strongly suppresses these factors, disrupting transcriptional programs essential for GSC proliferation.

We find that CDK12/CDK13 inhibition has a profound impact on the expression of genes involved in transcription and cell cycle regulation. Concordantly, THZ531 rapidly suppresses active DNA replication. However, apoptosis and DNA damage are observed after 24 h exposure to THZ531, but not after 6 h. These observations suggest that THZ531 compromises DNA synthesis primarily, and that this subsequently may lead to DNA damage and apoptosis. At the same time, we find that the expression of key cyclin genes together with core DDR genes is profoundly downregulated within 6 h treatment of THZ531, explaining the remarkable arrest of cell cycle progression. Glioblastomas have constitutive and aberrant activation of the DDR pathway, showing high genomic instability (Bao et al, 2006; Bartkova et al, 2010; Rasmussen et al, 2016), and dependence on expression of several key regulators of the DDR pathways, including BRCA1, for their survival (Rasmussen et al, 2016). Our transcriptomic analyses revealed that key DDR components are suppressed rapidly and strongly by CDK12/CD13 inhibition, providing a potential explanation for the remarkable sensitivity of GSCs compared to other cancer cells (Fig. 3G).

Since the initial identification of THZ531 in 2016, there has been great interest in understanding the requirement and roles of CDK12/CDK13 in cancer cells. Furthermore, loss or mutation of CDK12 is reported for several cancers, including ovarian, breast and prostate cancers (Ekumi et al, 2015; Johnson et al, 2016; Wu et al, 2018). These studies correlate loss of CDK12 with altered expression of core DNA damage response genes. Recent studies associated expression changes after loss of CDK12/CDK13 with gene length, expression level, and intronic polyadenylation cleavage of affected mRNAs (Bayles et al, 2019; Dubbury et al, 2018; Krajewska et al, 2019). In agreement with other studies focusing on

breast cancer and neuroblastomas (Bayles et al, 2019; Dubbury et al, 2018; Krajewska et al, 2019; Quereda et al, 2019), we find that the inhibition of CDK12/CDK13 strongly downregulated the expression of core DDR genes. Furthermore, our observations are in agreement with recent findings in K562 chronic myeloid leukemia cells, where inhibition of CDK12/CDK13 results in a global loss of RNAPII CTD phosphorylation and extensive genome-wide transcriptional changes (Fan et al, 2020). Moreover, similar results are obtained when CDK12 is inhibited in HEK-293 cells (Tellier et al, 2020). At higher concentrations, we observe that the proliferation of control cells is also affected by both CDK12/CDK13 inhibitors, THZ531 and SR-4835. Therefore, other cancer cells potentially require a higher dose of CDK12/CDK13 inhibitors to observe the global changes in RNAPII phosphorylation and subsequent shutdown of nascent transcription. It will be relevant for future work to understand the factors that govern the differences leading to sensitivity of CDK12/CDK13 inhibition among different cancers.

THZ531 treatment caused a rapid and robust block in DNA replication without triggering DNA damage or replication stress. Therefore, we characterized early responses at 6 h post-treatment. The absence of γH2AX and other double-strand break (DSB) markers, such as FK2 and 53BP1, indicated no DSB induction. Using multiple assays to measure SSBs and DSBs, including phosphorylated RPA (pRPA) foci formation and neutral and alkaline comet assays, we further confirmed that THZ531 did not cause single- or double-strand DNA breaks. Additionally, p21 levels remained unchanged, showing no activation of cell cycle checkpoints, and stable RRM2 levels suggested no reduction in the dNTP pool. These data emphasize that THZ531 inhibits DNA replication without causing direct damage to DNA.

CDK12/CDK13 inhibition drastically slows replication fork progression as shown by DNA fiber assays despite the lack of activation of DDR pathways, suggesting their unique role in maintaining replication fork stability independently of classical DNA damage signaling. In contrast, we find that CDK9 inhibition accelerated replication fork progression, highlighting distinct effects of tCDKs on DNA replication dynamics. This is consistent with previous studies showing that transcriptional inhibitors like DRB or Triptolide accelerate replication fork speed without activating DNA damage responses (Kotsantis et al, 2016; Petropoulos et al, 2024). The distinct effect of CDK12/CDK13 inhibition underscores their essential function in regulating transcription and DNA replication, distinguishing them from other transcriptional regulators. Further investigation into their role in DNA replication dynamics is crucial for understanding their therapeutic potential in cancer.

In summary, our study demonstrates the therapeutic potential of targeting transcriptional addiction in glioblastoma through CDK9 and CDK12/CDK13 inhibition, with CDK12/CDK13 suppression showing greater selectivity for glioblastoma inhibition. Unlike CDK9 inhibition, which accelerates replication fork progression, CDK12/CDK13 inhibition disrupts DNA replication without triggering DNA damage responses, linking transcription elongation to replication regulation. Identification and further characterization of small-molecule inhibitors targeting CDK12/CDK13 with improved pharmacological properties, in particular the ability to cross the blood–brain barrier, would have a large therapeutic potential for glioblastoma treatment.

# Methods

### Reagents and tools table

| Reagent/resource | Reference or source | Identifier or catalog number |
|---|---|---|
| **Experimental models** | | |
| GSCs (G7, G14, G144, G166, G25, G26, G30) (Human) | Gift from S.M. Pollard (Pollard et al, 2009) (Engström et al, 2012) | N/A |
| GSCs (U3013, U3017) (Human) | HGCC, Uppsala (Xie et al, 2015) | N/A |
| GSCs (P3, BG5, BG7) (Human) | Gift from Rolf Bjerkvig (Wang et al, 2009) | N/A |
| Glioblastoma organoids (Human) | This study, (Golebiewska et al, 2020; Oudin et al, 2021) | N/A |
| U87 MG (Human) | Gifted from Prof. Pål Kristian Selbo, Oslo University Hospital | HTB-14 |
| HeLa (Human) | ATCC | CCL-2 |
| U2OS (Human) | ATCC | HTB-96 |
| HepG2 (Human) | Gifted from Dr. Ingrun Alseth, University of Oslo | HB-8065 |
| Cancer cell lines for Fig. 3G (pancreatic, ovarian, uterine, prostate) | Crown Bioscience, Inc. (Leicester, UK) | Table EV3 |
| MCF10a (Human) | Gifted from Prof. Ragnhild Eskeland, University of Oslo | N/A |
| MCF-7, SKBR-3, MDA-MB-231 (Human) | Gifted from Dr. Gunnhild Mælandsmo, Oslo University Hospital | N/A |
| 293FT (Human) | Thermo Fisher Scientific | R70007 |
| Athymic nude mice | Envigo/Crown Bioscience, Inc. (Leicester, UK) | N/A |
| **Recombinant DNA** | | |
| sgRNA sequences for competition assays | This study | Table EV1 |
| U6-sgRNA-SFFV-puro-P2A-EGFP | (Müller et al, 2021) | Addgene: #57827 |
| lentiCas9-Blast | Gifted from Feng Zhang | Addgene #52962 |
| H2B-mCherry | This study | Vector Builder: VB191021-1151hqb |
| psPAX2 | Gift from Didier Trono | Addgene #12260 |
| pMD2.G | Gift from Didier Trono | Addgene #12259 |
| pAG/MNase | Gift from Steve Henikoff | Addgene #123461 |
| **Antibodies** | | |
| Mouse monoclonal, anti-RNA Polymerase II CTD | MBL international | MABI0601, RRID: AB_2728735 |
| Mouse monoclonal anti-Pol II CTD, phospho Ser2 | MBL international | MABI0602, RRID: AB_2747403 |
| Mouse monoclonal, anti-Phospho RNA Polymerase II CTD, phospho Ser5 | MBL international | MABI0603, RRID: AB_2728736 |
| Rabbit monoclonal anti-Acetyl-Histone H3 (Lys27) (D5E4) | CST | #8173 |
| Rabbit monoclonal anti-tri-methyl-histone H3 (K27) (C36B11) | CST | 9733S |
| Rabbit monoclonal anti-gamma H2A.X (phospho S139) [EP854(2)Y] | Abcam | ab81299 |
| Rat monoclonal anti-RNA Pol II CTD phospho Thr4 | Active Motif | 61461 |
| Rat monoclonal anti-RNA pol II CTD phospho Ser7 | Active Motif | 61087 |
| Mouse monoclonal anti-Vinculin | Sigma | V9131 |
| Rabbit polyclonal anti-PARP | CST | 9542 |
| Rabbit polyclonal anti-GAPDH | Santa Cruz Biotechnology | sc25778 |
| Goat anti-rabbit IgG H&L (HRP) | Abcam | ab6721 |
| Rabbit polyclonal anti-human-CDK12 | Abcam | ab246887 |
| In-house rabbit polyclonal anti-CDK12 antibody 1308-1436 | GenScript | Lot: U372FHK070-5/EK0522 |
| In-house rabbit polyclonal anti-CDK13 antibody 58-189 | GenScript | Lot: U372FHK070-5/EK0526 |
| Mouse monoclonal anti-Human Ki-67 | Agilent | GA626 (also GA62661-2) |
| Rabbit polyclonal anti-RFP | BioNordika (Rockland) | 600-401-379 |
| Rat monoclonal anti-BrdU | Abcam | ab6326 |
| Alexa Fluor 594-conjugated goat anti-rat IgG | Thermo Fisher Scientific | A-11077 |
| Purified Mouse monoclonal anti-BrdU (IdU) | Becton Dickinson (BD Biosciences) | 347580, RRID: AB_400326 |
| Alexa Fluor 488-conjugated goat anti-mouse IgG | Thermo Fisher Scientific | A11029 |
| Rabbit monoclonal anti-phospho-Rb (Ser807/811) | Cell signaling | 8516 s |
| Rabbit monoclonal anti-phospho-FoxM Thr600 | Cell signaling | 14655S |
| Rabbit monoclonal anti-phospho-histone H2A.X (Ser139) | Cell signaling | 9718S |
| Mouse monoclonal anti-pS139H2AX (γH2AX) | Millipore | 05-636 |

| Reagent/resource | Reference or source | Identifier or catalog number |
|---|---|---|
| Mouse monoclonal anti-FK2 | Sigma-Aldrich | ST1200 |
| Mouse monoclonal anti-53BP1 | Sigma-Aldrich | Mab3802 |
| Rabbit monoclonal anti-P21 Waf1/Cip1 | Cell signaling | 2947S |
| Mouse monoclonal anti-RRM2 | Santa Cruz | SC398294 |
| Rabbit polyclonal anti-Phospho RPA2 S33 | Thermo Fisher Scientific | A300-246A |
| Goat anti-rabbit Alexa Fluor 488 IgG | Thermo Fisher Scientific | A32731 |
| Goat anti-rabbit Alexa Fluor 568 IgG | Thermo Fisher Scientific | A11036 |
| Goat anti-rabbit Alexa Fluor 546 IgG | Thermo Fisher Scientific | A11010 |
| Donkey anti-rat Alexa Fluor 594 | Thermo Fisher Scientific | A21209 |
| Rabbit polyclonal anti-pS10H3 | Millipore | 06-570 |
| **Oligonucleotides and other sequence-based reagents** | | |
| PCR-primers | This study | Table EV2 |
| **Chemicals, enzymes and other reagents** | | |
| DMSO | MERCK | D8418 |
| THZ531 | MedChemExpress | HY-103618 |
| SR-4835 | MedChemExpress | HY-130250 |
| NVP-2 | MedChemExpress | HY-12214A |
| Temozolamide | Selleckchem | S1237 |
| Abemaciclib | MedChemExpress | HY-16297AR |
| Lomustine | MedChemExpress | HY-13669R |
| Gefitinib | MedChemExpress | Y-50895 |
| Etoposide | Sigma | E1383 |
| Hydroxyurea | Thermo Fisher | A10831.03 |
| DMEM | Gibco | 41966052 |
| DMEM/F-12 | Gibco | 11320033 |
| Neurobasal | Gibco | 21103049 |
| Fetal Bovine Serum | Merck | F7524 |
| NSC Media | In House (Pollard, 2013) | N/A |
| N2 | Fisher Scientific | 17502001 |
| B27 | In House | N/A |
| Poly D lysin | Merck | A-003-E |
| Laminin | Bio-Techne | 3446-005-001 |
| Human EGF | Peprotech | 100-15 |
| Human FGF basic | Peprotech | 100-18B |
| Trypan blue | Invitrogen | T10282 |
| CellTiter-Glo Cell Viability Assay | Promega | G9241 |
| MTT assay | Merck | 11465007001 |

| Reagent/resource | Reference or source | Identifier or catalog number |
|---|---|---|
| Presto blue | Fisher Scientific | A13262 |
| Crystal Violet Stain | Sigma | C6158 |
| Methanol | Sigma | 1179337 |
| Formaldehyde | Sigma | F8775-4X |
| Acetic Acid | Sigma | 695092 |
| CldU | Sigma-Aldrich | C6891 |
| IdU | Sigma-Aldrich | 17125 |
| Triton X | Sigma | T8787 |
| BSA | Sigma | A9418 |
| Methylcellulose | Sigma | M7027 |
| Novex 6% Tris Glycine Gel | Life | XP00062BOX |
| SYBR Green Master Mix | Fisher Scientific | A25742 |
| HEPES | Merck Life Sciences | H3375-25G |
| Spermidine | Merck Life Science | S0266-5G |
| Roche complete tablets | Sigma | 4906837001 |
| Digitonin | Merck Life Sciences | 300410-250MG |
| EGTA | Merck Life Sciences | E4378-10G |
| RNase A, DNase and protease-free (10 mg/mL) | Fisher Scientific | EN0531 |
| Vectashield mounting media | Vector Laboratories | H-1000-10 |
| EnVision FLEX, high pH (Dako Omnis) | Agilent | GV80011-2 |
| EnVision FLEX TRS, low pH (Dako Omnis) | Agilent | GV80511-2 |
| EnVision FLEX Rabbit LINKER (Dako Omnis) | Agilent | GV80911-2 |
| EnVision FLEX Mouse LINKER (Dako Omnis) | Agilent | GV82111-2 |
| Antibody Diluent (FLEX) (Agilent Dako) | Agilent | K800621-2 |
| Hematoxylin (Dako Omnis) | Agilent | GC80811-2 |
| Wash Buffer (20x) (Dako Omnis) | Agilent | GC80711-2 |
| LIVE/DEAD Near-IR | Life Technologies | L10119 |
| Nocodazole | Merck/Sigma | M1404-10MG |
| EU (5-Ethyl Uridine) | Thermo Fisher Scientific | E10345 |
| EdU (5-ethynyl-2'-deoxyuridine) | Thermo Fisher Scientific | A10044 |
| HU (Hydroxyurea) | Sigma-Aldrich | H8627 |
| **Software** | | |
| FIJI/ImageJ | (Schindelin et al, 2012) | N/A |
| TrackMate plugin (FIJI) | (Tinevez et al, 2017) | N/A |
| TIBCO, version 12.4 | https://docs.tibco.com/pub/spotfire/general/sr/sr/topics/spotfire_analyst_desktop_12_4.html | N/A |
| GraphPad Prism 5.0, 10.0 | GraphPad Software Inc., La Jolla, CA, USA | N/A |

| Reagent/resource | Reference or source | Identifier or catalog number |
|---|---|---|
| Galaxy server, SlamDunk pipeline | http://github.com/t-neumann/slamdunk | N/A |
| EaSeq v. 1.2 | (Lerdrup et al, 2016) | N/A |
| g:Profiler | https://biit.cs.ut.ee/gprofiler/, (Raudvere et al, 2019) | N/A |
| R | https://www.R-project.org/ | N/A |
| **Other** | | |
| μ-Slide 8-well Ibidi treat | Inter instrument (IBIDI) | 80826 |
| Gene expression of large-scale datasets from TCGA | The Cancer Genome Atlas | N/A |
| 3' UTR annotations | (Muhar et al, 2018) | N/A |
| Slides scanner | Hamamatsu 99 NanoZoomer 2.0-HT (Hamamatsu Photonics) | N/A |
| Trans blot | Bio-Rad | N/A |
| Comet Assay Kit | Abcam | ab238544 |
| Click-iT Plus EdU Alexa Fluor 594 Flow Cytometry Assay Kit | Fisher Scientific | C10646 |
| Click-iT™ EdU Cell Proliferation Kit for Imaging, Alexa Fluor™ 488 dye | Fisher Scientific | C10337 |
| RNeasy Plus Mini Kit | Qiagen | 74134 |
| QuantSeq 3' mRNA-Seq Library Prep Kit FWD with PCR add-on kit | Lexogen | SKU:015.24 |
| MinElute PCR Purification Kit | Qiagen | 28004 |
| NEBNext® Ultra™ II DNA Library Prep Kit for Illumina | New England Biolabs | E7645 |
| Chemiluminescent substrate kit | Fisher Scientific | 34580 |

## Cell culture

GSC lines and organoid models were derived from IDH1/2-wt glioblastoma patient tumors. GSCs G7, G144, G166, G14, G25, G26, and G30 were a kind gift from S.M. Pollard. GSC cell lines U3013 and U3017 were acquired from HGCC, Uppsala (Xie et al, 2015). GSC cell lines P3, BG5 and BG7 were a kind gift from Rolf Bjerkvig (Wang et al, 2009). All GSC cells were cultured in neural stem cell (NSC) medium supplemented with EGF and FGFb as previously described (Pollard et al, 2009). For culturing GSCs in serum-containing media, serum was added immediately before the viability assay. U87 MG, HeLa, U2OS, and HepG2 cell lines were cultured in DMEM with FBS. MCF10a and breast cancer cell lines, MCF-7, SKBR-3, and MDA-MB-231 were kind gifts from Prof. Ragnhild Eskeland, University of Oslo and Dr. Gunnhild Mælandsmo, Oslo University Hospital, Oslo. Cell lines used were routinely tested for Mycoplasma. Standardized 3D glioblastoma organoids were generated from single cells isolated from patient-derived orthotopic xenografts as described (Ermini et al, 2025;

Golebiewska et al, 2020; Oudin et al, 2021). Four GBM IDH-wt organoid models representing typical GBM alterations (CDKN2A/B loss, CDK4/6 and EGFR amplification, TP53, PTEN, PIK3CA, EGFR mutations) were used. To minimize in vitro drift, tumor cells were freshly isolated from intracranially propagated xenografts and reformed into uniform organoids using a modified protocol (Ermini et al, 2025). Cells were seeded at 1000 per well in 384-well plates and cultured on an orbital shaker for 72 h to generate compact organoids prior to treatment.

## Lentivirus production and transduction

For lentivirus production, 293FT cells were transfected with psPAX2, pMD2.G, and expression constructs, using lipofectamine. After 8 h, cells were washed and cultured in NSC medium. After 48 h, the medium was collected and passed through a 0.45-μm filter. For transduction, GSCs were cultured in NSC medium containing virus particles supplemented with 8 μg/ml polybrene. After 48 h of transduction, cells were passaged and cultured in selection medium for 2 days.

## Cell viability assays

### MTT assay
About 3000–10,000 cells were seeded in 96-well plates and treated with inhibitors. After 72 h, the MTT assay (Merck, 11465007001) was performed, according to the manufacturer's instructions.

### Proliferation assay
About 200,000 cells were seeded in duplicate in six-well plates. Cells were counted with trypan blue and replated every 3–4 days.

### Clonogenic survival assay
About 10,000 cells were seeded in duplicate in six-well plates. After 14 days, the medium was removed, and the cells were given a wash with PBS. Crystal violet staining (0.05% crystal violet, 1% formaldehyde, and 1% methanol in PBS) was added for 20 min (min), after which the cells were rinsed with water and left to air dry.

### Glioblastoma organoid viability assay
Standardized organoids were cultured in 25 μL DMEM with 10% FBS, 2 mM L-glutamine, 0.4 mM NEAA, and 100 U/ml Pen/Strep at 37 °C, and 5% $CO_2$. Organoids were treated for 72 h with THZ531, SR-4835, Abemaciclib, or Lomustine (MedChemExpress) using twofold dilution series (THZ531: 16 μM–500 nM; SR-4835: 4 μM–125 nM; Abemaciclib/Lomustine: 32 μM–500 nM). DMSO served as control ($n = 18$), and all concentrations were tested in triplicate. Organoid morphology was monitored with the Incucyte® system, and viability was assessed using CellTiter-Glo® 3D (Promega); luminescence was read on a ClarioStar (BMG Labtech). Relative viability was normalized to DMSO, and dose-response curves with $IC_{50}$ values were obtained via nonlinear regression (four-parameter variable-slope) in GraphPad Prism.

### Competition assays
Stably expressing Cas9 G7 cells were generated by transducing GSC G7 cells with lentivirus produced from lentiCas9-Blast (Addgene #52962). The sgRNAs were expressed from U6-sgRNA-SFFV-puro-P2A-EGFP, which is derived from (Addgene: #57827) and reported here (Müller

et al, 2021). The sgRNA sequences can be found in Table EV1. The stably expressing Cas9 G7 cells were transduced with the sgRNAs expressing lentiviruses to achieve 40–60% GFP-expressing cells. The transduced cells were subsequently cultured for a period of 21 days, and the population of sgRNA-expressing GFP-positive cells was analyzed at the indicated days by flow cytometry.

## Cell migration experiment

Migration patterns of human GSCs were studied in 96-well glass-bottom plates (Greiner Sensoplate (M4187-16EA, Merck)) after treatment with DMSO control, 500 nM THZ531 or 1.0 µM Gefitinib overnight. Live-cell imaging was performed using an ImageXpress Micro Confocal High-Content microscope equipped with an environmental control gasket, maintaining 37 °C and 5% $CO_2$, and controlled by the MetaXpress 6 software (Molecular Devices). Images were acquired in widefield mode using a $20 \times 0.45$ NA Ph1 air objective, a phase contrast ring, and transmitted light for visualizing the contour of cells. Cell migration was registered for a total time of 8 h. For each well, two sites were imaged with a time interval of 4 min between frames. Acquired time-lapse movies were analyzed using the TrackMate (Tinevez et al, 2017) plugin in the Fiji ImageJ software (Schindelin et al, 2012) and in-house Python-based scripts (Python 3.7.6). An independent two-sample $t$-test, using the SciPy library, was performed in order to compare differences between groups. Statistical significance levels were defined as: $*p < 0.05$, $**p < 0.01$, $***p < 0.001$, $****p < 0.0001$. Python script is available on request.

## Neurosphere invasion experiment

Organotypic mouse brain slices were made from p8 black6 wildtype mice, as previously described (Kennedy and Rinholm, 2017), and cultured for four days before spheroids were added. H2B-mCherry-expressing G7 spheroids were generated using 5000 cells in 20 µL GSC media supplemented with 0.3% methylcellulose by the hanging drop method. Sixteen to 24 h later, they were added on top of the organotypic brain slices, and drug treatment was initiated 2 days later. Drugs were given with medium changes every 2–3 days. Seven days after spheroids were added to the organotypic slices, the slices were fixed in 4% paraformaldehyde and immunolabelled with anti-RFP (BioNordika-Rockland, 600-401-379, 1:200 dilution) to enhance the mCherry.

Six to eight images were captured per slice, with a Zeiss LSM700 confocal microscope using a 40x objective (NA 1.0), covering the neurosphere and the slice area surrounding it. Images were stitched together manually using BigStitcher (Horl et al, 2019) and analyzed using Fiji (Version 2.14.0/1.54 f). Analysis was done in a blinded manner, using automatic counting of particles above 20 µm² in size. Regions of interest (ROIs) were drawn around the center of the neurosphere, with diameters of 200, 400, 600, and 800 µm. Particles were then counted within each ROI. Statistical significance was calculated using one-way ANOVA with Dunnett post hoc test.

## Antibody generation

Antigens for CDK12 (amino acids 1308–1406) and CDK13 (amino acids 1404–1505) were cloned with a His-tag, expressed in *E. coli*, and purified using standard protein purification methods. These purified antigens were then used to immunize rabbits, and the resulting antibodies were purified using standard procedures. Additional details on the antibody characterization are available upon request.

## IHC for CDK12 and CDK13 expression in human glioblastoma tissue

IHC for CDK12 and CDK13 expression was performed using in-house raised polyclonal rabbit antibodies (dilution 1:50). Anonymized FFPE samples from routine diagnostics were analyzed for Ki-67 as well as for CDK12 and CDK13 expression, with limited clinical metadata recorded (Appendix Table S1). IHC was performed on 2–3 µm thick glioblastoma sections ($n = 5$) and non-GBM CNS tissue ($n = 2$) using the Dako Omnis platform. After heat-induced retrieval (EnVision FLEX TRS, low pH, 97 °C, 20 min), slides were incubated with rabbit anti-CDK12 or anti-CDK13 (1:50, 30 min), blocked for endogenous peroxidase, enhanced with EnVision FLEX + Linker, detected using FLEX/HRP polymer and DAB, counterstained with hematoxylin, and mounted. A fasciitis sample (C12) served as control. For the detection of proliferating cells, mouse anti-Ki-67 ready-to-use (RTU) antibody GA626 (Agilent) was used. Slides were scanned on a NanoZoomer 2.0-HT with fixed acquisition settings, and CDK12/CDK13 immunoreactivity (negative–strong) was evaluated. Ki-67-stained whole-slide images were reviewed by a neuropathologist to define low, intermediate, and high proliferation areas and were matched to corresponding CDK12 regions. Three ROIs per proliferation category and patient were analyzed in QuPath (v0.6.0) (Bankhead et al, 2017), using optimized nuclear and DAB thresholds (Pai et al, 2022). ROI-level counts, percentages, and densities of Ki-67- and CDK12-positive cells were exported to R (v4.5.1) for processing using the tidyverse and ggpubr packages. Correlations between markers were assessed using Spearman's ρ, and CDK12 variation across proliferation groups was tested by one-way ANOVA.

## Mouse experiments

SR-4835 (MedChemExpress) was dissolved in 10% DMSO/90% (30%) Hydroxypropyl-b-Cyclodextrin (hp-BCD) and TMZ (Selleckchem), was dissolved in 10% DMSO in distilled water and administered per os (PO) using gavage. SR-4835 was administered daily for 5 consecutive days each week, followed by a 2-day break, whereas TMZ was administered daily. For the combination treatment, compounds were dosed at the same time, with TMZ dosed first, followed by SR-4835.

A subcutaneous U87-MG-luc model was established by Crown Bioscience Inc. (Leicester, UK) using the parental U87-MG cell line from ECACC, transduced in-house to express luciferase. Animals were housed in IVC housing with a 12 h light/dark cycle and access to Teklad 2919 and sterile water ad libitum.

### Tolerability assay

Tolerability studies were performed in a cohort of nine mice (Athymic nude mice from Envigo) treated with SR-4835 at doses of 10, 20, or 30 mg/kg. Both 30 mg/kg regimens (daily and alternate-day dosing) were toxic, whereas 20 mg/kg daily was well tolerated. To assess brain exposure, mass spectrometry was performed on brain tissue collected 24 h after the final dose.

### Efficacy experiments

The in vivo efficacy of the SR-4835 or Temozolamide, was evaluated in the clinically relevant subcutaneous CDX U87-MG-luc xenograft model. These experiments were conducted at Crown Bioscience, Inc. (Leicester, UK) in 7–8-week-old athymic nude mice (Envigo, UK). Forty-eight mice were enrolled in the efficacy study, eight mice per cohort, for six cohorts. Eight million U87-MG-luc cells were injected subcutaneously into athymic nude mice acquired from Envigo. All animals were randomly allocated to the six different studies. Randomization was performed on day 9 post injection, prior to the treatment start. The treatments were undertaken for 2 weeks, with the 5 days on, 2 days off cycle per week for SR-4835 and daily for TMZ. Tumors were measured three times a week, and tumor volumes were estimated by measuring the tumor in two dimensions using electronic callipers.

### Humane endpoints

Any mouse with a tumor volume/measurement at terminal size (e.g., mean diameter 15 mm) was terminated. After one measurement of body weight loss (BWL) >10% on the day of dosing, a treatment break was given, and treatment resumed when the body weight recovered to BWL <5% (compared to day 1 treatment). Any mouse with BWL >15% for 3 consecutive measurements (compared to day 1 treatment) was euthanized, and any mouse with BWL >20% was Schedule 1 culled.

## Ethics

For glioblastoma organoids, tumor tissue samples were collected from patients having given informed consent, and ethical approval has been obtained from the research ethics committee in Luxembourg (National Committee for Ethics in Research (CNER), as described (Golebiewska et al, 2020). The use of fully anonymized human tissue (IHC) was also approved by CNER, PRECISION-PDX, reference number 201201/06. The experiments conformed to the principles set out in the WMA Declaration of Helsinki and the Department of Health and Human Services Belmont Report. Mice experiments were carried out by Crownbio, UK, study number CSU4328, PPL number P85D8ACF6. Animal experiments complied with the UK Animals Scientific Procedures Act 1986 (ASPA) in line with Directive 2010/63/EU of the European Parliament and the Council of 22 September 2010 on the protection of animals used for scientific purposes. The study was conducted in accordance with relevant community standards and reporting guidelines, where applicable.

### SR-4835 dose response

About 3000–5000 cells were seeded in 96-well plates, and the next day, cells were treated with a 3.16-fold dilution in triplicate (31.6, 10, 3.16, and 1 µM and 316, 100, 31.6, 10, and 3.16 nM). After 72 h, the CellTiter-Glo assay (Promega) was performed according to the manufacturer's instructions.

## Immunoblotting

Cells were lysed directly in 1.25X Laemmli sample buffer, sonicated and denatured at 95 °C for 5 min. Samples were loaded and the protein separated in Novex tris-glycine 6% gels (Life Technologies, XP00062BOX) and transferred to nitrocellulose membranes. Membranes were incubated in blocking buffer (5% FBS in TBST

(TBS with 0.2% Tween-20) for 1.5 h at room temperature and incubated with primary antibody (dilutions are indicated in Appendix Table S3) in the blocking buffer overnight at 4 °C. Membranes were given three washes in TBST for 10 min, then incubated for 35 min with appropriate secondary antibodies in blocking buffer and washed for another 3 × 10 min. Chemiluminescent detection was performed using a Chemiluminescent substrate kit from Fisher Scientific.

**qPCR primers** can be found in Table EV2.

## SLAM-seq

SLAM-seq was performed according to the previously described protocol (Muhar et al, 2018). Briefly, cells were seeded at ~70% confluency and treated with THZ531 or vehicle control. 1 h before harvest, 4sU was added to the medium at a final concentration of 500 µM. The samples were kept in the dark as RNA was extracted with Qiagen's RNeasy Plus Mini Kit. 3 µg RNA was alkylated with iodoacetamide (Sigma, 10 mM) for 15 min, and the RNA was repurified by ethanol precipitation. About 250 ng RNA was used to make libraries with Lexogen's QuantSeq 3′ mRNA-Seq Library Prep Kit FWD for Illumina and PCR Add-on Kit for Illumina. Deep sequencing was performed using the NovaSeq platform (Illumina).

## CUT&RUN

Cells were treated with THZ531 or vehicle control before they were harvested, washed and bound to Concanavalin A-coated magnetic beads. The cells were then permeabilized with wash buffer (20 mM HEPES, pH 7.5, 150 mM NaCl, 0.5 mM spermidine, and a Roche complete tablet per 50 ml) containing 0.02% Digitonin. The cell-bead suspension was incubated with 0.5–1 µg of respective antibody in a total volume of 50 µL overnight at 4 °C on a nutator. After three washes with 1 mL Digitonin buffer, cells were resuspended in 50 µL volume with pAG-MNase and nutated for 10 min at RT. Cells were given two washes with Digitonin buffer, chilled on ice, and ice-cold $CaCl_2$ was added, before nutating for 2 h at 4 °C. STOP buffer was added (340 mM NaCl, 20 mM EDTA, 4 mM EGTA, 0.02% Digitonin, 50 µg/mL RNAse A, and 50 µg/mL glycogen), and tubes were incubated at 37 °C for 10 min in a Thermomixer, to release fragments into solution. After centrifugation at $16,000 \times g$ for 5 min at 4 °C, tubes were placed on a magnet stand and the liquid transferred to new tubes. DNA was extracted using Qiagen's MinElute PCR Purification Kit, according to the manufacturer's instructions. Quantification was done by Qubit analysis, and libraries were made with 2 ng DNA using the NEBNext® Ultra™ II DNA Library Prep Kit for Illumina (New England Biolabs, E7645), according to the manufacturer's instructions. Deep sequencing was performed using the NovaSeq platform (Illumina).

## SLAM-seq data processing

3′ UTR annotations were obtained from (Muhar et al, 2018). All further processing was done on the Galaxy server, using the SlamDunk pipeline (http://github.com/t-neumann/slamdunk). Prior to mapping, the quality of the sequencing of the reads was inspected using FastQC (v.0.72 + galaxy1) (https://www.bioinformatics.babraham.ac.uk/projects/fastqc/). Adapters were trimmed from raw reads using cutadapt through the trim_galore

(v.0.4.3.2) wrapper tool with adapter overlaps set to 3 bp. The reads were then processed using SlamDunk (v.0.4.1 + galaxy2). Settings were adjusted to alignment against the human genome (GRCh38), 12 bp trimming from the 5' end, with a multi-mapper retention strategy for 100 alignments, filtering for variants with a 0.2 variant fraction, filtering for a base-quality cutoff of ≥27, and filtering for ≥1 T→C conversions.

## Cut&Run data processing

Prior to mapping, the quality of sequenced reads were inspected using FastQC (v. 0.10.1) (https://www.bioinformatics.babraham.ac.uk/projects/fastqc/), fastqScreen (v. 0.11.4, (Wingett & Andrews, 2018)), and MultiQC (v. 1.7, (Ewels et al, 2016)) and reads were mapped to hg38 using Bowtie2 (v. 2.2.9, (Langmead and Salzberg, 2012)) and the settings "--local --very-sensitive-local --no-unal --no-mixed --no-discordant --phred33 -I 10 -X 700", and filtered for an insert size between 20 and 120 base pairs using samtools (v. 1.10, (Li et al, 2009)) in accordance with instructions from the CUT&RUN protocol (Skene et al, 2018) in the following pipe: "samtools view -h -f 66 | awk -F'\t' 'function abs(x){return ((x < 0.0) ? -x:x)} {if ((abs($9) >= 20 && abs($9) <= 120) || $1 ~ /^@/) print $0}' | samtools view -Sb –". Filtered mapped reads were deduplicated and imported into EaSeq v. 1.2 (Lerdrup et al, 2016) using default settings, and unless specified, subsequent analysis and visualization was performed using the integrated tools in EaSeq.

## Genome-wide data sources

All Cut&Run and SLAM-seq data were deposited at NCBI's Gene Expression Omnibus (Edgar et al, 2002) under the accession number GSE186311. Refseq gene annotations (O'Leary et al, 2016) were acquired from the UCSC table browser (Karolchik et al, 2004). CDK and cyclin genes were identified based on matching the strings "CDK" or "CCN" to gene symbols. RPL/RPS gene symbols were obtained from http://ribosome.med.miyazaki-u.ac.jp/rpg.cgi?mode=orglist&org=Homo%20sapiens, Histone gene symbols were obtained from http://www.informatics.jax.org/mgihome/nomen/gene_name_initiative.shtml, housekeeping gene symbols were obtained from (Eisenberg and Levanon, 2013), DDR-gene symbols were obtained from https://www.mdanderson.org/documents/Labs/Wood-Laboratory/human-dna-repair-genes.html, BRCA-ness gene symbols were obtained from and gene symbols for glioma transcription factors and targets were obtained from (Suva et al, 2014).

## Cut&Run and SLAM-seq visualization and integration

Graphs of average Cut&Run signal as well as heatmaps were generated in EaSeq using the "Average" and "HeatMap"-tools, respectively. K-means clustering was performed using the "Cluster"-tool with the clustering methodology set to k-means, the offset set to ±1kbp and log-transformation disabled. Output from the SLAM-seq processing was analyzed for differential expression using DeSeq2 (Love et al, 2014) with default settings, and size factors were estimated on the total mRNA reads for global normalization. Transcripts were sub-grouped according to adjusted $p$ values from the differential expression analysis, with the group "Most down" having adjusted $p$ values below $10^{-5}$ and the group

"Less down" having adjusted $p$ values between $10^{-5}$ and 0.05. Volcano plots were generated based on log2 fold differences and adjusted $p$ values from DeSeq2 and visualized using EaSeq and the "Scatter"-tool or Microsoft Excel 2016. The number of selected gene subsets found within the significantly regulated genes was counted and compared to that expected by chance using Chi-square testing, and p-values for all shown comparisons were Bonferroni-adjusted before being plotted in bubble diagrams. Cut&Run values for 1D heatmaps of signal at TSSs were quantified using the "Quantify"-tool and default settings except for using offsets of ±1 kbp and visualized together with "basemean" and log2 fold difference values from the DeSeq2 output of Nascent transcripts using the "ParMap"-tool. The order of the TSSs was determined based on first grouping as mentioned above, and then the average expression value in all conditions (Basemean). GO-term enrichment analysis of significantly regulated transcripts was done using g:Profiler (https://biit.cs.ut.ee/gprofiler/, (Raudvere et al, 2019)). Beeswarm plots were made using R (https://www.R-project.org/) and the beeswarm package (The Bee Swarm Plot, an Alternative to Stripchart, version 0.2.0, A Eklund (2016), CRAN). Integration of expression data with cell-cycle-related transcriptional changes was done using published results (Bostrom et al, 2017). The "polar coordinates" from transcripts that were found to be significant in their work was used as the X-axis when visualizing the moving average log2-fold difference in the expression of 100 transcripts (Y-axis).

## Flow cytometry

Cell cycle changes were analyzed using the Click-iT EdU Alexa Fluor 647 Flow Cytometry Assay Kit (Invitrogen, CA, USA). Cells were labeled with EdU for 3 h after drug treatment with THZ531. Dead cells were marked using LIVE/DEAD Near-IR (Life Technologies, L10119), and Hoechst was used to mark DNA. The TUNEL assay (apoptosis) (Roche Diagnostics, Switzerland) and staining for γH2AX (DNA damage) and pS10H3 (mitosis) were performed as described previously (Dale Rein et al, 2015). Flow cytometry was performed using BD LSR II SORP (BD Biosciences, CA, USA). Analyses were done using the FlowJo software (FlowJo, LLC, OR, USA).

## Quantitative image-based microscopy (QIBC)

QIBC was performed as previously described (Somyajit et al, 2017; Toledo et al, 2013). Briefly, images were acquired with a ScanR inverted microscope high-content screening station (Evident IX83 inverted microscope) equipped with widefield optics, a 20x, 0.75-NA (UPLSAPO 20x) air objective, fast excitation and emission filter-wheel devices for DAPI, FITC, Cy3, and Cy5 wavelengths, an MT20 illumination system, and a digital monochrome Hamamatsu ORCA-R2 CCD camera (yielding a spatial resolution of 320 nm per pixel at 203 and binning of 1). Images were acquired in an automated fashion with the ScanR acquisition software (Olympus, 3.4). Depending on cell confluency, 100 images were acquired, containing more than 5000 cells per condition. Acquisition times for the different channels were adjusted for nonsaturated conditions in a 12-bit dynamic range, and identical settings were applied to all the samples within one experiment. Images were processed and analyzed with ScanR analysis software. First, a dynamic background correction was applied to all

images. The DAPI signal was then used for the generation of an intensity-threshold-based mask to identify individual nuclei as main objects. This mask was then applied to analyze pixel intensities in different channels for each individual nucleus. After the segmentation of nuclei, foci were segmented as above, and the desired parameters for the different nuclei or foci were quantified, with single parameters (mean and total intensities, foci count, and foci intensities) as well as calculated parameters (sum of foci intensity per nucleus). These values were then exported and analyzed with TIBCO Software, version 12.4. This software was used to quantify absolute, median, and average values in cell populations and to generate all color-coded scatter plots. Within one experiment, similar cell numbers were compared for the different conditions (at least 5000–10,000 cells), and for visualization, low x-axis jittering was applied (random displacement of objects along the x-axis) to make overlapping markers visible.

## DNA fiber assay

DNA fiber assay was done as described (Somyajit et al, 2017; Somyajit et al, 2021) with some modifications. Cells (100,000) were pulse-labeled with 25 mM CldU for the indicated time (please see figure panels for respective labeling protocols), washed three times with DMEM medium, and pulse-labeled with 250 μM IdU with or without the indicated treatment for the indicated time. Labeled cells were harvested on ice-cold PBS and mixed with unlabeled cells (1:3). Subsequently, 4 μL of the cell suspension was placed on SuperFrostTM slides and mixed with 8 mL of lysis buffer (0.5% SDS, 200 mM Tris, pH 7.5, and 50 mM EDTA) followed by vigorous pipetting for in situ lysis. After 2 min of incubation, slides were tilted to allow the lysate to flow along the slide slowly until the end of the slide. Next, slides were fixed in methanol: acetic acid (3:1) for 12–15 min, washed four times in PBS and transferred to 2.5 M HCl for DNA denaturation for 80 min. Afterward, slides were neutralized by washing four times in PBS and blocked in a blocking buffer (1x PBS, 0.1% Triton X, and 1% BSA) for 5 min. CldU was stained by incubating slides with rat anti-BrdU antibody (Abcam ab6326, 1:100 in blocking buffer) for 90 min. Afterward, slides were washed once with PBS containing 0.1% Tween-20, followed by three wash steps with PBS, fixed with 4% formaldehyde, and incubated with Alexa Fluor 594-conjugated goat anti-rat IgG (1:100; Thermo Fisher Scientific; A-11077) for 60 min. Slides were washed four times with PBS, and IdU was stained using mouse anti-BrdU antibody (1:100; Becton Dickinson, 347580) over-night at 4 °C, followed by Alexa Fluor 488–conjugated goat anti-mouse IgG (1:100; Thermo Fisher Scientific; A11029) for 90 min. Fibers were acquired using an Olympus BX53 Upright Fluorescence Microscope with a 40x air objective. For the quantification of replication structures, at least 200 DNA fibers were counted per experiment. The lengths of red (CldU) or green (IdU) labeled patches were measured using Fiji ImageJ software (National Institutes of Health). Fork speed in kb/min was calculated by multiplying the measured length in μm by a conversion factor of 2.59 kb/μm and dividing by the duration of the labeling pulse.

## Statistical analysis

GraphPad Prism 5.0 and 10.0 Software (GraphPad Software Inc., La Jolla, CA, USA) was used for statistical analysis. Statistical analysis of the result was performed by the Mann–Whitney test and $p$ values are from two-sided $t$-tests corrected for multiple testing with the Benjamini–Hochberg method. We have used one-way ANOVA with Kruskal–Wallis test, Dunn's and Dunnett multiple comparison test. For multiple comparisons, a two-way ANOVA followed by Tukey's multiple comparisons test was used. Data represent mean ± SD. Statistical significance levels were defined as: * $p < 0.05$, ** $p < 0.01$, *** $p < 0.001$, **** $p < 0.0001$.

### The paper explained

**Problem**

Glioblastoma (GBM) is a highly prevalent type of malignant brain tumors that are aggressive and rapidly fatal, with a median overall survival of around one year. The overall survival is only extended for a few months by the standard treatment of surgical resection, radio-therapy and chemotherapy with DNA-alkylating agents. Improving glioblastoma treatment will benefit from identifying and targeting additional biological processes in the disease.

**Results**

Here, we show that the hypertranscriptional state of GBMs is specifically sensitive to the inhibition of the key regulators of transcriptional elongation CDK12/CDK13.

Our work demonstrates that CDK12/CDK13 inhibition blocks glioblastoma stem cell (GSC) proliferation, impairs glioblastoma organoid survival, reduces GSC migration and invasion capacity, and suppresses tumor growth in xenograft models. We unveil that blocking CDK12/CDK13 impairs global mRNA synthesis and GSC-specific transcriptional programs, and induces DNA replication arrest prior to DNA-damage and cell-cycle checkpoint activation.

**Impact**

We expose a critical vulnerability in glioblastoma, where inhibiting CDK12/CDK13 disrupts the hyperactive transcriptional machinery in GBMs and the GBM-specific transcriptional programs. Our work presents CDK12/CDK13 inhibition as a promising strategy to target transcriptional addictions driving this aggressive disease.

## Data availability

The datasets produced in this study are available in the following databases: Cut&Run and SLAM-seq: NCBI's Gene Expression Omnibus (Edgar et al, 2002), GSE186311 (https://www.ncbi.nlm.nih.gov/geo/query/acc.cgi?acc=GSE186311).

The source data of this paper are collected in the following database record: biostudies:S-SCDT-10_1038-S44321-026-00393-w.

## Peer review information

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

## Acknowledgements

We would like to thank the Advanced Light Microscopy (ALM) and Flow Cytometry Core Facility at Oslo University Hospital for help with microscopy and flow cytometry experiments. DPP was supported by funding from the Health Southeast Region Agency, Norway (HSØ 2018045, HSØ 2021037, HSØ2024022, and HSØ2025082), the Research Council of Norway through its Centres of Excellence scheme, project number 332713, CRESCO, Nansenfondet, and SPARK program from the University of Oslo. SL was supported by a scholarship from the Medical Student Research Program, University of Oslo, Norway. ML was supported by the Danish National Research Foundation (DNRF115). AL, AG, and SPN were supported by the National Research Fund (FNR) of Luxembourg (C20/BM/14646004/GLASS-LUX, INTER/TRANSCAN22/17612718/PLASTIG). BB and KBMF were supported by the European Union (MARKOPOLO consortium, GA No. 101156161). Views and opinions expressed are those of the author(s) only and do not necessarily reflect those of the European Union or the European Health and Digital Executive Agency (HADEA). Neither the European Union nor the granting authorities can be held responsible. MN and JER were supported by the University of Oslo, Nansenfondet and Unifor Frimed.

## Author contributions

**Silje Lier**: Data curation; Formal analysis; Investigation; Methodology; Writing—review and editing; Performed main glioblastoma stem cell (GSC) functional assays; generated CUT&RUN and SLAM-seq data; conducted migration assays; carried out flow cytometry and CRISPR/Cas9 competition assays; contributed to data analysis and manuscript preparation. **Sara B Markusson**: Data curation; Formal analysis; Writing—original draft; Project administration; Writing—review and editing; Contributed to GSC functional assays; co-wrote the manuscript. **Anja Kocijancic**: Data curation; Investigation; Methodology; Writing—review and editing; Contributed to GSC functional assays; performed QIBC analyses and DNA fiber assays. **Martine Narum**: Investigation; Methodology; Writing—review and editing; Conducted migration assays on organotypic slice cultures. **Solveig O Lund**: Data curation; Investigation; Methodology; Writing—review and editing; Contributed to GSC functional assays. **Bianka Böllering**: Data curation; Formal analysis; Investigation; Methodology; Writing—review and editing; Performed QuPath analyses for IHC quantification. **Anuja Lipsa**: Data curation; Investigation; Methodology; Writing—review and editing; Generated patient-derived organoids; performed organoid viability assays. **Mirra L C Søegaard**: Data curation; Formal analysis; Investigation; Methodology; Writing—review and editing; Assisted with data analyses. **Idun D Rein**: Data curation; Formal analysis; Investigation; Methodology; Writing—review and editing; Performed flow cytometry and CRISPR/Cas9 competition assays. **Petra Santha**: Data curation; Formal analysis; Investigation; Methodology; Writing—review and editing; Contributed to GSC functional assays and performed statistical analyses. **Preeti Jain**: Data curation; Investigation; Methodology; Writing—review and editing; Contributed to GSC functional assays. **Anna Lång**: Data curation; Investigation; Methodology; Writing—review and editing; Performed migration assays. **Emma Lång**: Data curation; Investigation; Methodology; Writing—review and editing; Performed migration assays. **Niklas Meyer**: Data curation; Investigation; Methodology; Writing—review and editing; Set up organotypic slice cultures; contributed critical input. **Aparajita Dutta**: Formal analysis; Assisted with data analyses. **Santosh Anand**: Data curation; Investigation; Methodology; Writing—review and editing; Assisted with data analyses. **Sugith B Badugu**: Formal analysis; Performed QIBC analyses and DNA fiber assays. **Gaute J Nesse**: Data curation; Formal analysis; Supervision; Investigation; Methodology; Writing—review and editing; Performed cloning and protein purification. **Rune J Forstrøm**: Data curation; Formal analysis; Supervision; Investigation; Methodology; Writing—review and editing; Performed cloning and protein purification. **Arne Klungland**: Formal analysis; Supervision; Funding acquisition; Investigation; Methodology; Writing—review and editing; Provided critical input on study design and manuscript. **Ashish Anand**: Resources; Formal analysis; Supervision; Investigation; Methodology; Writing—review and editing; Assisted with data analyses. **Steven M Pollard**: Resources; Data curation; Formal analysis; Supervision; Investigation; Methodology; Writing—review and editing; Provided glioblastoma stem cell models. **Stig O Bøe**: Data curation; Formal analysis; Supervision; Writing—review and editing; Contributed to experimental design in the migration assays. **Johanne E Rinholm**: Data curation; Formal analysis; Supervision; Investigation; Methodology; Writing—review and editing; Contributed to experimental design; supervised organotypic slice experiments; provided critical input. **Katrin B M Frauenknecht**: Data curation; Formal analysis; Supervision; Funding acquisition; Investigation; Methodology; Writing—original draft; Writing—review and editing; Performed immunohistochemistry; contributed to experimental design and manuscript writing; supervised QuPath analyses. **Anna Golebiewska**: Conceptualization; Resources; Data curation; Formal analysis; Supervision; Funding acquisition; Investigation; Methodology; Writing—original draft; Project administration; Writing—review and editing; Contributed to experimental design; generated patient-derived organoids; contributed to manuscript writing. **Simone P Niclou**: Data curation; Supervision; Writing—review and editing; Helped with patient-derived organoids. **Kumar Somyajit**: Data curation; Formal analysis; Supervision; Investigation; Methodology; Writing—review and editing; Contributed to experimental design; supervised QIBC and DNA fiber assays; contributed to manuscript writing. **Mads Lerdrup**: Data curation; Formal analysis; Supervision; Funding acquisition; Investigation; Methodology; Writing—original draft; Writing—review and editing; Contributed to experimental design; led data analyses; co-wrote the manuscript. **Deo P Pandey**: Conceptualization; Resources; Data curation; Formal analysis; Supervision; Funding acquisition; Investigation; Methodology; Writing—original draft; Project administration; Writing—review and editing; Conceived the study; designed experiments; supervised all experimental work; performed migration assays and flow cytometry/CRISPR assays; contributed to data analysis; co-wrote the manuscript.

Source data underlying figure panels in this paper may have individual authorship assigned. Where available, figure panel/source data authorship is listed in the following database record: biostudies:S-SCDT-10_1038-S44321-026-00393-w.

## Disclosure and competing interests statement

The authors declare no competing interests.

# Expanded View Figures

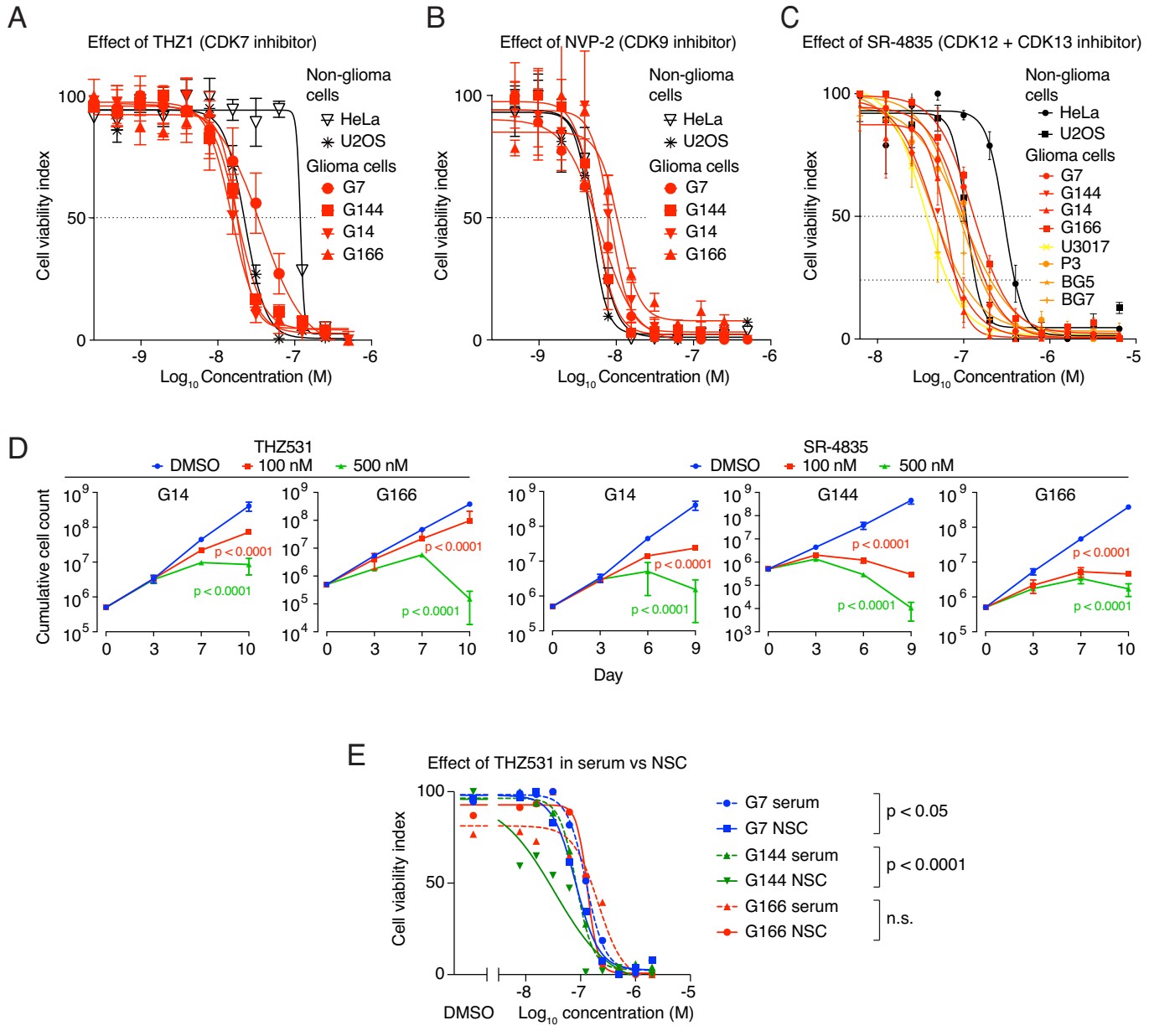

**Figure EV1.  Effect of tCDK inhibition on GSC viability and proliferation.**

(A) Four high grade glioblastoma cell lines and two non-glioblastoma cell lines were treated with increasing doses of THZ1. After 72 h, cells were subjected to an MTT assay. The graph displays a dose-response curve with percent cell viability relative to the DMSO control for each cell line. Data represent mean ± SD of three replicates. (B) Four high-grade glioblastoma cell lines and two non-glioblastoma cell lines were treated with increasing doses of NVP-2. After 72 h, cells were subjected to an MTT assay. The graph displays a dose-response curve with percent cell viability relative to the DMSO control for each cell line. Data represent mean ± SD of three replicates. (C) Eight high-grade glioblastoma cell lines and two non-glioblastoma cell lines were treated with increasing doses of SR-4835. After 72 h, cells were subjected to an MTT assay. The graph displays a dose-response curve with percent cell viability relative to the DMSO control for each cell line. Data represent mean ± SD of three replicates. (D) In vitro cell proliferation assay of GSCs treated as indicated. Data represent mean ± SD of two replicates. Data were analyzed by two-way ANOVA followed by Tukey's multiple comparisons test, with significant differences as compared to the control. Exact p-values are provided in Appendix Table S4. (E) Three high-grade glioblastoma cell lines were cultured in serum-free or serum-containing media (with serum added immediately before the viability assay) and were treated with increasing doses of THZ531. After 72 h, the cell viability was assessed using CellTiter-Glo. The graph displays a dose-response curve with percent cell viability relative to the DMSO control for each cell line. Data represent mean ± SD of three replicates. Data were analyzed using two-way ANOVA followed by Tukey's multiple comparisons test. G7 serum vs. G7 NSC: $p = 0.010252731$; G144 serum vs. G144 NSC: $p = 9.56E-13$.

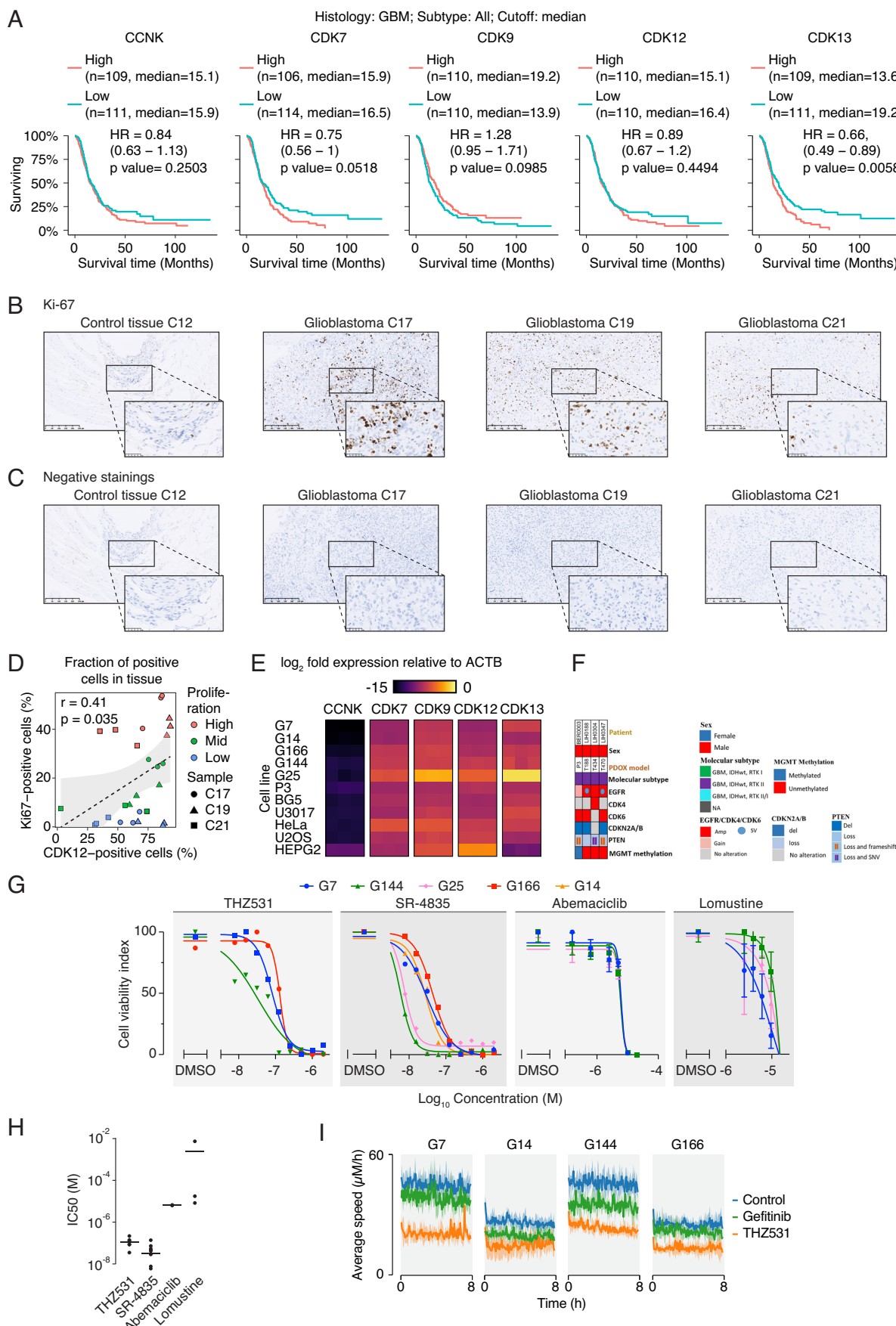

◀ **Figure EV2.   tCDK expression correlates with glioblastoma patient survival, proliferation, and sensitivity to CDK12/CDK13 inhibition.**

(A) Kaplan–Meier survival analyses generated using GlioVis showing the association between overall survival of glioblastoma patients and expression levels of tCDKs. Patients were stratified into high and low expression groups based on median gene expression. Statistical significance was assessed using the log-rank test. (B) Representative images of Ki-67 immunohistochemistry in cortex/infiltration zone/cell-rich tumor of glioblastoma patients. (C) Negative stainings for the patient samples used in the study. (D) Correlation analysis of CDK12-positive cells and proliferation status in patient tissue is shown in Figs. 2A and EV2B. One-way ANOVA was used to compare CDK12 expression across low, intermediate, and high proliferation categories. Correlation calculated with Spearman correlation, $r = 0.41$; $p = 0.035$. (E) Heatmap showing the mRNA expression of tCDKs in the cell lines used in the study. (F) Summary of the patient characteristics of the GBM organoid models. (G) Four high-grade glioblastoma cell lines were treated with increasing doses of inhibitors as indicated. After 72 h, the cell viability was assessed using CellTiter Glo. The graph displays a dose-response curve with percent cell viability relative to the DMSO control for each cell line. Data represent mean ± SD of three replicates. (H) Dot-plot showing IC50 values for dose response of inhibitors on a panel of GSCs shown in (G). (I) Effect of THZ531 treatment on the migration of glioblastoma cells. The average speed of migration is plotted over time. Gefitinib, an EGFR inhibitor is used a positive control.

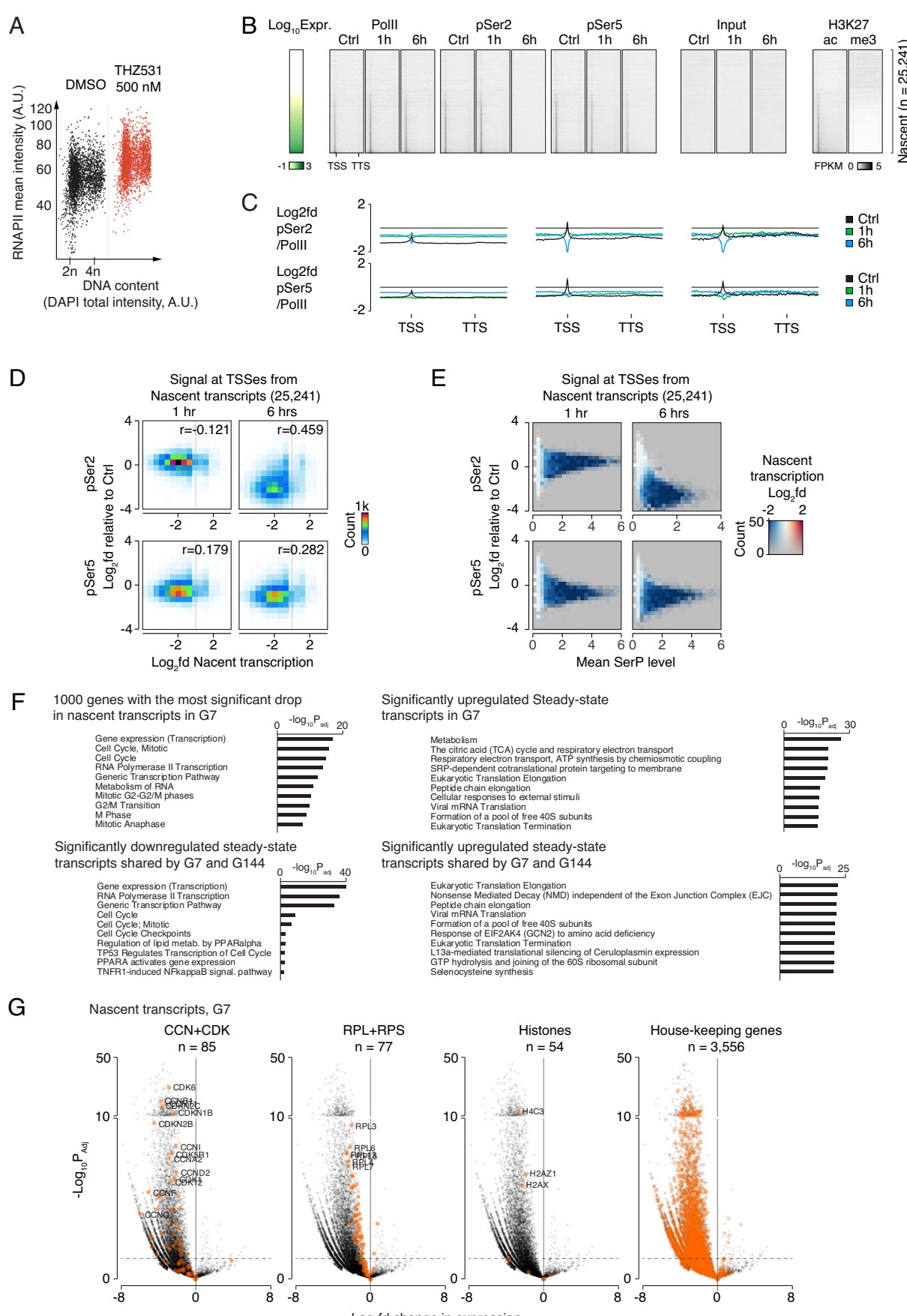

◀  **Figure EV3.   Effect of CDK12/CDK13 inhibition on genome wide RNAPII phosphorylation and mRNA synthesis in GSCs.**

(A) QIBC analysis of pSer2 for G7 cells treated with vehicle or 500 nM THZ531 for 6 h. (B) Heatmaps of Cut&Run signal from RNAPII, RNAPII phosphorylation states, selected histone mark modifications, as well as input at gene bodies and immediate upstream and downstream regions (±25% of gene length) from G7 cells treated with THZ531 for either 1 or 6 h, and DMSO controls. Genes were ordered vertically based on their total expression level. The horizontal extent of each gene and the upstream and downstream regions corresponding to a quarter of the gene length is fitted within the same visual space in the heatmaps, regardless of its absolute extent. TSS and TTS illustrate transcription start sites and termination sites, respectively. Cut&Run and input levels are FPKM normalized. (C) Graphs of average Cut&Run signal from RNAPII phosphorylation states normalized to RNAPII levels at all gene bodies and surrounding loci. RNAPII and RNAPII modification states were obtained as described in Fig. 4A. The horizontal extent of each gene and the upstream and downstream regions corresponding to half a gene length is fitted within the same visual space in the heatmaps regardless of its absolute extent. TSS and TTS illustrate transcription start sites and termination sites, respectively. Cut&Run levels are FPKM normalized. (D) 2D-histograms showing the relationship between changes in nascent transcription (X-axis) and pSer2 (top) or pSer5 (bottom) at TSSs after 1 h (left) and 6 h (right). (E) Colored MA-plots showing the combined relationship between changes pSer2 (top) or pSer5 (bottom) at TSSs after 1 h (left) and 6 h (right) colored according to changes in nascent transcript levels. (F) Bar diagrams showing the most significantly enriched gene ontology (GO) terms from selected subsets of genes from steady-state and nascent RNA. X-axes represent −log10 *p* values from Fisher's one-tailed tests adjusted for multiple testing using g:Profiler's default g:SCS algorithm. (G) Volcano plots showing the overall transcriptional differences in nascent transcripts, as in Fig. 4L, but with certain gene populations highlighted. X-axes show the log2 fold difference in transcription in G7 cells treated with THZ531 for 6 h compared to DMSO controls. Y-axes show the −log10 transformed *p* values from two-tailed Wald tests Benjamini–Hochberg corrected for multiple testing. Colored dots illustrate the transcriptional changes of the listed gene populations.

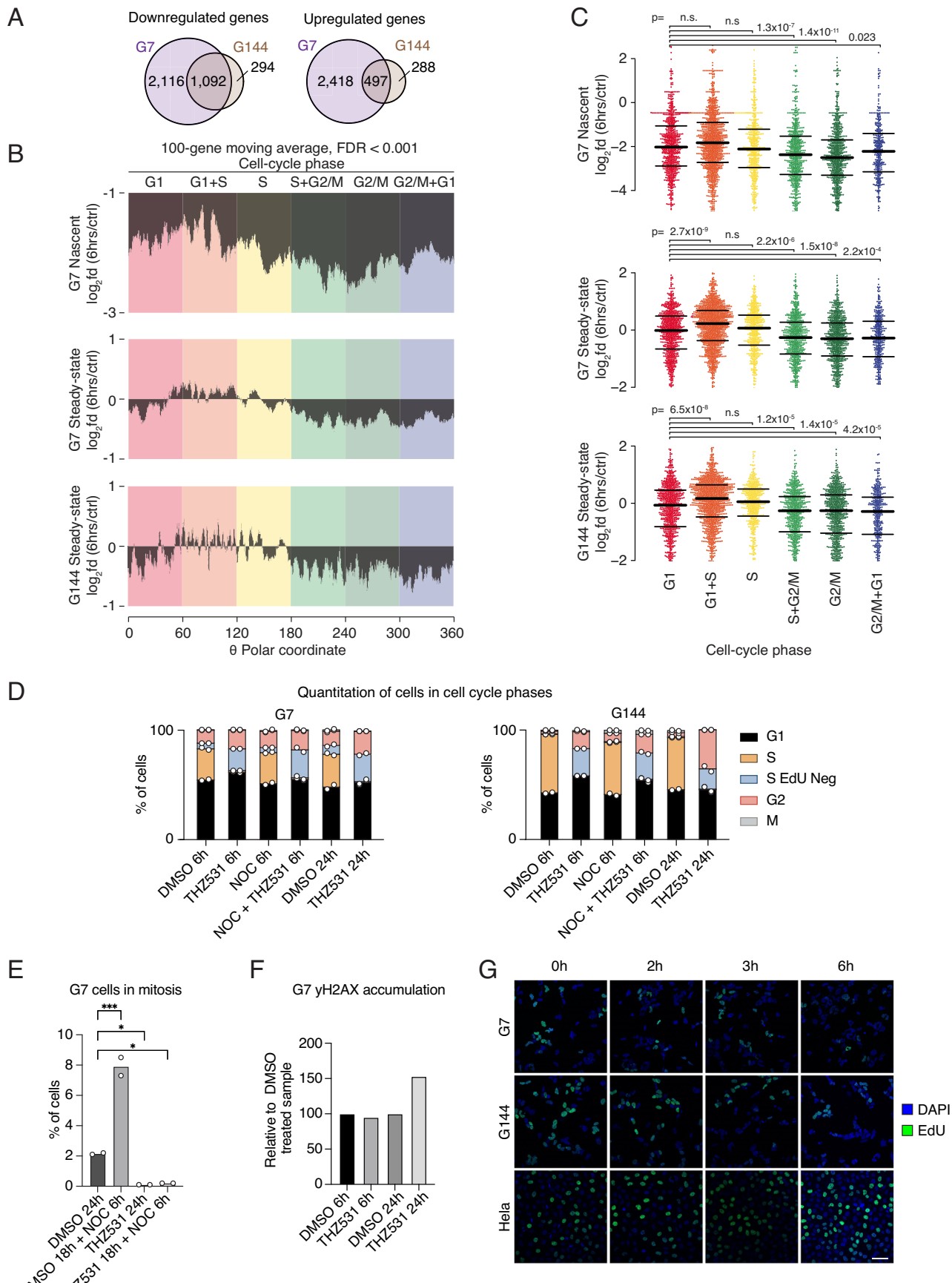

◀ **Figure EV4.  Effect of CDK12/CDK13 inhibition on GSC cell cycle.**

(**A**) Venn diagrams illustrating the overlap in the populations of genes being downregulated (left) as well as upregulated (right) in G7 cells compared to G144 cells. (**B**) Graphs illustrates the moving average of transcriptional changes in nascent (top) or steady-state (middle, bottom) transcripts from G7 (top, middle) or G144 (bottom) cells treated with THZ531 for 6 h compared to DMSO controls. Transcripts were ordered according to the previously published classification of cell-cycle timing. Only transcripts, for which transcriptional timing could be assessed with an FDR-value of 0.001 or better, were included. (**C**) Beeswarm plots of transcriptional changes as in (**B**), with transcripts grouped into six overall groups based on previously published transcriptional timing. *P* values were obtained using Mann–Whitney *U*-tests and Bonferroni-corrected for multiple testing. (**D**) Bar diagrams of cell cycle distributions of G7 and G144 cells treated as indicated. Doses used: THZ531 (500 nM), Nocodazole (1 μg/mL). (**E**) Flow cytometry analysis of % of G7 cells in mitosis following 500 nM THZ531 ± 1 μg/mL Nocodazole treatment for 24 h. Data were analyzed by one-way ANOVA followed by Dunnett's post hoc test *$p < 0.05$, ***$p < 0.001$. Significant differences as compared to the control: DMSO 18 h + NOC 6 h: $p = 0.0004$; THZ531 24 h: $p = 0.0194$; THZ531 18 h + NOC 6 h: $p = 0.0231$. Data represent mean ± SD of two replicates for (**D**, **E**). (**F**) Flow cytometric assays of γH2AX accumulation in G7 cells treated with 500 nM THZ531 for 6 and 24 h. (**G**) Cells were treated with 500 nM THZ531 at indicated times, and EdU incorporation was performed in the last 1 h by adding 10 μM EdU. Cell staining was done using Click-IT chemistry according to the manufacturer's instructions. Scale bar: ~50 μm (approximated based on a comparable image acquired using the same imaging setup).

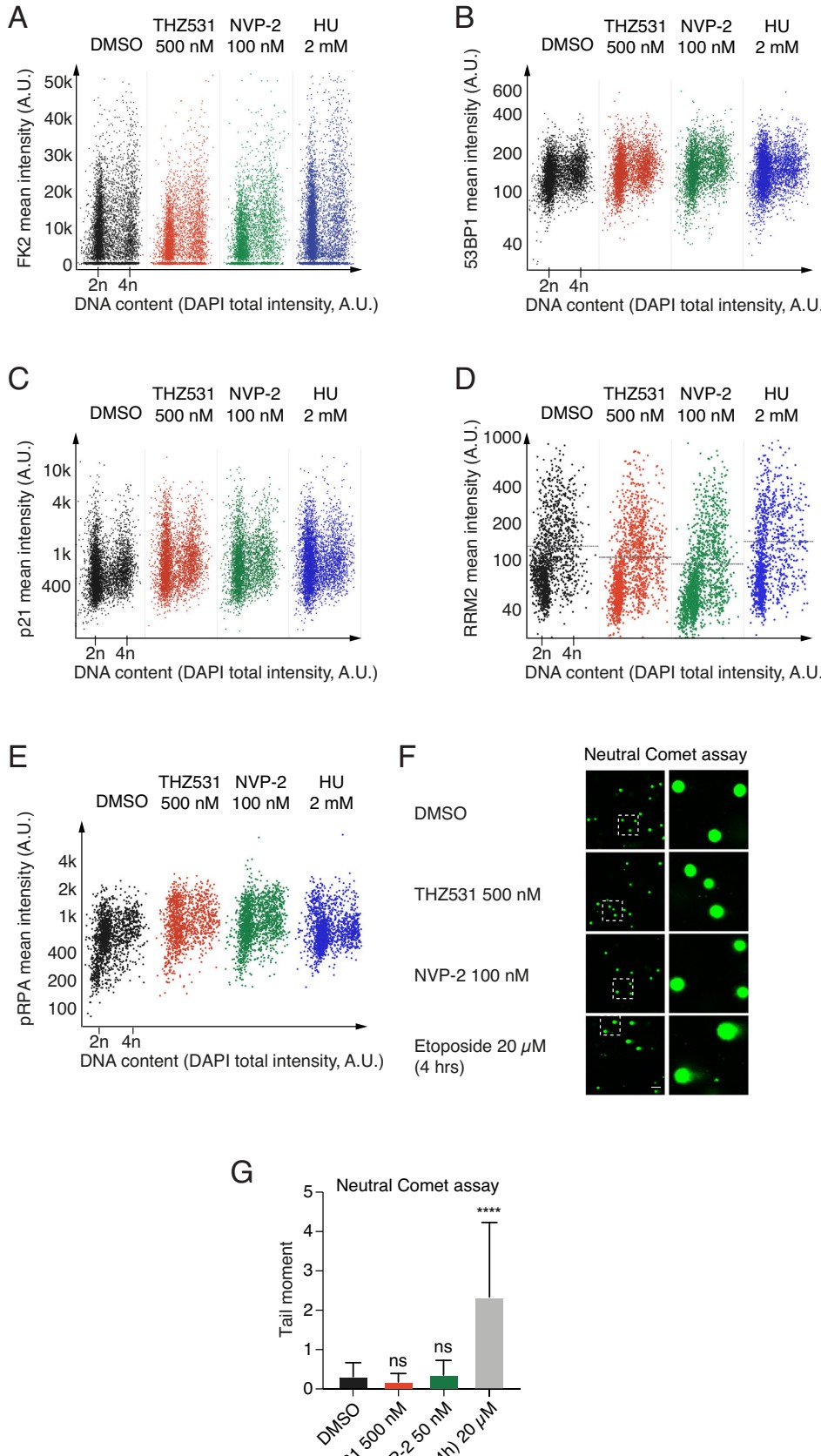

◀ **Figure EV5. Effect of CDK12/CDK13 inhibition on markers of replication stress and DDR in GSCs.**

(A–E) QIBC analyses of (A) FK2, (B) 53BP1, (C) p21, (D) RRM2, and (E) pRPA expression (Y-axis) relative to DNA content (X-axis) in G7 cells under vehicle control, 500 nM THZ531 treatment, and control treatments with 100 nM NVP-2 and 2 mM HU. (F) Neutral comet assays showing tail movements depicting DNA damage in G7 cells for vehicle control, 500 nM THZ531 treatment, and control treatments with 50 nM NVP-2 and 20 μM Etoposide. Scale bar: 100 μm. (G) Quantification of the tail moments. Approximately 100 cells were measured per sample. Data were presented as mean ± SD of duplicates from three biological replicates. Statistical analysis by one-way ANOVA with Kruskal–Wallis test and Dunn's multiple comparison correction, significant differences as compared to the control: Etoposide (4 h) 20 μM: <0.000000000000001. *$p < 0.05$, **$p < 0.01$, ***$p < 0.001$. ****$p < 0.0001$.

