## [Peer Review File · EMBO Molecular Medicine]

CDK12/CDK13 inhibition disrupts transcriptional elongation and fork progression in glioblastoma

Silje Lier, Sara Markusson, Anja Kocijancic, Martine Narum, Solveig Lund, Bianka Böllering, Anuja Lipsa, Mirra Søegaard, Idun Rein, Petra Santha, Preeti Jain, Anna Lång, Emma Lång, Niklas Meyer, Aparajita Dutta, Santosh Anand, Sugith Badugu, Gaute Nesse, Rune Forstrøm, Arne Klunghand, Ashish Anand, Steven Pollard, Stig Ove Bøe, Johanne Rinholm, Katrin Frauenknecht, Anna Golebiewska, Simone Niclou, Kumar Somyajit, Mads Lerdrup, and Deo Prakash Pandey

Corresponding authors: Deo Prakash Pandey (deopan@ous-hf.no), Mads Lerdrup (mads.lerdrup@bric.ku.dk)

Review Timeline:

Transferred from Review Commons:	21st Aug 25
Editorial Decision:	16th Sep 25
Revision Received:	12th Dec 25
Editorial Decision:	15th Jan 26
Revision Received:	29th Jan 26
Accepted:	9th Feb 26

Editor: Zeljko Durdevic

Transaction Report:

Review
COMMONS

This manuscript was transferred to EMBO Molecular Medicine following peer review at Review Commons.

Review #1**1. Evidence, reproducibility and clarity:****Evidence, reproducibility and clarity (Required)**

The authors in this manuscript studied the role of a transcriptional cyclin-dependent kinase CDK12/CDK13 in glioblastoma. These cyclin-dependent kinases phosphorylate at ser2 residue in the C-terminal of RNA Pol II. Pharmacological inhibition of CDK12/CDK13 kinase with inhibitor decreases cell proliferation in multiple glioma cell lines and in patient-derived organoids. The CDK12/CDK13 inhibitor also reduces tumor growth in a mouse xenograft model. Mechanistically, the authors showed that genome-wide inhibition of CDK12/CDK13 attenuates RNA Pol II phosphorylation, disrupting transcriptional elongation and decreasing cell cycle progression. So, the authors proposed that targeting CDK12/CDK13 kinases can be used as a therapeutic strategy in glioblastoma. The authors have done extensive work in this manuscript to understand the role of CDK12/CDK13 in glioblastoma, but it is still a descriptive paper lacking mechanistic details.

****Comments:****

1. Figure 1 shows that CDK12/CDK13 inhibitor decreases cell viability, colony-forming ability, cell competition assay, and cell migration. The rationale behind choosing CDK12/CDK13 inhibitor in glioma is unclear from the manuscript. What is the CDK12/CDK13 expression in multiple glioma cells vs non-glioma cells? The authors should include normal astrocytes as a control for cell viability assay. The p value is missing in numerous Figure panels.
2. Figure 2A shows the expression of CDK12 by immunohistochemistry in glioblastoma tissues. Including the non-glioma tissue samples as another control and including a quantification graph with the statistics is essential. In Figure 2B-D, the authors discussed the treatment of glioma patient-derived organoids with CDK12/CDK13 inhibitors. From the Figure, the organoids are resistant to THZ531 and SR-4835 inhibitors. To rule out this possibility, the immunoblot assay with cleave PARP will be essential to execute. Again, statistics need to be included in Figure 2C-D.
3. The mouse subcutaneous xenograft experiment was carried out in U87 cells with CDK12/CDK13 inhibitors. However, the glioma stem cells are a more appropriate model for glioma biology, and it is not clear why authors suddenly chose U87 cells. Again, statistics are absent in multiple sub-panels.
4. The authors have performed CUT & RUN experiments in G7 cells with CDK12/CDK13

inhibitors and decided to use 1hr and 6hr time points for the assay. Although the inhibitor THZ531 is supposed to inhibit RNA Pol II phosphorylation at the Ser2 residue, it decreases the Pol II phosphorylation at the ser5 residue quite a bit. Therefore, it is crucial to determine the effect coming from ser2 vs ser5 phosphorylation and gene expression regulation.

5. There are a lot of supplementary Figures where axes are not labeled correctly or missing.

6. The statistical section needs to be included in the manuscript.

2. Significance:

Significance (Required)

In this manuscript, the authors studied the role of CDK12/CDK13 in glioblastoma and performed extensive studies to uncover the importance of these kinases in glioblastoma. Understanding more mechanistic details of how these kinases are involved in glioma progression will uncover more therapeutic opportunities in glioblastoma.

3. How much time do you estimate the authors will need to complete the suggested revisions:

Estimated time to Complete Revisions (Required)

(Decision Recommendation)

Between 3 and 6 months

4. Review Commons values the work of reviewers and encourages them to get credit for their work. Select 'Yes' below to register your reviewing activity at Web of Science Reviewer Recognition Service (formerly Publons); note that the content of your review will not be visible on Web of Science.

Yes

Review #2

1. Evidence, reproducibility and clarity:

Evidence, reproducibility and clarity (Required)

****Summary:****

Lier et al. present a set of results showing that pharmacological inhibition of CDK12/13,

cyclin-dependent kinases that phosphorylate RNA polymerase II (RNAPII), alters the proliferative behavior and transcriptional program of glioblastoma cells. A set of 2D and 3D cultures of patient-derived cell lines with stem-like properties (GSC), as well as subcutaneous xenografts of the U87 cell line, were used as in vitro and in vivo models, respectively. Among the CDKs tested, only CDK13 expression was found to be associated with worse patient survival, while CDK12-immunoreactive cells were detected in patient glioblastoma tissues. The response of GSCs to the CDK12 and CDK13 inhibitor TZH541 included cell cycle blockade and decreased migration. Reduction in RNAPII phosphorylation in TZH541-treated cells was verified using one of the GSC lines. Genome-wide exploration of the transcriptional consequences of TZH541 treatment of 2 GSCs using CUT&RUN and SLAM-seq technologies revealed major transcriptional repression, particularly of genes associated with cell proliferation.

****Main comments:****

Although I found this study very interesting, I noted points requiring clarification, particularly in order to fully support the authors conclusions. My recommendations focus on the glioblastoma cell biology experiments, my area of expertise.

- The rationale for studying only CDK12 expression in patient glioblastoma tissues needs clarification. In contrast with CDK13, the authors found no association between CDK12 expression levels and patient survival (Sup Fig. 1A). Do the authors obtain similar results using independent datasets of glioblastoma tissue transcriptomes (e.g. CGGA)? With regard to the major effect of CDK12/13 inhibition on glioblastoma cell proliferation, determining whether CDK12/13 expression is observed in proliferating areas of the patients' tumor tissues (Ki67 IHC) would help support the authors' conclusion that their "results provide proof-of-concept for the potential of CDK12 and CDK13 as therapeutic targets for glioblastoma". The main data regarding CDK expression the status in patients' tumors and their possible association with patient survival should be rearranged in the same figure and described in the same paragraph of the results.

- Fig.1 caption "Inhibition of CDK12/13 specifically affects proliferation of glioma cells" is not entirely consistent with the results. This inhibition also appears to induce cell death, at least in some of the GSC tested, as indicated with cell counts (Fig. 1C., sup Fig.1 G) and an 8-fold increase in the % of apoptotic cells after a 24h-TZH treatment shown in Fig. 5E. All data concerning the effects of TZH on proliferation and survival (including detailed effects on the cell cycle) should be brought together rather than split between the 1st and last figure.

- The reason for which serum-treated GSC were used should be explicated (sup Fig. 1C).

Serum being usually used to trigger GSC "differentiation", did the authors want to verify whether CDK12/13 inhibitors affected GSC in a specific manner? If yes, it is necessary to demonstrate that serum-treated GSC have lost their stem-like properties.

- The viability of patient-derived 3D organoids (GBO) was assessed by measuring ATP production. It is therefore not possible to distinguish between decreased cell proliferation and increased cell death as responsible for the signal decrease. This limitation in the interpretation of the results needs to be made explicit. I was also misled by the use of GBO. This abbreviation is currently used to designate fragments of patient tumor tissue amplified in culture, which retain the cellular heterogeneity and the extracellular matrix of the original tumor and therefore provide an actual ex vivo model of the tumor. To avoid any misunderstanding, I recommend referring to experimental models obtained from dissociated patient-derived cell lines as "3D organoids" or "cellular spheroids", and avoiding to designate them as ex vivo models since they do not recapitulate the complexity of the tumor.

- Although the abstract contains a statement indicating that CDK12/13 genetic ablation inhibits cell migration, I did not find the corresponding results in the article. The demonstration that CDK12/13 inhibition decreases cell migration is weaker than the demonstration of its effect on proliferation. Contrary to the experiments evaluating cell proliferation, cell migration was assessed using a single technical approach. Moreover, the method used to assay TZH effects on cell migration rather measures cell motility than cell migration over long distances in a 3D and complex environment as observed in diffuse glioma. Since these data add nothing significant to the article, I would delete them.

- In my opinion, the information from the in vivo experiments is limited and should be presented in a supplementary rather than a main figure. The data were obtained with a single cell model, U87 cells of uncertain origin, and using subcutaneous xenografts that provide an environment totally different from the patient's actual tumor. In this context, the data obtained provide little information on the response of cancer cells in a complex and specific environment well known to promote tumor growth and resistance to therapies. I understand that the use of intracerebral xenografts is not feasible, since the inhibitor does not appear to reach the brain. With this technical limitation, an alternative would be to deliver the compound directly inside the brain tumor. A cannula can be implanted into the tumor after it has formed, and connected to an Alzet minipump filled with the drug. These experiments are technically difficult, however, and success is not guaranteed. Another alternative would be to use GBO, as described by Jacob et al (2019) as a surrogate for tumor tissue, provided the authors can obtain tissue fragments from patient surgical resections or intracerebral xenografts of patient-derived cell lines. These alternatives are optional.

****Minor comments:****

- Fig. 4A and Fig. 5E-F: Results from a single experiment? If yes, they must be repeated at least once.
- For the sake of clarity, all y-axes in graphs presenting MTT or CellTiter-Glo assay results should be labeled "cell viability index", as they only provide a measure of overall cell or organoid metabolic activity, and thus an indirect assessment of cell viability.
- Statistical analyses are missing for 3 of the 4 cell lines presented in Figure 1F.
- Some GO terms are truncated in sup Fig. 3.
- The legend to Fig. 5B-D shows the mean and SD of 2 replicates. Please show individual points.
- Sup Fig1 D-F: unit of concentration is missing (M?)

2. Significance:

Significance (Required)

Despite growing interest in the roles of CDK12/13 roles in cancers and their targeting for cancer therapy, their involvement in glioblastoma growth remains unexplored. The results presented in this study outline the potential of CDK12/13 inhibition in controlling the growth of glioblastoma, at least in vitro, and thus provide meaningful information on its potential usefulness for this aggressive brain tumor with a high proliferation rate. Obtaining the full proof-of-concept that CDK12/13 constitute relevant targets for glioblastoma therapies will however require additional experiments demonstrating efficacy of CDK12/13 inhibition in complex environments, as encountered in the patients' tumor.

3. How much time do you estimate the authors will need to complete the suggested revisions:

Estimated time to Complete Revisions (Required)

(Decision Recommendation)

Between 1 and 3 months

4. Review Commons values the work of reviewers and encourages them to get credit for their work. Select 'Yes' below to register your reviewing activity at Web of Science Reviewer Recognition Service (formerly Publons); note that the content of your review will not be visible on Web of Science.

No

Full Revision

Manuscript number: RC-2023-02082

Corresponding author(s): Deo Prakash, Pandey & Mads, Lerdrup

1. General Statements [optional]

We were pleased to receive the reviewers' comments and grateful for their suggestions on how to improve our manuscript. We appreciate that both reviewers find the work extensive and meaningful.

In response to the reviewers' comments, we have performed a comprehensive series of additional experiments, which significantly enhance both the preclinical and mechanistic insights presented in our manuscript. As a result, we have added 26 new figure panels (main and supplementary), leading to the inclusion of one new figure and a major restructuring of two existing figures.

To summarize, Reviewer 1 acknowledges the extensive work we have undertaken to investigate the role of CDK12/CDK13 in glioblastoma and would like to see additional mechanistic details. Reviewer 2 recognizes the value of our work in exploring the potential usefulness of CDK12/CDK13 inhibition in the treatment of aggressive brain tumors and would like to see additional experiments, which demonstrate the efficacy of CDK12/CDK13 inhibition in complex environments to reinforce our proof-of-concept.

To address these, we have performed two overall categories of experiments strengthening the preclinical and mechanistic parts of our work. In summary, these are:

A) *New mechanistic insights into CDK12/CDK13 requirement in glioblastoma*

- We dissected the mechanism of CDK12/CDK13 inhibition and identified DNA replication arrest as an early effect compared to DNA damage and apoptosis.
- New QIBC (quantitative image-based cytometry) experiments revealing that CDK12/CDK13 inhibition induces a distinct mechanism causing severe suppression of replication fork progression without DNA damage or cell cycle checkpoint activation.
- Direct assessment of DNA damage using comet assays to demonstrate that early CDK12/CDK13 inhibition does not cause damage to DNA.
- New benchmarking data shows that CDK12/CDK13 inhibition leads to a distinct and more pronounced effect on replication compared to CDK9 inhibition (CDK9 also phosphorylates Ser2 of the RNA polymerase II C-terminal domain (RNAPII CTD)).
- We strengthened our key observations that CDK12/CDK13 inhibition abrogates pSer2 of the RNAPII CTD and nascent RNA synthesis using independent methodology.

B) *Further translational characterization of CDK12/CDK13 inhibition in glioblastoma*

- We expanded and strengthened the evaluation of CDK12 and CDK13 expression in glioblastoma patient tissues by performing immunohistochemistry (IHC) using newly

generated antibodies specific to CDK12 and CDK13. Our analysis revealed that both CDK12 and CDK13 exhibit nuclear expression in glioblastoma patient samples. Additionally, Ki-67 IHC demonstrated that the expression of CDK12 and CDK13 is independent of the proliferation status of the cells.

- We established a migration assay using glioblastoma stem cells (GSC) G7 in organotypic mice brain slices and examined the effect of CDK12/CDK13 inhibition on glioblastoma migration *ex vivo*. Our results showed that inhibiting CDK12/CDK13 significantly reduced glioblastoma migration, providing further support for the role of CDK12/CDK13 in glioblastoma progression, as demonstrated in our earlier findings.

These new data reinforce our primary findings that CDK12/CDK13 inhibitors exploit the transcriptional addiction of glioblastoma cells, thereby underscoring their potential as therapeutic targets for glioblastoma. Our work also offers mechanistic insights into how the glioblastoma stem cells have acquired transcriptional addiction to CDK12/CDK13 involving phosphorylation of RNAPII CTD, nascent RNA synthesis and DNA replication dependent on CDK12/CDK13 activity.

3. Color codes:

4. Reviewer comment

5. Authors' comments

6. Authors' actions

Reviewer #1 (Evidence, reproducibility and clarity (Required)):

The authors in this manuscript studied the role of a transcriptional cyclin-dependent kinase CDK12/CDK13 in glioblastoma. These cyclin-dependent kinases phosphorylate at ser2 residue in the C-terminal of RNA Pol II. Pharmacological inhibition of CDK12/CDK13 kinase with inhibitor decreases cell proliferation in multiple glioma cell lines and in patient-derived organoids. The CDK12/CDK13 inhibitor also reduces tumor growth in a mouse xenograft model. Mechanistically, the authors showed that genome-wide inhibition of CDK12/CDK13 attenuates RNA Pol II phosphorylation, disrupting transcriptional elongation and decreasing cell cycle progression. So, the authors proposed that targeting CDK12/CDK13 kinases can be used as a therapeutic strategy in glioblastoma. The authors have done extensive work in this manuscript to understand the role of CDK12/CDK13 in glioblastoma, but it is still a descriptive paper lacking mechanistic details.

Authors' comments:

We thank the reviewer for acknowledging the breadth of our work and the importance of strengthening the mechanistic insights. In response, we performed extensive additional experiments (see authors' actions below) revealing that CDK12/CDK13 inhibition suppresses

RNAPII phosphorylation and nascent transcription independently of cell cycle-regulated transcriptional programs and induces DNA replication arrest without triggering DNA damage or checkpoint activation. These findings define a clearer mechanistic role for CDK12/CDK13 in sustaining glioblastoma growth.

Authors' actions:

To address the reviewer's point, we have expanded the mechanistic characterization and validated our observations using multiple independent approaches:

RNAPII CTD pSer2 and nascent RNA synthesis: Using QIBC, we show that CDK12/CDK13 inhibition causes a rapid and potent reduction of RNAPII CTD pSer2 phosphorylation and suppression of nascent mRNA synthesis (revised manuscript Fig. 4B and 4D, Suppl. Fig. 3A). The results from these additional assays strengthen our previous findings acquired using immunoblotting, CUT&RUN, SLAM-seq and RT-qPCR.

CDK12/CDK13 inhibition acts independently of cell cycle-regulated transcription: Our new data demonstrate clearly that downregulation of nascent transcription occurs without suppression of cell cycle-regulated programs, including the pRB/E2F driven G1/S program and the pFoxM1 driven G2 program (revised manuscript Fig. 5F and 5G). This indicates a direct effect of CDK12/CDK13 on global RNAPII elongation across cell cycle phases (Suppl Fig. 6B-C) rather than an indirect consequence of cell cycle control.

Distinct early versus late responses to CDK12/CDK13 inhibition: We find that sustained CDK12/CDK13 inhibition for 24 h or longer induces apoptosis and activates the DNA damage response (DDR), consistent with earlier reports. Our new thorough analyses reveal that early inhibition does not trigger these effects. Comet assays and QIBC analyses of DDR markers (γ H2AX, FK2, pRPA, 53BP1) show no evidence of DNA damage or DDR activation after 6 h of treatment (revised manuscript, Fig 6A, Suppl. Fig. 7A-B). Similarly, cell cycle checkpoint marker p21 and dNTP metabolism regulator RRM2, are not altered (Suppl. Fig. 7C-D). This absence of an early DDR or checkpoint response defines a previously unreported temporal window in which CDK12/CDK13 inhibition suppresses transcription and DNA replication without activating DNA damage signaling.

Suppression of replication origin firing: New DNA fiber assays (Fig. 6D-E) show that CDK12/CDK13 inhibition suppresses replication origin firing in GSCs, directly confirming that the replication arrest observed earlier by EdU incorporation (Fig. 5A) is a specific effect of CDK12/CDK13 inhibition and distinct from CDK9 inhibition.

Collectively, our added results demonstrate that CDK12/CDK13 inhibition in glioblastoma potently suppresses RNAPII phosphorylation and nascent transcription, acting independently of cell cycle-regulated transcriptional programs. In parallel, it induces a profound and early arrest of DNA replication and replication fork progression through a distinct mechanism that does not trigger DNA damage or checkpoint activation. These findings highlight the essential role of CDK12/CDK13 in coordinating RNAPII phosphorylation and transcriptional elongation, and uncover a previously unreported link to DNA replication control.

Comments:

1. Figure 1 shows that CDK12/CDK13 inhibitor decreases cell viability, colony-forming ability, cell

competition assay, and cell migration. The rationale behind choosing CDK12/CDK13 inhibitor in glioma is unclear from the manuscript. What is the CDK12/CDK13 expression in multiple glioma cells vs non-glioma cells? The authors should include normal astrocytes as a control for cell viability assay. The p value is missing in numerous Figure panels.

Authors' comments:

We thank the reviewer for these constructive points. In the revision, we have clarified the rationale for focusing on CDK12/CDK13. Briefly, we found that GSC cell proliferation was most sensitive to CDK12/CDK13 inhibitors compared to other cancer cells (Figure 1A). In contrast, CDK7 and CDK9 inhibitors showed no specificity for glioma cell proliferation, as nearly all tested cancer cells were similarly sensitive to their inhibition (Suppl. Fig. 1A and B). The selective inhibition of glioblastoma cells by CDK12/CDK13 inhibitors was the rationale for choosing CDK12/CDK13 inhibitors for further studies. We have provided expression data comparing glioblastoma and non-glioblastoma cells (Suppl. Fig. 2D). We purchased and established human astrocyte cultures twice, however in both cases, the cultures underwent senescence before we were able to obtain reliable cell viability assays. We have ensured all relevant figure panels include p-values.

Authors' actions:

To address these points, we have made the following changes in the revised manuscript:

Rationale: We have substantially expanded the introduction and the opening of the results section to explain that glioblastomas exhibit hypertranscription driven by RNAPII dysregulation, that transcriptional cyclin dependent kinases (tCDKs) are essential for RNAPII function, and that CDK12/CDK13 inhibition specifically suppresses glioblastoma ("*Inhibition of CDK12/CDK13 arrests glioblastoma cell proliferation and migration,*" revised manuscript, lines 76–79, 87–93, 105–108, 121–124). We also highlight their limited prior evaluation in glioblastoma and the therapeutic potential of targeting glioblastoma transcriptional addiction through CDK12/CDK13 inhibition.

Expression analysis: As suggested, we have included new analyses showing IHC of glioblastoma and control CNS tissues using newly generated specific antibodies, revealing robust nuclear expression of CDK12/CDK13 in tumors (Fig. 2A–B, Suppl. Fig. 2B–C). This is in addition to the earlier CDK12 IHC performed with a commercial antibody in the previous manuscript, which confirmed the same findings and has now been replaced by our CDK12 antibody. We include TCGA/GlioVis survival analysis showing that high CDK13 expression correlates with poorer patient survival, whereas no significant correlation is seen for other tCDKs, reinforcing the rationale for focusing on CDK12/CDK13 (now located in the section, "*CDK12 and CDK13 are expressed in human glioblastoma patients*").

Statistical analyses: All relevant main and supplementary figure panels have been updated to display p-values, with statistical methods detailed in figure legends and Methods (Fig. 1B–C, 2E, 3A, 3E and the Suppl Fig. 1E, 5B–C, 5E, 6E).

2. Figure 2A shows the expression of CDK12 by immunohistochemistry in glioblastoma tissues. Including the non-glioma tissue samples as another control and including a quantification graph with the statistics is essential. In Figure 2B-D, the authors discussed the treatment of glioma patient-

derived organoids with CDK12/CDK13 inhibitors. From the Figure, the organoids are resistant to THZ531 and SR-4835 inhibitors. To rule out this possibility, the immunoblot assay with cleave PARP will be essential to execute. Again, statistics need to be included in Figure 2C-D.

Authors' comments:

We thank the reviewer for these suggestions regarding controls, quantification, and interpretation of the organoid results. We apologize for any lack of clarity. We show IHC controls for non-glioblastoma CNS tissue and secondary antibody are included in Fig. 2A (top right panel) and Suppl. Fig. 2B. Regarding organoid drug responses, we find no evidence of resistance to THZ531 or SR-4835; Fig. 2C–E show that GBOs are sensitive to these inhibitors and to Abemaciclib, with resistance observed only to Lomustine. We acknowledge that the figure presentation could be clarified to avoid misinterpretation. The viability assay used for GBOs (CellTiter-Glo) is optimized for small organoid cultures (<1000 cells) and measures ATP levels as a proxy for viable cells; apoptosis assays for such low cell numbers in 3D cultures are currently unreliable. However, apoptosis induction following THZ531 treatment is validated already in 2D GSC cultures using TUNEL and cleaved PARP assays (Fig. 1E and 1F, previously Fig. 5G and 5E, respectively).

Authors' actions:

In the revised manuscript, we have clarified control data presentation, described the rationale and methodology for our expression and organoid assays, and cross-referenced supporting apoptosis data from 2D cultures to strengthen our interpretation of the organoid results.

Inclusion of non-glioma controls: We have emphasized in the text and figure legends that non-glioblastoma CNS tissue controls are shown in Fig. 2A–B and Suppl. Fig. 2B–C (lines 168–169, 173–174, 827–831, Suppl. legends lines 24–26).

Custom antibody development and expression profiling: We tested several commercial antibodies for CDK13 and did not find them suitable for IHC due to high background and non-specificity. Therefore, we generated and validated specific antibodies for both CDK12 and CDK13. We find heterogeneous nuclear expression for CDK12 and CDK13 in glioblastoma tissue versus restricted expression in inflammatory cells of control CNS tissue (Fig. 2A, B).

Validation of organoid assay sensitivity: We have revised the results text to emphasize that GBOs are sensitive to THZ531, SR-4835, and Abemaciclib, with resistance only to Lomustine. (lines 197–199, 200–202).

Cross-validation of apoptosis induction: To improve flow and clarity, we have reorganized the figures to group responses to CDK12/CDK13 inhibition in glioblastoma. Specifically, we have moved the data showing results from cleaved PARP and γ H2AX immunoblots (previously Fig. 5G, now Fig. 1E) and TUNEL assay (previously Fig. 5E, now Fig. 1F). Altogether, this demonstrates that ≥ 24 h inhibition induces apoptosis and activates the DNA damage response in 2D GSCs.

Statistical analyses: Figure legends and the Suppl. Methods section now include details of the statistical analyses (lines 296–303 in Suppl. Methods).

3. The mouse subcutaneous xenograft experiment was carried out in U87 cells with CDK12/CDK13 inhibitors. However, the glioma stem cells are a more appropriate model for glioma biology, and it is not clear why authors suddenly chose U87 cells. Again, statistics are absent in multiple sub-panels.

Authors' comments:

We appreciate the reviewer's acknowledgment that GSCs are a biologically more faithful model for glioblastoma. In this study, we have used 15 patient-derived glioblastoma models (11 GSCs in Fig. 1 and 4 GBOs in Fig. 2) from two independent research environments to demonstrate that CDK12/CDK13 inhibition compromises glioblastoma proliferation *in vitro*. We have also established orthotopic mouse models using GSCs; however, the available CDK12/CDK13 inhibitors do not sufficiently cross the BBB, making these models unsuitable for *in vivo* efficacy testing. To overcome this limitation, we used a subcutaneous xenograft model, which allows reliable assessment of drug efficacy but requires large tumor cell numbers (~8 million per graft; 50 mice in this study). The slow doubling time of GSCs (40–60 h) made it impractical to generate the required number of cells. We therefore selected the U-87 MG cell line, which can be expanded to the necessary scale and is widely used for proof-of-concept efficacy studies. Importantly, U-87 MG cells display SR-4835 IC₅₀ values comparable to the GSCs, supporting their relevance for *in vivo* evaluation of CDK12/CDK13 inhibition. We have clarified this in the manuscript, please see lines 223–229 in the Results section "*Inhibition of CDK12/CDK13 suppresses GBM migration, invasion and tumor growth in vivo*".

Authors' actions:

Following the reviewer's feedback that GSCs are more appropriate glioblastoma models, and with the constraint that current inhibitors do not cross the BBB, we assessed the effect of CDK12/CDK13 inhibition on GSC migration and proliferation on organotypic mouse brain slices. This model preserves normal brain cytoarchitecture and is a relevant *ex vivo* system for studying glioblastoma migration (<https://doi.org/10.15252/embr.202356964>). Using GSC G7 cells, we show that CDK12/CDK13 inhibition markedly reduces glioblastoma migration and proliferation in this setting (Fig. 3B–C). While not equivalent to the *in vivo* testing in live animals, it provides a close and physiologically relevant approximation supporting the inhibitory effect of CDK12/CDK13 targeting on glioblastoma growth.

In addition, our tolerability studies showed that SR-4835 was toxic at 30 mg/kg (daily or alternate days) but well tolerated at 20 mg/kg daily. In total, 9 mice were tested across these dosing regimens. Mass spectrometry of brain tissue 24 h after the final dose showed concentrations below the limit of quantification in all but one sample (7.6 ng/ml), demonstrating that SR-4835 does not effectively cross the BBB. These results are now presented in a table in Suppl. Methods (Suppl. Methods lines 113–120, Suppl. Table S2).

We have also included appropriate statistical analyses for all results performed on U-87 MG cells, with details provided in the legends of Fig. 3E (lines 852–854) and the Suppl. Methods (lines 296–303).

4. The authors have performed CUT & RUN experiments in G7 cells with CDK12/CDK13 inhibitors and decided to use 1hr and 6hr time points for the assay. Although the inhibitor THZ531 is supposed to

inhibit RNA Pol II phosphorylation at the Ser2 residue, it decreases the Pol II phosphorylation at the ser5 residue quite a bit. Therefore, it is crucial to determine the effect coming from ser2 vs ser5 phosphorylation and gene expression regulation.

Authors' comments:

We thank the reviewer for highlighting the importance of distinguishing between the effects of pSer2 and pSer5 RNAPII phosphorylation on gene expression regulation. Our data confirm that THZ531 treatment in GSCs markedly reduces both pSer2 levels and, to a lesser extent, pSer5 levels. We agree that it is important to determine their relative contributions to transcriptional changes.

Authors' actions:

We have added quantitative correlation analyses (scatter plots and MA plots presented in Suppl. Fig. 3D–E) showing that pSer2 reduction correlates more strongly with decreases in nascent mRNA than pSer5 reduction, indicating that loss of pSer2 is the principal driver of transcriptional inhibition (lines 280–287).

5. There are a lot of supplementary Figures where axes are not labeled correctly or missing.

Authors' comments: Thank you for pointing this out.

Authors' actions:

We have now gone through the supplement and have addressed this in Suppl. Fig. 1A–C, 3G and 4A.

6. The statistical section needs to be included in the manuscript.

Authors' comments: This is now included, see below.

Authors' actions:

We have added a section in the Suppl. Methods summarizing the statistical analyses described in the figure legends of their respective figures. Furthermore, we have provided additional statistical testing in Fig. 1B, 1C, 1F and 1G, 2E, 3C and 3E, 6C and 6E, and the Suppl. Fig. 1E, 5B, 5C, 5E and 6E. The analyses are described in their respective figure legends as well as in the statistical section of Suppl. Methods.

Reviewer #1 (Significance (Required)):

In this manuscript, the authors studied the role of CDK12/CDK13 in glioblastoma and performed extensive studies to uncover the importance of these kinases in glioblastoma. Understanding more mechanistic details of how these kinases are involved in glioma progression will uncover more therapeutic opportunities in glioblastoma.

Reviewer #2 (Evidence, reproducibility and clarity (Required)):

Summary:

Lier et al. present a set of results showing that pharmacological inhibition of CDK12/13, cyclin-dependent kinases that phosphorylate RNA polymerase II (RNAPII), alters the proliferative behavior and transcriptional program of glioblastoma cells. A set of 2D and 3D cultures of patient-derived cell lines with stem-like properties (GSC), as well as subcutaneous xenografts of the U87 cell line, were used as in vitro and in vivo models, respectively. Among the CDKs tested, only CDK13 expression was found to be associated with worse patient survival, while CDK12-immunoreactive cells were detected in patient glioblastoma tissues. The response of GSCs to the CDK12 and CDK13 inhibitor TZH541 included cell cycle blockade and decreased migration. Reduction in RNAPII phosphorylation in TZH541-treated cells was verified using one of the GSC lines. Genome-wide exploration of the transcriptional consequences of TZH541 treatment of 2 GSCs using CUT&RUN and SLAM-seq technologies revealed major transcriptional repression, particularly of genes associated with cell proliferation.

Main comments:

Although I found this study very interesting, I noted points requiring clarification, particularly in order to fully support the authors conclusions. My recommendations focus on the glioblastoma cell biology experiments, my area of expertise.

Authors' comments:

We are grateful for the reviewer's keen interest in our manuscript and appreciate various insightful observations on the challenges within glioblastoma biology. In the revision, we addressed concerns by validating CDK12/CDK13 function in complex tumor-like settings, refining CDK12 and CDK13 expression analyses in patients using in-house-generated antibodies. In addition, we have strengthened the mechanistic insights into how CDK12/CDK13 inhibition disrupts transcription and replication in glioblastoma.

Authors' actions:

To address the reviewer's points, we performed additional experiments and provided explanations in the revised manuscript. In summary, we provide:

Validation in complex glioblastoma models: As proposed by the reviewer, we expanded the description of our patient-derived glioblastoma organoid (GBO) models to highlight how they recapitulate patient tumor complexity (lines 185–189). We also set up and performed migration assays in organotypic mouse brain slices using GSC G7 cells, showing that CDK12/CDK13 inhibition markedly impairs glioblastoma migration (Fig. 3B–C).

CDK12/CDK13 expression profiling in glioblastoma patient tissue: We generated specific in-house antibodies for CDK12 and CDK13 (as suitable commercial antibodies for IHC were unavailable for CDK13) and found nuclear expression of both proteins in glioblastoma patient tissue (Fig. 2A–B). Comparison with Ki-67 staining showed that CDK12/CDK13 expression was independent of proliferative status (Suppl. Fig. 2B).

Altogether, we demonstrate impairment of GSC proliferation, migration, and 3D growth in patient-derived organoid models upon CDK12/CDK13 inhibition.

In addition to strengthening of the biological relevance of CDK12/CDK13 inhibition in glioblastoma, we also provide novel mechanistic insights. Specifically, CDK12/CDK13 inhibition suppresses RNAPII pSer2 phosphorylation and nascent mRNA synthesis, leading to early DNA replication arrest and reduced fork progression without DNA damage or checkpoint activation. This reveals a novel mechanism of replication regulated by transcriptional inhibition via CDK12/CDK13. These findings establish CDK12/CDK13 as promising therapeutic targets whose inhibition disrupts essential transcription and replication in glioblastoma.

1. The rationale for studying only CDK12 expression in patient glioblastoma tissues needs clarification. In contrast with CDK13, the authors found no association between CDK12 expression levels and patient survival (Sup Fig. 1A). Do the authors obtain similar results using independent datasets of glioblastoma tissue transcriptomes (e.g. CGGA)? With regard to the major effect of CDK12/13 inhibition on glioblastoma cell proliferation, determining whether CDK12/13 expression is observed in proliferating areas of the patients' tumor tissues (Ki67 IHC) would help support the authors' conclusion that their "results provide proof-of-concept for the potential of CDK12 and CDK13 as therapeutic targets for glioblastoma". The main data regarding CDK expression the status in patients' tumors and their possible association with patient survival should be rearranged in the same figure and described in the same paragraph of the results.

Authors' comments:

We thank the reviewer for raising this point. In the earlier submission, to study the expression in patient glioblastoma tissue, we were limited by availability of antibodies. We tested and assessed several commercial antibodies, but were not able to identify any CDK13 antibody with sufficient specificity in IHC. We have now raised and validated new antibodies against CDK12 and CDK13.

We understand the reviewer's reasoning and acknowledge that this was not sufficiently clear in the previous submission. Two additional factors further complicate the interpretation: 1) CDK12 and CDK13 have overlapping functions and can compensate for each other's loss (<https://doi.org/10.1126/sciadv.aaz5041>), and 2) Essential genes often do not correlate with patient survival (<https://doi.org/10.1038/s41698-025-01029-x>), as observed for the tCDKs CDK7, CDK9, and CDK12 in our study. Consistently, CDK7 and CDK9 are classified as essential genes in DepMap. However, CDK13 expression did correlate with a poorer survival. As suggested, we have now included Ki-67 immunostaining to mark proliferative tumor regions (Suppl. Fig. 2B).

Authors' actions:

We have reorganized the results so that CDK expression data, survival analysis, and IHC including Ki-67 are presented together in the same section and figures, now clearly separated in three figures:

Functional proliferation assays: We now show dose-response, serum dependency, apoptosis, and CRISPR validation results demonstrating that GSCs are selectively sensitive to CDK12/CDK13 inhibition (New Fig. 1, Suppl. Fig. 1).

Expression analyses: We establish clinical relevance by linking CDK13 expression to poor survival and confirm nuclear CDK12/CDK13 expression in glioblastoma using our newly generated

antibodies, independently of proliferation status assessed through Ki-67 staining (New Fig. 2A–B, Suppl. Fig. 2A, 2D).

2. Fig.1 caption "Inhibition of CDK12/13 specifically affects proliferation of glioma cells" is not entirely consistent with the results. This inhibition also appears to induce cell death, at least in some of the GSC tested, as indicated with cell counts (Fig. 1C., sup Fig.1 G) and an 8-fold increase in the % of apoptotic cells after a 24h-TZH treatment shown in Fig. 5E. All data concerning the effects of TZH on proliferation and survival (including detailed effects on the cell cycle) should be brought together rather than split between the 1st and last figure.

Authors' comments:

We appreciate these comments and incorporate these suggestions in the manuscript to reorganize the figures so that all proliferation and apoptosis data are presented together, with migration and invasion data moved to a dedicated section. We retained the cell cycle data in Fig. 5 to introduce the mechanistic characterization of how CDK12/CDK13 inhibition causes DNA replication arrest.

Authors' actions:

We moved apoptosis and DNA damage response data from the original Fig. 5D-E into Fig. 1, now shown as panels Fig. 1D–E, alongside proliferation and clonogenic assays (Fig. 1B–C). The cell cycle analyses are kept in Fig. 5 as they are part of the mechanistic insights.

In addition to the reviewer's suggestion, we have also moved the motility data from the original Fig. 1 to new Fig. 3A (mentioned above) and grouped migration and invasion assays in new Fig. 3A–C, including organotypic brain slice experiments, to clearly separate these phenotypes from proliferation/survival data.

3. The reason for which serum-treated GSC were used should be explicated (sup Fig. 1C). Serum being usually used to trigger GSC "differentiation", did the authors want to verify whether CDK12/13 inhibitors affected GSC in a specific manner? If yes, it is necessary to demonstrate that serum-treated GSC have lost their stem-like properties.

Authors' comments:

We appreciate the opportunity to clarify the rationale for using serum-treated GSCs. While GSCs are routinely cultured in serum-free media supplemented with N2, B27, EGF, and FGFb, the control cancer cell lines in our study (e.g., breast cancer, HeLa, U2OS) are maintained in serum-containing media. We therefore included a serum-treated GSC condition to test whether the presence of diverse macromolecules in serum could alter drug bioavailability or the cellular response to CDK12/CDK13 inhibition. Our data show that GSCs remain equally susceptible to the inhibitors regardless of serum presence. To minimize serum-induced differentiation, serum was added immediately prior to drug treatment.

Authors' actions:

We have clarified that serum was added immediately before drug treatment to avoid GSC differentiation and that the experiment was designed to assess whether serum components

influence drug response (lines 132–135). Please see Methods (lines 530–531), and the Suppl. Fig. 1E legend (line 17).

4. The viability of patient-derived 3D organoids (GBO) was assessed by measuring ATP production. It is therefore not possible to distinguish between decreased cell proliferation and increased cell death as responsible for the signal decrease. This limitation in the interpretation of the results needs to be made explicit. I was also misled by the use of GBO. This abbreviation is currently used to designate fragments of patient tumor tissue amplified in culture, which retain the cellular heterogeneity and the extracellular matrix of the original tumor and therefore provide an actual ex vivo model of the tumor. To avoid any misunderstanding, I recommend referring to experimental models obtained from dissociated patient-derived cell lines as "3D organoids" or "cellular spheroids", and avoiding to designate them as ex vivo models since they do not recapitulate the complexity of the tumor.

Authors' comments:

We apologize for providing insufficient details concerning our GBO modelling which was performed using the fragments of patient tumor tissue amplified in culture, and we have now updated the description in the methods to improve clarity. Briefly, we derive GBOs from patient tumors by short-term culture of tissue fragments. Such organoids are of a very primary nature and contain extracellular matrix and tumor microenvironment components. To avoid propagation in vitro, we perform implantation of GBOs to immunodeficient animals to create patient-derived orthotopic xenografts (PDOXs). We have established that serial propagation of patient material via series of short-term GBO cultures and PDOXs allows for multiplication of GBM patient tumors without major clonal selection and genetic/phenotypic adaptation (Golebiewska, 2020, DOI: 10.1007/s00401-020-02226-7). To perform robust drug screening ex vivo in GBOs, we further developed a specific protocol based on the material isolated directly from well-established and characterized PDOXs (Oudin, 2021, DOI: 10.1016/j.xpro.2021.100534). The protocol includes reconstitution of 3D GBOs of uniform size, which allows for reliable ex vivo readouts. Importantly, GBM primary cells are able to reassemble into 3D structures of heterogeneous nature, including reconstitution of extracellular matrix.

Authors' actions:

We have updated in the Results section that ATP-based viability assays cannot distinguish between reduced proliferation and increased cell death (lines 195–197). We have revised Methods and Results to specify that GBOs are short-term 3D cultures of freshly isolated patient tumor fragments, not cell-line-derived spheroids (lines 183–189, 531–535).

5. Although the abstract contains a statement indicating that CDK12/13 genetic ablation inhibits cell migration, I did not find the corresponding results in the article. The demonstration that CDK12/13 inhibition decreases cell migration is weaker than the demonstration of its effect on proliferation. Contrary to the experiments evaluating cell proliferation, cell migration was assessed using a single technical approach. Moreover, the method used to assay TZH effects on cell migration rather measures cell motility than cell migration over long distances in a 3D and complex environment as observed in diffuse glioma. Since these data add nothing significant to the article, I would delete them.

Authors' comments:

We thank the reviewer for pointing out the comment in first sentence, which we have addressed in the abstract now. It is correct that strictly speaking, our assays in the earlier submission measured the effect of CDK12/CDK13 inhibition on glioblastoma motility rather than migration. To address the concern and provide a more physiologically relevant evaluation, we have established an organotypic brain slice migration assay, which complements our previous motility data.

Authors' actions:

We have strengthened the dataset by adding a migration assay using GSC G7 cells on organotypic mouse brain slices, which retain brain cytoarchitecture and allow assessment of migration over longer distances in a physiologically relevant environment. These new results show that THZ531-mediated CDK12/CDK13 inhibition significantly reduces glioblastoma migration (Figure 3B-C).

6. In my opinion, the information from the in vivo experiments is limited and should be presented in a supplementary rather than a main figure. The data were obtained with a single cell model, U87 cells of uncertain origin, and using subcutaneous xenografts that provide an environment totally different from the patient's actual tumor. In this context, the data obtained provide little information on the response of cancer cells in a complex and specific environment well known to promote tumor growth and resistance to therapies. I understand that the use of intracerebral xenografts is not feasible, since the inhibitor does not appear to reach the brain. With this technical limitation, an alternative would be to deliver the compound directly inside the brain tumor. A cannula can be implanted into the tumor after it has formed, and connected to an Alzet minipump filled with the drug. These experiments are technically difficult, however, and success is not guaranteed. Another alternative would be to use GBO, as described by Jacob et al (2019) as a surrogate for tumor tissue, provided the authors can obtain tissue fragments from patient surgical resections or intracerebral xenografts of patient-derived cell lines. These alternatives are optional.

Authors' comments:

We appreciate the reviewer's careful considerations of the difficulties in testing currently available compounds *in vivo*, and for the insightful suggestions. We would like to reemphasize the clinical significance of our data in GBOs (please see the response above), which relies on models of equal complexity compared to the model in Jacob's protocol. The model utilized in our study represents compact, complex 3D structures *ex vivo*, and are derived from the GBM patient tumors and propagated as orthotopic patient-derived xenografts.

Taken together, the results from GSCs, GBOs, organotypic brain slices, GBOs, and subcutaneous xenografts provide proof-of-concept that CDK12/CDK13 inhibition can suppress glioblastoma growth. These findings emphasize that novel BBB-penetrant CDK12/CDK13 inhibitors hold strong potential for therapeutic application in patients (please see Discussion, lines 438-440, and 442-446).

Authors' actions:

In the revised manuscript, we have clarified the value and relevance of our GBO models as physiologically meaningful surrogates for patient tumor tissue (lines 184-189).

Minor comments:

- Fig. 4A and Fig. 5E-F: Results from a single experiment? If yes, they must be repeated at least once.

Authors' comments:

The results are representative of a minimum of three independent biological experiments.

Authors' actions:

In the revised manuscript, we have clarified this in the legends of Fig. 4A (lines 863–864) and Suppl. Fig. 6F (previously Fig. 5F) (lines 91–93 in Suppl. legends). Additionally, we have added individual values and statistical analysis to Fig. 1F (previously Fig. 5E), (lines 817–819).

- For the sake of clarity, all y-axes in graphs presenting MTT or CellTiter-Glo assay results should be labeled "cell viability index", as they only provide a measure of overall cell or organoid metabolic activity, and thus an indirect assessment of cell viability.

Authors' comments:

We thank the reviewer for this suggestion.

Authors' actions:

We have incorporated this in the revised manuscript (Fig. 3D and Suppl. Fig. 1A–C, 1E, and 2F).

- Statistical analyses are missing for 3 of the 4 cell lines presented in Figure 1F.

Authors' comments:

We thank the reviewer for bringing this to our attention.

Authors' actions:

We have supplied Fig. 3A with statistical analyses for the remaining cell types. Details of statistical analyses are provided in the Suppl. Methods (lines 28–30).

- Some GO terms are truncated in sup Fig. 3.

Authors' comments:

We appreciate that the reviewer brought this to our attention.

Authors' actions:

We have corrected Suppl. Fig. 3F and 4J.

- The legend to Fig. 5B-D shows the mean and SD of 2 replicates. Please show individual points.

Authors' comments:

We thank the reviewer for this suggestion.

Authors' actions:

We have revised Fig. 5B and Suppl. Fig. 5D–E to include the individual points.

- *Sup Fig1 D-F: unit of concentration is missing (M?)*

Authors' comments:

We appreciate the reviewer's careful observation, and for highlighting this.

Authors' actions:

We have corrected this in the current Suppl. Fig. 1A–1C.

Reviewer #2 (Significance (Required)):

Significance:

Despite growing interest in the roles of CDK12/13 roles in cancers and their targeting for cancer therapy, their involvement in glioblastoma growth remains unexplored. The results presented in this study outline the potential of CDK12/13 inhibition in controlling the growth of glioblastoma, at least in vitro, and thus provide meaningful information on its potential usefulness for this aggressive brain tumor with a high proliferation rate. Obtaining the full proof-of-concept that CDK12/13 constitute relevant targets for glioblastoma therapies will however require additional experiments demonstrating efficacy of CDK12/13 inhibition in complex environments, as encountered in the patients' tumor.

16th Sep 2025

Dear Dr. Pandey,

Thank you for the submission of your revised manuscript to EMBO Molecular Medicine. We have now heard back from the two referees who agreed to re-evaluate your manuscript. As you will see from their reports below, both referees are supportive of the study, but referee #2 remains critical about the organoid model and the correlation between CDK12/CDK13 expression and proliferative status. Please address this concerns in an additional and final round of revision. In addition, for the next submission please adhere to the journal's formatting requirements. Please check our Author Guidelines for more information <https://www.embopress.org/page/journal/17574684/authorguide#manuscriptpreparation>

Please amend the following:

- Submit a complete Author Checklist. <https://www.embopress.org/pb-assets/embosite/EMBO%20Press%20Author%20Checklist-1642513524327.xlsx>
- Supplementary methods should be moved to the main Methods section.
- Supplementary figures should be Expanded View figures with their legends placed at the end of the main manuscript file after main figure legends. Update their callouts in the main text.
- Movie files should be renamed to Movie EV1 etc. and zipped with their legend as .txt file. Update their callouts in the main text.
- In Methods, please include a statement that informed consent was obtained from the patient and that the experiments conformed to the principles set out in the WMA Declaration of Helsinki and the Department of Health and Human Services Belmont Report.

We would welcome the submission of a revised version within three months for further consideration. Please let us know if you require longer to complete the revision.

I look forward to receiving your revised manuscript.

Yours sincerely,

Zeljko Durdevic

Zeljko Durdevic
Senior Editor
EMBO Molecular Medicine

*** Instructions to submit your revised manuscript ***

- 1) a .docx formatted version of the manuscript text (including Figure legends and tables)
- 2) Separate figure files*
- 3) supplemental information as Expanded View and/or Appendix. Please carefully check the authors guidelines for formatting

Expanded view and Appendix figures and tables at
<https://www.embopress.org/page/journal/17574684/authorguide#expandedview>

4) a letter INCLUDING the reviewer's reports and your detailed responses to their comments (as Word file).

5) The paper explained: EMBO Molecular Medicine articles are accompanied by a summary of the articles to emphasize the major findings in the paper and their medical implications for the non-specialist reader. Please provide a draft summary of your article highlighting

6) Author contributions: the contribution of every author must be detailed in a separate section.

7) EMBO Molecular Medicine now requires a complete author checklist (<https://www.embopress.org/page/journal/17574684/authorguide>) to be submitted with all revised manuscripts. Please use the checklist as guideline for the sort of information we need WITHIN the manuscript. The checklist should only be filled with page numbers where the information can be found. This is particularly important for animal reporting, antibody dilutions (missing) and exact values and n that should be indicated instead of a range.

8) Every published paper now includes a 'Synopsis' to further enhance discoverability. Synopses are displayed on the journal webpage and are freely accessible to all readers. They include a short stand first (maximum of 300 characters, including space) as well as 2-5 one sentence bullet points that summarise the paper. Please write the bullet points to summarise the key NEW findings. They should be designed to be complementary to the abstract - i.e. not repeat the same text. We encourage inclusion of key acronyms and quantitative information (maximum of 30 words / bullet point). Please use the passive voice. Please attach these in a separate file or send them by email, we will incorporate them accordingly.

You are also welcome to suggest a striking image or visual abstract to illustrate your article. If you do please provide a jpeg file 550 px-wide x 300-600px high.

9) A Conflict of Interest statement should be provided in the main text

10) Please note that we now mandate that all corresponding authors list an ORCID digital identifier. This takes <90 seconds to complete. We encourage all authors to supply an ORCID identifier, which will be linked to their name for unambiguous name identification.

Currently, our records indicate that the ORCID for your account is 0000-0001-5493-3197.

Link Not Available

11) Include a Reagents and Tools Table as part of the Methods section, which can be downloaded from our author guidelines (<https://www.embopress.org/page/journal/17574684/authorguide#structuredmethods>)

Photos 400-800 DPI

*Additional important information regarding figures and illustrations can be found at
<https://bit.ly/EMBOPressFigurePreparationGuideline>. See also figure legend preparation guidelines:
<https://www.embopress.org/page/journal/17574684/authorguide#figureformat>

**** Reviewer's comments ****

Referee #1 (Comments on Novelty/Model System for Author):

I have reviewed the revised manuscript, and I am satisfied with the changes. I have no further comments. Thank you.

Referee #2 (Comments on Novelty/Model System for Author):

Despite growing interest in the roles of CDK12/13 roles in cancers and their targeting for cancer therapy, their involvement in glioblastoma growth remains unexplored. The results presented in this study outline the potential of CDK12/13 inhibition in controlling the growth of glioblastoma.

Referee #2 (Remarks for Author):

The authors have addressed part of my concerns, as listed below.

- The authors' response confirms my understanding that the GBO model is a 3D in vitro model, rather than an ex vivo model, as it does not correspond to tumor tissue organoids. Rather, it corresponds to spheroids obtained from tumor cell suspensions. Contrary to GBO derived from culture of tumor tissue fragments, such spheroids reproduce only in part the complexity of patient tumors, in which tumor cells with variable genomic alterations and various types of non-cancerous cells coexist. As described in the paper cited in the authors' response (Oudin et al. 2021), the spheroids used in the present study are derived from serial intracranial implantations of tumor fragments from patients, a process involving clonal selection. Organoid formation is then achieved by seeding human tumor cells purified from the xenografts in U-shaped culture wells. I easily understand that due to technical limitations, this model is the most suitable for accurate pharmacological evaluation. Naming the model with a non-misleading term would simply clarify the limitations of the study without calling into question the value of the data presented in this paper that broaden our understanding of glioblastoma growth. Therefore, I reiterate my recommendation that any ambiguity surrounding the nature of this model be resolved by referring to it by a name other than 'GBO' in the text and figures.
- The rationale behind using serum supplementation in some of the experiments has been clarified.
- Additional experiments provide further evidence that CDK12/CDK13 inhibition impairs cell migration.
- The development of in-house antibodies enabled the authors to convincingly document CDK12 and CDK13 expression in patients' glioblastoma tissues. However, the affirmation that CDK12/CDK13 expressions are independent of proliferative status is not clearly evident in Supplementary Figure 2B, which depicts only Ki67 immunostaining. I recommend quantifying the number of cells expressing either CDK12 or CDK13 in distinct areas of tumors with low or high Ki67 expression. These quantified results could then be used to perform correlation analyses between Ki67 and either CDK12 or CDK13 expression.
- The reorganization of the figures facilitates the reading and analysis of the results.

Rev_Com_number: RC-2023-02082

New_manu_number: EMM-2025-22462-T

Corr_author: Pandey

Title: CDK12/13 inhibition disrupts transcriptional elongation and fork progression in glioblastoma

Full Revision

Manuscript number: EMM-2025-22462-T

Corresponding author(s): Deo Prakash, Pandey & Mads, Lerdrup

1. General statements

We appreciate the efforts the reviewers have taken to reconsider our study, and are grateful for the suggestions on how to improve our manuscript.

In response to the reviewers comments we have supplemented our immunohistochemistry experiments with quantification and a correlation analysis to clarify the relationship between proliferative status and CDK12 positive cells in tumor tissue. Furthermore, we have revised our use of the nomenclature used to describe GBM organoids and further clarified the methodology of organoid acquisition.

2. Point by point description of the revisions

Color code:

Reviewer comment

Authors' comments

Authors' actions

Reviewer #1 (Comments on Novelty/Model System for Author):

I have reviewed the revised manuscript, and I am satisfied with the changes. I have no further comments. Thank you.

We sincerely thank the reviewer for reviewing our manuscript.

Reviewer #2 (Comments on Novelty/Model System for Author):

Despite growing interest in the roles of CDK12/13 roles in cancers and their targeting for cancer therapy, their involvement in glioblastoma growth remains unexplored. The results presented in this study outline the potential of CDK12/13 inhibition in controlling the growth of glioblastoma.

Reviewer #2 (Remarks for Author):

The authors have addressed part of my concerns, as listed below.

- The authors' response confirms my understanding that the GBO model is a 3D in vitro model, rather than an ex vivo model, as it does not correspond to tumor tissue organoids. Rather, it

corresponds to spheroids obtained from tumor cell suspensions. Contrary to GBO derived from culture of tumor tissue fragments, such spheroids reproduce only in part the complexity of patient tumors, in which tumor cells with variable genomic alterations and various types of non-cancerous cells coexist. As described in the paper cited in the authors' response (Oudin et al. 2021), the spheroids used in the present study are derived from serial intracranial implantations of tumor fragments from patients, a process involving clonal selection. Organoid formation is then achieved by seeding human tumor cells purified from the xenografts in U-shaped culture wells. I easily understand that due to technical limitations, this model is the most suitable for accurate pharmacological evaluation. Naming the model with a non-misleading term would simply clarify the limitations of the study without calling into question the value of the data presented in this paper that broaden our understanding of glioblastoma growth. Therefore, I reiterate my recommendation that any ambiguity surrounding the nature of this model be resolved by referring to it by a name other than 'GBO' in the text and figures.

Authors' comments:

We would like to thank the reviewer for this additional feedback and understand the need to clarify the terminology. We also agree that our patient-derived organoid model that we applied for the drug screening does not fully recapitulate the complex environments of patient tumors. We appreciate that reviewer recognizes the technical limitations surrounding the experimental setup and agrees on the suitability of the utilized model for pharmacological evaluation.

Authors' actions:

Based on the reviewer's feedback from the current and the first round of reviewing, we have decided to avoid the ambiguity by omitting the term "GBOs" entirely. Instead, we refer to the model as 'organoids' when mentioned after its initial description. Furthermore, we agree with the reviewer's suggestion to omit the term *ex vivo* and have updated the manuscript accordingly. Finally, we have updated the results section to explicitly mention the removal of the tumor microenvironment from the mouse glioblastoma tissue, and that this was a deliberate choice to allow standardization of functional readouts.

- The rationale behind using serum supplementation in some of the experiments has been clarified.

Authors' comments:

We thank the reviewer for directing our focus to this point and are pleased to learn that the reviewer finds it satisfactory.

- Additional experiments provide further evidence that CDK12/CDK13 inhibition impairs cell migration.

Authors' Comments:

We appreciate the reviewers' recognition of this addition to the study.

- The development of in-house antibodies enabled the authors to convincingly document CDK12 and CDK13 expression in patients' glioblastoma tissues. However, the affirmation that CDK12/CDK13 expressions are independent of proliferative status is not clearly evident in Supplementary Figure 2B, which depicts only Ki67 immunostaining. I recommend quantifying the number of cells expressing either CDK12 or CDK13 in distinct areas of tumors with low or high

Ki67 expression. These quantified results could then be used to perform correlation analyses between Ki67 and either CDK12 or CDK13 expression.

Authors' comments:

We appreciate the reviewer's insightful suggestion of quantifying cells expressing CDK12/CDK13 in specific tumor areas having high or low Ki67 expression, to clarify the relationship between CDK12/CDK13 expression and proliferative status.

Authors' actions:

Based on Ki67 expression, we defined areas of high, intermediate, and low proliferation. In the corresponding CDK12 areas, we quantified the number of CDK12 positive cells. We utilized this data to perform a correlation analysis (lines 682-691 in Methods), which we have included in the revised manuscript. In summary, we find a moderate but significant correlation between CDK12 expression and Ki-67 expression in glioblastoma tissues, please see lines 184-187 and Figure EV2D in the revised manuscript.

-The reorganization of the figures facilitates the reading and analysis of the results.

Authors' comments:

We are pleased to learn that the reviewer finds the reorganization satisfactory.

3. Response to the editorial decision letter

We greatly appreciate the editors' interest in our manuscript, and we have adhered to the requirements to the best of our ability.

Color code:

1. *Editors' comment*
2. *Authors' comments*
3. *Authors' actions*

- Submit a complete Author Checklist. <https://www.embopress.org/pb-assets/embosite/EMBO%20Press%20Author%20Checklist-1642513524327.xlsx>

Authors' actions:

We have completed the authors checklist to accompany the revised manuscript.

- Supplementary methods should be moved to the main Methods section.

Authors' actions:

We have integrated the supplemental methods in the main Methods section of our revised manuscript.

- Supplementary figures should be Expanded View figures with their legends placed at the end of the main manuscript file after main figure legends. Update their callouts in the main text.

Authors' actions:

In the revised submission, we have restructured the supplementary figures to be Expanded View figure 1-5, and moved the remaining two supplementary figures to the Appendix. We have updated the callouts in the revised manuscript.

- Movie files should be renamed to Movie EV1 etc. and zipped with their legend as .txt file. Update their callouts in the main text.

Authors' actions:

We have renamed the movie files and zipped them with their respective legends as requested. In the main text of the revised manuscript, we have updated the callouts.

- In Methods, please include a statement that informed consent was obtained from the patient and that the experiments conformed to the principles set out in the WMA Declaration of Helsinki and the Department of Health and Human Services Belmont Report.

Authors' actions:

In the revised manuscript, we have provided this statement in the method section, lines 731-732.

15th Jan 2026

Dear Dr. Pandey,

Thank you for the submission of your revised manuscript to EMBO Molecular Medicine. I am pleased to inform you that we will be able to accept your manuscript pending the following final amendments:

1) Authors:

- We note name discrepancies between the manuscript and our submission system: Aparajita Dutta vs. Aparajita Datta, Sugith B. Badugu vs. Sugith B. Babu and Rune F. Johansen vs. Rune Johansen Forstrøm. Please correct.
- The email correspondence to Johanne Egge Rinholm could not be delivered. Please update their e-mail addresses and make sure to enter correct e-mail addresses for all authors in our submission system.

2) Figures:

- We note that some images/panels are reused. In Figure 2C control organoids images are the same for THZ531, Abemaciclib and SR-4835. Please clarify.
- Please rename the EV figure files to Figure EV1, etc.

3) Source Data: All provided source data files are empty. Please make sure to upload a full set of correct source data files for all main figures.

4) In the main manuscript file, please do the following:

- Please address all comments suggested by our data editors listed below:

o Figure legends:

1. Please define the annotated p values ****/**/*/* as well as provide the exact p-values for the same in the legend of figure 6E, EV4 E as appropriate.
2. Please note that the exact p values are not provided in the legends of figures 1C, G; 2E, 3A, C, E; 6C, EV1 D, E; EV5 G.
3. Please indicate the statistical test used for data analysis in the legends of figures 3A, 4F, G, L; EV2 A, D; EV3 F, G.
4. Please note that the box plots need to be defined in terms of minima, maxima, centre, bounds of box and whiskers, and percentile in the legends of figures 3A, G.
5. Please note that information related to n is missing in the legends of figures 3C, F, G; 6E.
6. Please note that the error bars are not defined in the legends of figures 2D, F.
7. Please note that the scale bar needs to be defined for figures 3B, 6B, EV5 F.
8. Please note that scale bar and its definition are missing for figure EV4 G.

- Remove "Total character count".

- Indicate in legends exact n and exact p values, not a range, along with the statistical test used. To keep the figures "clear" some authors found providing an Appendix table Sx with all exact p-values preferable. You are welcome to do this if you want to.

- Rename "Conflict of interest declaration" to "Disclosure Statement & Competing Interests" and place it after the "Acknowledgements". We updated our journal's competing interests policy in January 2022 and request authors to consider both actual and perceived competing interests. Please review the policy <https://www.embopress.org/competing-interests> and update your competing interests if necessary.

- Author contributions: Please remove it from the manuscript and specify author contributions in our submission system. CRediT has replaced the traditional author contributions section because it offers a systematic machine-readable author contributions format that allows for more effective research assessment. You are encouraged to use the free text boxes beneath each contributing author's name to add specific details on the author's contribution. More information is available in our guide to authors:

<https://www.embopress.org/page/journal/17574684/authorguide#authorshipguidelines>

- In the expanded view figure legends, rename the figures to "Figure EV1" etc.

- Please use the following format to report the accession number of your data:

[data type]: [full name of the resource] [accession number/identifier] ([doi or URL or identifiers.org/DATABASE:ACCESSION])

Please check "Author Guidelines" for more information.

<https://www.embopress.org/page/journal/17574684/authorguide#availabilityofpublishedmaterial>

5) Funding: Merge it with Acknowledgments. Please make sure that information about all sources of funding are complete in both our submission system and in the manuscript. RESCO, Nansenfondet and SPARK program from University of Oslo; Medical Student Research Program, University of Oslo; INTER/TRANSCAN22/17612718/PLASTIG, University of Oslo, Nansenfondet and Unifor Frimed are currently missing in our submission system.

6) Movies: Please remove their legends from the manuscript file and correct the callouts to Movie EV1 and Movie EV2.

7) Tables: Please upload all Dataset files as EV tables in .xlsx format. Rename them to Table EV1 etc., place their legend in the file and update their callouts in the main text.

8) Appendix: Please rename Appendix Supplementary Figure 1 etc. to Appendix Figure S1, etc.

9) Synopsis:

- Synopsis image: Please provide the image as a high-resolution jpeg file 550 px-wide x 300-600 pixels high.

10) As part of the EMBO Publications transparent editorial process (see our Editorial at <http://embomolmed.embopress.org/content/2/9/329>), EMBO Molecular Medicine will publish online a Review Process File (RPF) to accompany accepted manuscripts. This file will be published in conjunction with your paper and will include the anonymous referee reports, your point-by-point response and all pertinent correspondence relating to the manuscript. Let us know if you want to remove or not any figures from it prior to publication. Please note that the Authors checklist will be published at the end of the RPF.

11) Please provide a point-by-point letter INCLUDING my comments as well as the reviewer's reports and your detailed responses (as Word file).

I look forward to reading a new revised version of your manuscript as soon as possible.

Yours sincerely,

Zeljko Durdevic

Zeljko Durdevic
Senior Editor
EMBO Molecular Medicine

*** Instructions to submit your revised manuscript ***

When preparing your revised manuscript, please refer to our guidelines: <https://link.springer.com/journal/44321/submission-guidelines#cms-Revised-submissions>. We perform an initial quality control of all revised manuscripts before re-review; failure to include requested items will delay the evaluation of your revision.

We require:

2) Individual production quality figure files as .eps, .tif, .jpg (one file per figure). For guidance, download the 'Figure Guide PDF': <https://media.springernature.com/original/springer-cms/rest/v1/content/27825798/data/v1>.

3) A .docx formatted letter INCLUDING the reviewers' reports and your detailed point-by-point responses to their comments. As part of the EMBO Press transparent editorial process, the point-by-point response is part of the Review Process File (RPF), which will be published alongside your paper.

4) A complete author checklist, which you can download from our author guidelines. Please insert information in the checklist that is also reflected in the manuscript. The completed author checklist will also be part of the RPF.

6) It is mandatory to include a 'Data Availability' section after the Materials and Methods. Before submitting your revision, primary datasets produced in this study need to be deposited in an appropriate public database, and the accession numbers and database listed under 'Data Availability'. Please remember to provide a reviewer password if the datasets are not yet public.

7) For data quantification: please specify the name of the statistical test used to generate error bars and P values, the number (n) of independent experiments (specify technical or biological replicates) underlying each data point and the test used to calculate p-values in each figure legend. The figure legends should contain a basic description of n, P and the test applied. Graphs must include a description of the bars and the error bars (s.d., s.e.m.).

9) Our journal encourages inclusion of *data citations in the reference list* to directly cite datasets that were re-used and obtained from public databases. Data citations in the article text are distinct from normal bibliographical citations and should directly link to the database records from which the data can be accessed. In the main text, data citations are formatted as follows: "Data ref: Smith et al, 2001" or "Data ref: NCBI Sequence Read Archive PRJNA342805, 2017". In the Reference list, data citations must be labeled with "[DATASET]". A data reference must provide the database name, accession number/identifiers and a resolvable link to the landing page from which the data can be accessed at the end of the reference.

12) Author contributions: You will be asked to provide CRediT (Contributor Role Taxonomy) terms in the submission system. These replace a narrative author contribution section in the manuscript.

13) A Conflict of Interest statement should be provided in the main text.

14) Every published paper includes a 'Synopsis' to further enhance discoverability. Synopses are displayed on the journal webpage and are freely accessible to all readers. They include a short stand first (maximum of 300 characters, including space) as well as 2-5 one-sentences bullet points that summarizes the paper. Please write the bullet points to summarize the key NEW findings. They should be designed to be complementary to the abstract - i.e. not repeat the same text. We encourage inclusion of key acronyms and quantitative information (maximum of 30 words / bullet point). Please use the passive voice. Please attach these in a separate file or send them by email, we will incorporate them accordingly.

15) Include a Reagents and Tools Table as part of the Methods section, which can be downloaded from our author guidelines.

Photos 400-800 DPI

*Additional important information regarding figures and illustrations can be found at
<https://media.springernature.com/original/springer-cms/rest/v1/content/27825798/data/v1>

The authors addressed the remaining editorial issues.

9th Feb 2026

Dear Dr. Pandey,

We are pleased to inform you that your manuscript is accepted for publication and is now being sent to our publisher to be included in the next available issue of EMBO Molecular Medicine.

You may qualify for financial assistance for your publication charges - either via a Springer Nature fully open access agreement or an EMBO initiative. Check your eligibility: <https://link.springer.com/journal/44321/how-to-publish-with-us>

Zeljko Durdevic
Senior Editor
EMBO Molecular Medicine

>>> Please note that it is EMBO Molecular Medicine policy for the transcript of the editorial process (containing referee reports and your response letter) to be published as an online supplement to each paper. If you do NOT want this, you will need to inform the Editorial Office via email immediately. More information is available here: <https://link.springer.com/partners/embo-press/editorial-policies#Peer%20review>